# CHAMMI-75: PRE-TRAINING MULTI-CHANNEL MODELS WITH HETEROGENEOUS MICROSCOPY IMAGES

Vidit Agrawal[1,2], John Peters[1,2], Tyler N. Thompson[1,2], Mohammad Vali Sanian[3,4], Chau Pham[5], Nikita Moshkov[6], Arshad Kazi[1,2], Aditya Pillai[1,2], Jack Freeman[1], Byunguk Kang[7,8], Samouil L. Farhi[8], Ernest Fraenkel[7,8], Ron Stewart[1], Lassi Paavolainen[3,4], Bryan A. Plummer[5], and Juan C. Caicedo[1,2]

[1]Morgridge Institute for Research, Madison, WI, USA
[2]University of Wisconsin-Madison, Madison, WI, USA
[3]Institute for Molecular Medicine Finland (FIMM), Helsinki, Finland
[4]University of Helsinki, Helsinki, Finland
[5]Boston University, Boston, MA, USA
[6]Institute of Computational Biology, Helmholtz Munich, Neuherberg, Germany
[7]Massachusetts Institute of Technology, Cambridge, MA, USA
[8]Broad Institute of MIT and Harvard, Cambridge, MA, USA

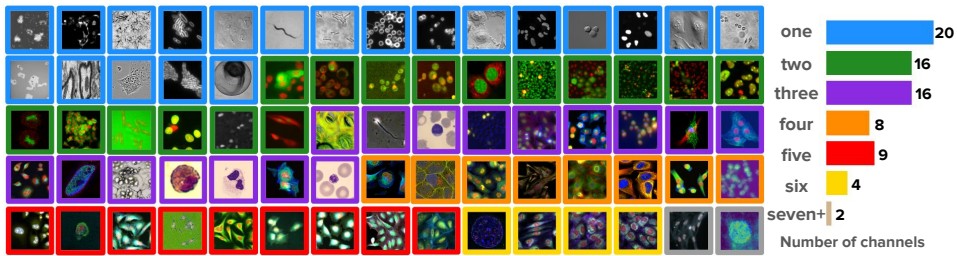

Figure 1: CHAMMI-75: a heterogenous dataset of multi-channel microscopy images. Pseudo-color images representing all 75 sources, and edge colors indicating their original number of channels.

## Abstract

Quantifying cell morphology using images and machine learning is a powerful strategy to study the response of cells to treatments. However, models used to quantify cellular morphology are typically trained with a single microscopy imaging type. This results in specialized models that cannot be reused across biological studies because the technical specifications do not match (e.g., different number of channels). Here, we present CHAMMI-75, an open access dataset of heterogeneous, multi-channel microscopy images from 75 diverse biological studies. We curated this resource from publicly available sources to investigate cellular morphology models that are channel-adaptive and can process any microscopy image type. Our experiments show that training with CHAMMI-75 can improve performance in multi-channel bioimaging tasks primarily because of its high diversity in microscopy modalities. This work paves the way to create the next generation of cellular morphology models for biological studies.

## 1 INTRODUCTION

Microscopy is a versatile scientific tool in experimental biology and allows researchers to acquire images of cells under controlled experimental conditions. Unlike natural images, which are consistently acquired and stored in a three-channel, RGB format, microscopy images can have a varied number of channels; anywhere from one to dozens, each encoding

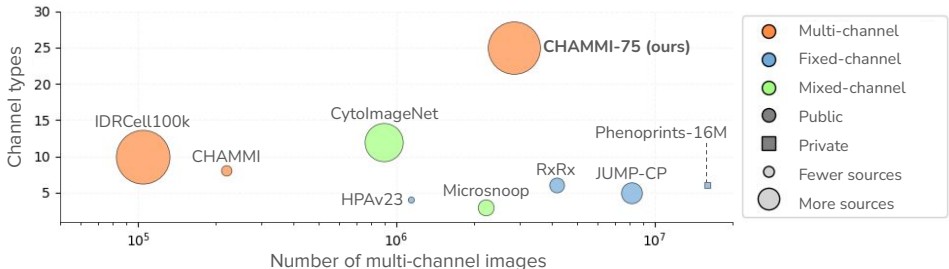

Figure 2: Comparison of existing microscopy datasets used for representation learning of cell morphology. `CHAMMI-75` is the largest dataset of multi-channel microscopy images. Others datasets: IDRCell100k (Bourriez et al., 2024), CHAMMI (Chen et al., 2023), CytoImageNet (Hua et al., 2021), HPAv23 (Gupta et al., 2024), Microsnoop (Xun et al., 2024), RxRx Recursion (2025), JUMP-CP (Chandrasekaran et al., 2023a), and Phenoprints-16M (Kenyon-Dean et al., 2024).

a different type of signal. Deep learning is widely adopted to analyze microscopy images (Moen et al., 2019; Volpe et al., 2023; Pratapa et al., 2021; Xing et al., 2017), but the most common strategy to create such models is to modify architectures developed for RGB images by changing and fixing the number of channels according to the problem (Doron et al., 2023; Gupta et al., 2024). This limits the ability to reuse models from experiment to experiment, or to pre-train large-scale models that accumulate universal knowledge of cellular biology.

Multi-channel imaging models have emerged to address the limitations of existing vision architectures, enabling the processing of varied number of channels at test time (Kraus et al., 2024; Bourriez et al., 2024; Pham & Plummer, 2024; Pham et al., 2025). A common trend in such architectures is the separation of channels as individual modalities, resulting in flexible, variable-length inputs with diverse channel types. While these initiatives represent an important step towards creating foundation models for cellular imaging, most of them are still small-scale, proof-of-concept experiments. The main reason is the lack of standardized and well-curated datasets to properly test these ideas at large scale. Multi-channel microscopy images are publicly available, but it is not straightforward to put them together in a single, useful resource for machine learning research because of their technical differences, varied formats, and inconsistent metadata.

We present a new multi-channel microscopy image dataset that we call `CHAMMI-75` (Figure 1). Our dataset is larger than prior work (Figure 2) and combines diverse heterogeneous sources of biological images into a single resource to investigate cellular morphology. `CHAMMI-75` contains more technical and biological variation than typically used to train cellular imaging models (Figure 3). For example, `CHAMMI-75` includes images of 16 organisms and 223 cell lines collected with different microscopes, at different resolutions, and with different numbers of channels. Our goal was to obtain a representative visual sample of the cellular biology universe that can be observed with microscopy. With the accelerated progress of machine learning, this diversity comes as a valuable source of information to train the next generation of multi-channel imaging models. Ultimately, we believe that foundation models for microscopy imaging should be able to understand cell morphology at all scales, in all cell types, and independently of the imaging technology used for observation.

In this work, we address the data gap that hinders the development of generalizable models for microscopy imaging. Progress in machine learning is frequently possible thanks to rigorous, large-scale curation efforts to create open datasets. For example, ImageNet (Deng et al., 2009) and LAION (Schuhmann et al., 2022) fundamentally shifted the focus of representation learning research and supported breakthroughs that would have been impossible without such data. In computational biology, challenging and realistic data is similarly necessary to advance the field. While significant progress has been made with fixed-channel image models, many open problems still exist to achieve a general understanding of cellular states regardless of the imaging technology. `CHAMMI-75` is the first resource to integrate heterogeneous multi-channel imaging data at this scale in a way that directly facilitates the investigation of these challenges.

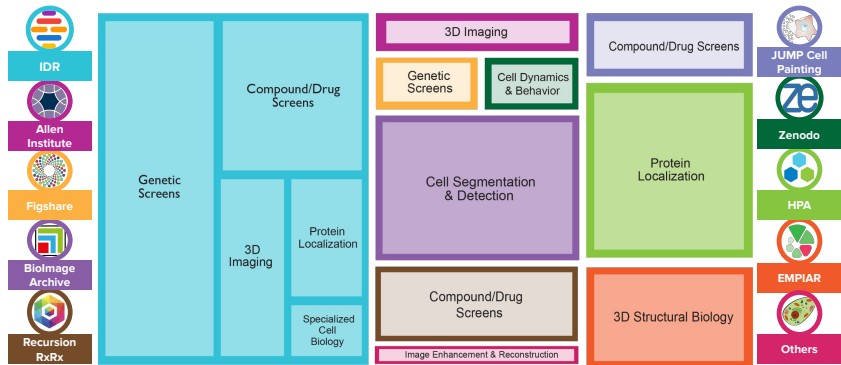

Figure 3: Diversity of sources and biological studies in `CHAMMI-75`. The treemap illustrates the distribution of images according to the hosting platforms they were obtained from (colors) and the type of biological study (inner rectangles). We sampled from 18 different sources to ensure broad coverage of biological study types.

The purpose of `CHAMMI-75` is to facilitate the creation of models for cell phenotyping, which is the task of quantifying morphological differences between cellular states (e.g., healthy vs. perturbed cells). This goal is different from cell segmentation, a critical task in bioimage analysis that has seen the emergence of robust foundation models (Stringer et al., 2021; Pachitariu & Stringer, 2022; Pachitariu et al., 2025; Archit et al., 2025; Marks et al., 2025). These models demonstrate success in generalized prediction of cell masks across modalities, which is often a required step before cell phenotyping. `CHAMMI-75` provides the data necessary to train and evaluate models that can detect subtle biological classification signals, not available through segmentation alone.

The contributions of our work are: (1) a large, heterogeneous dataset of multi-channel microscopy images, with 25 channel types and high technical and biological variation. (2) Three new datasets and benchmarks to evaluate multi-channel model performance, which represent real-world contemporary biological studies with novel channel combinations (e.g., 14-channel images). (3) A systematic experimental evaluation to investigate the usefulness of our dataset as a pre-training resource, using self-supervised learning. The results show that the diversity of `CHAMMI-75` is a strength for yielding models that perform well in a variety of challenging biological tasks. (4) A top performing model pre-trained with `CHAMMI-75`, which we call MorphEm. (5) The data, code, and models are publicly available for future research and development in the field: `https://github.com/CaicedoLab/CHAMMI-75`.

## 2 Related Work

Figure 2 provides a high-level overview comparison of our dataset and prior work. The biological imaging community has a long tradition of publicly sharing the datasets obtained in their studies. Prominent initiatives to store and share imaging datasets include the Image Data Repository (IDR) (Williams et al., 2017), the Bioimage Archive (Hartley et al., 2022), and the Cell Painting Gallery (Weisbart et al., 2024). Other major projects create large datasets of cells exposed to many treatments to serve as a map to investigate various biological questions. These include the RxRx datasets (Sypetkowski et al., 2023), along with the JUMP-CP dataset (Chandrasekaran et al., 2023b), which cover a large number of genetic and chemical perturbations. Similarly, the Human Protein Atlas (Thul et al., 2017), the OpenCell project (Cho et al., 2022), and Allen Institute for Cell Science (Viana et al., 2023) have created image datasets to map the localization of proteins in human cells. While all these resources have multi-channel images and are publicly available, they are typically used and analyzed separately. Closest to ours is IDRCell100k (Bourriez et al., 2024), a dataset created for multi-channel model development containing 100K images from 79 different sources. Inspired by their work, we extend the effort by collecting 30× more images with superior data curation, quality control, and metadata annotations.

Figure 4: Content and distribution of images in `CHAMMI-75` according to the integrated metadata. a) Selected metadata fields and summary statistics of their diversity. b) Sparse, long-tail distribution of channel configurations across studies. None of the studies has all channel types, and none of the channels is used in all studies.

Prior work has explored channel-adaptive imaging (e.g., (Bao et al., 2023; Bourriez et al., 2024; Kraus et al., 2024; Pham & Plummer, 2024; Pham et al., 2025)), and where the number and configuration of the input channels is not fixed, they either train on private data (Kraus et al., 2024; Kenyon-Dean et al., 2024), or on small, publicly available datasets for proof-of-concept experiments (Chen et al., 2023). Other work either focuses on weakly- or self-supervised methods over fixed channels (Caicedo et al., 2018; Moshkov et al., 2024; Doron et al., 2023; Kim et al., 2025), or developing channel-agnostic methods (Pawlowski et al., 2016; Ando et al., 2017; Xun et al., 2024; Morelli et al., 2025; Lian et al., 2025; De Lorenci et al., 2025). These studies provided insight into how image-based profiling can reveal the response of cells to biological reagents (Caicedo et al., 2017; 2016), and it can be scaled to high-throughput experiments using robotic automation (Boutros et al., 2015). As our dataset contains varying numbers of channels, our experiments use representatives of both channel-agnostic and channel-adaptive modeling.

## 3 The `CHAMMI-75` Dataset

`CHAMMI-75` is a collection of 2,792,462 fields-of-view (FoV) for pre-training sampled from 74 publicly available microscopy imaging studies (Table A.1), and 2,474,875 FoV held out for test and organized in six benchmarks[1] (Table A.1). Each FoV in the pre-training set is a multi-channel microscopy image with up to seven channels, and may have a single or multiple cells (Figure 1). Our dataset is the largest multi-channel image dataset for model pre-training in microscopy. Compared to existing microscopy imaging datasets, `CHAMMI-75` draws from many more sources and contains many more channel types (Figure 2).

The process of building `CHAMMI-75` followed three major phases. We started with a *data acquisition* phase for selecting hosting platforms, identifying source datasets, downloading raw data, and standardizing image formats. Next, we conducted a *metadata integration* phase where all the experimental details of the downloaded images were collected, organized, and integrated across all sources. The final phase was *data curation*, which focused on strategically sampling the most representative and informative images for learning. This phase transformed the downloaded image set collected in the first phase into `CHAMMI-75`.

**Data acquisition.** The `CHAMMI-75` dataset was collected from 18 different hosting platforms that store biological images from published scientific studies (Figure 3). The top platforms of imaging data according to the number of studies are: The Image Data Resource (IDR) (Williams et al., 2017), Zenodo (CERN & OpenAIRE, 2013), Mendeley Data (Elsevier, 2025), and Figshare (Singh, 2011). The full list of sources is provided in Appendix A.1. These repositories provide access to image sets created by biological labs around the world with open data licenses; 97% of datasets in our collection have a Creative Commons license (60% CC BY 4.0, 12% CC0 1.0). The 75 studies are listed in Table A5.

From these hosting platforms, we selected 75 source datasets with highly heterogeneous biological and technical settings. Biological diversity was defined in terms of organisms and types of perturbations, while technical diversity was defined in terms of microscopy techniques and imaging settings (Figure 4). More details in Appendix A.2.

---

[1]Four of the six benchmarks share images with the pre-training set. Total of unique sources: 76.

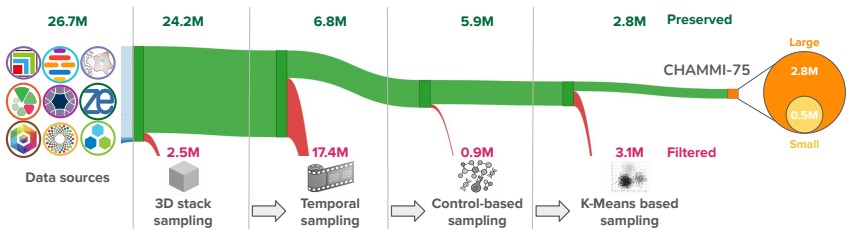

Figure 5: Dataset curation pipeline. Left: the dataset downloaded from the hosting platforms. Middle: the metadata is used to filter redundancy by randomly sampling a few: 2D slices from 3D images, frames from live microscopy videos, and wells from control conditions. Right: content-based clustering selects diverse, high-quality images as a final step to create `CHAMMI-75`.

**Metadata integration.** We collected metadata information to prepare consistent information for guiding the sampling of high-quality, diverse images for the pre-training set. Not all images are equally informative or useful, and we targeted biological and technical variables reported in the original studies for filtering purposes. As a result, the metadata table has 22 columns organized in 6 groups depending on the type of information (Figures 4 and A11). These values were obtained from the original sources by parsing information from (1) available resource descriptor files, (2) values encoded in image filenames, and (3) details reported in the source paper. Most studies have a scientific publication that describes experimental details, and for part of the metadata preparation, we used large-language models to assist with the identification and organization of certain information. The final metadata was parsed programmatically using deterministic rules and then manually curated and validated. (Appendix A.3).

**Data curation.** Data curation is an important effort to successfully scale representation learning (Vo et al., 2024; Siméoni et al., 2025). Our goal was to select a diverse sample that represents the main factors of variation in the ∼26M downloaded images. To this end, we systematically sampled images following information in the metadata table. Figure 5 illustrates the main steps of the dataset curation, designed to minimize redundancy and select informative images from all sources. The main filtering steps include 3D stack sampling, temporal sampling, control-based sampling, and K-means based filtering (Appendix A.4). By carefully curating images in this way, we selected 2.8M diverse multi-channel images from multiple sources, resulting in the pre-training dataset `CHAMMI-75`.

**Content annotations.** We analyzed image contents to produce annotations about the locations of single cells. Specifically, we configured cell segmentation pipelines for all sources using Cellpose (Stringer et al., 2021; Pachitariu & Stringer, 2022) primarily to detect the nucleus (when available) or the cell body. We recorded the center of mass coordinates for 1.8B single cells found in the 2.8M multi-channel images. This information is useful to generate crops that contain visible cells during training, thus avoiding empty, noisy, or non-informative regions. More details are reported in the Appendix A.5.

## 4 EVALUATION BENCHMARKS

Here we briefly describe six datasets and associated evaluation tasks that we adopt for performance evaluation, which represent real-world, contemporary image-based biological studies. Details about the benchmarks are reported in Figure 6 and in Appendix A.6.

**CHAMMI benchmark.** This benchmark (Chen et al., 2023) contains about 220K multi-channel images in three subsets with different numbers and types of channels, and includes six out-of-domain generalization tasks. The tasks include: cell-cycle stage classification (3 channels, WTC-11 dataset), protein localization classification (4 channels, HPA dataset), and replicate treatment retrieval (5 channels, LINCS Cell Painting dataset). We follow the standard evaluation protocol and report the CHAMMI score.

**IDR-0017.** A chemical-genetic interaction study (Breinig et al., 2015) that includes 122,880 two-channel images of 12 cell-lines treated with 1,280 compounds. The task is to identify

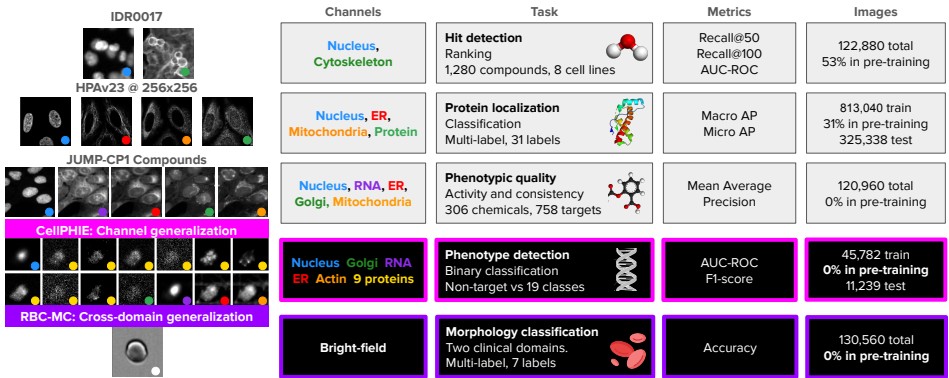

Figure 6: Illustration of the evaluation sets in `CHAMMI-75`. We adopted existing benchmarks (HPAv23 (Gupta et al., 2024) and JUMP-CP1 (Chandrasekaran et al., 2023a)), and introduce three new from real biological studies (IDR0017 (Breinig et al., 2015), CellPHIE (Kang et al., 2025), and RBC-MC (Doan et al., 2020)) for channel and cross-domain generalization.

hits among the combinatorial experiments by ranking gene-compound combinations that are likely to have a large effect with respect to controls. Ground truth hits were obtained from the original study, and performance is evaluated using AUROC as well as recall at the top 50 and 100 items of the ranked list.

**HPAv23 at 256×256.** HPA studies the localization of proteins in human cells by classifying images into the correct localization category (Ouyang et al., 2019; Le et al., 2022). We adopt the dataset HPAv23 with 1.1 million images of single cells and annotations for protein localization classification in 19 or 31 classes. The original image size is 1024x1024, and we reduced it to 256x256 to accelerate benchmarking (Apendix G.4.2). We preserve the rest of the benchmark intact, and report average precision scores accordingly.

**JUMP-CP1 Compounds.** We adopt a subset of the JUMP-CP (Chandrasekaran et al., 2023b) dataset, which includes 24K five-channel images of U2OS cells perturbed with 306 chemicals at two time points. There are two ranking tasks in this benchmark: 1) phenotypic activity: identify compounds with a response significantly different from negative controls, and 2) phenotypic consistency: measure whether groups of biologically related compounds are clustered together. We report mean average precision for both tasks.

**CellPHIE.** A pooled genetic perturbation screen to investigate Huntington's Disease (Kang et al., 2025), this dataset contains 57K 14-channel images of single cells perturbed with one of 19 genes. A non-targetting control is used as a reference to determine the effect of perturbations. The task is to classify single cells in a binary classification setting: non-targetting vs perturbed gene. Images from this dataset are not included in the pretraining dataset, resulting in a novel channel combination benchmark.

**Red Blood Cell Morphology Classification (RBC-MC).** A set of 130,560 single-channel bright-field images (48x48 pixels) of RBCs obtained via imaging flow cytometry from two distinct clinical sites (Swiss and Canadian blood banks) (Doan et al., 2020). The task is a multi-class classification into seven clinically relevant morphological categories associated with blood quality. The benchmark employs a cross-domain validation protocol with a linear probe trained on data from one site and tested on the other (Swiss vs Canadian).

## 5 EXPERIMENTS AND RESULTS

We identify two main multi-channel strategies that are emerging in recent work: (1) Bag of channels (BoC) models, which train backbone networks that read one channel at a time and then concatenate features for downstream tasks. Examples include Microsnoop (Xun et al., 2024), uniDINO (Morelli et al., 2025), and DINO-BoC (De Lorenci et al., 2025). (2) Multi-channel attention (MCA) models, which unravel channel tokens into a single, long sequence to model cross-channel associations. These include Channel-ViT (Bao et al.,

Table 1: Performance of models across benchmarks. Column legend: **mcT**: Multi-channel training, **mcM**: Multi-channel mechanism (✋ manual selection of channel combination, 👜 bag of channels, 🚂 variable-length sequence of channel tokens), **Dataset**: Pre-training dataset (🔬 Microscopy Images, 📷 Natural Images), **CM**: CHAMMI, **H**: HPAv23 256x256, **J1**: JUMP-CP1, **J2**: JUMP-CP2, **I**: IDR0017, **CP: CellPHIE (channel generalization)**, **R: RBC-MC (cross-domain generalization)**. All models are ViT-small and have been trained with SSL, except for the top-line results of SubCell (gray row), which is a collection of specialized, larger models (ViT-base) trained with multi-objective, weakly supervised learning. The result of best performing SubCell model for each benchmark is presented. Bold numbers are best result among SSL methods.

| | **Model Characteristics** | | | **Benchmarks ↑** | | | | | | |
| Model | mcT | mcM | Dataset | CM | H | J1 | J2 | I | CP | R |
|---|---|---|---|---|---|---|---|---|---|---|
| SubCell | ✗ | ✋ | 🔬 HPAv23 | *53.38* | *69.33* | *77.60* | *07.44* | *75.37* | *71.23* | *59.10* |
| DINOv2 | ✗ | 👜 | 📷 LVD-142M | 37.93 | 53.76 | 75.84 | **07.03** | 75.22 | 72.27 | 59.41 |
| OpenPhenom | ✓ | 🚂 | 🔬 RxRx + JUMP | 38.22 | 49.13 | 74.26 | 04.99 | 75.19 | 75.56 | 64.43 |
| IDRCell | ✓ | 👜 | 🔬 IDRCell100k | 37.38 | 44.05 | 72.37 | 04.97 | 75.19 | 79.14 | 55.85 |
| MorphEm (ours) | ✓ | 👜 | 🔬 CHAMMI-75 | **48.75** | **58.87** | **76.32** | 06.79 | **75.52** | **80.51** | **68.34** |

2023), CA-MAE (Kraus et al., 2024), and ChA-MAEViT (Pham et al., 2025), among others (Pham & Plummer, 2024; Bourriez et al., 2024). Here, we benchmark ViT models with the BoC approach, and analyze their scalability properties together with Channel-ViT as a representative architecture of MCA models.

## 5.1 Benchmarking experiments

We trained a ViT-small model with DINO-BoC (De Lorenci et al., 2025) on `CHAMMI-75` and compare the results against existing models used to obtain representations of cellular images. Performance is measured in the six benchmarks described in Section 4.

**Baselines.** We consider state-of-the-art pre-trained models that have been recently released for cellular image analysis. We start with SubCell (Gupta et al., 2024), a suit of ViT-base models trained with the HPAv23 dataset using multi-objective, weakly supervised learning. SubCell has four fixed-channel models (one 2ch, two 3ch, one 4ch) trained in two modes (MAE-Cells, ViT-ProtS). These models exhibit excellent performance in downstream tasks; however, manual configuration is needed to decide a channel combination and model type. The variation of results between SubCell models is substantial (Appendix F.1.1), making its usage challenging and computationally expensive to test in practice. We also evaluate OpenPhenom (Kraus et al., 2024), a channel adaptive ViT-small model trained on five and six-channel Cell Painting images, and DINOv2 (Oquab et al., 2023), which is a fixed RGB channel model adapted for multi-channel images using BoC. Finally, we trained a model with IDRCell100K (Bourriez et al., 2024), a multi-channel microscopy image dataset close to ours in number of sources (79 vs 75) but smaller (100k multi-channel images).

**Results.** Table 1 shows pre-training channel-adaptive architectures with SSL yields models that are generally useful in many tasks, regardless of the number of channels. SubCell sets top-line results across several benchmarks; its strong performance may be explained by factors such as training with biological objectives, larger ViT models, channel specialization, and manual selection of best results across their different settings. Our BoC model —which we call *MorphEm* (for Morphology Embeddings)— trained with `CHAMMI-75` obtained the best performance in six out of seven performance metrics, demonstrating generalization in tasks with varying channel configurations. The same model trained with IDRCell100k (a multi-channel microscopy image dataset) underperforms in most tasks, suggesting that `CHAMMI-75` contains additional informative images and higher quality data for learning. OpenPhenom also underperforms in several tasks, and while it is channel adaptive, it was trained exclusively with Cell Painting data (RxRx & JUMP-CP). Overall, our model exhibits strong performance in challenging tasks thanks to a combination of simple methods and high-quality, well-curated data.

## 5.2 Channel Generalization and Cross Modality Transfer

We explore channel generalization through the most challenging and realistic scenarios encountered in biological practice. First, generalization to *novel channel combinations*, which is very frequent in laboratories by combining known channels in novel ways. The CellPHIE (**CP**) benchmark, with its unique 14-channel configuration, serves as a real-world test for this capability. Second, generalization to *novel modalities and domains*. The RBC-MC (**R**) benchmark tests this by using single-channel bright-field imaging flow cytometry (modality) to classify red blood cell morphologies. Importantly, its paired cross-domain evaluation across clinical sites, challenges models to encode biologically relevant features.

The results are summarized in the last two columns of Table 1, and indicate that our MorphEm model trained with `CHAMMI-75` yields the best performance in both generalization tasks. This performance is in contrast with highly specialized models like SubCell, which lags behind in these novel conditions. Specifically, our smaller, SSL-trained model outperforms the larger, WSL-trained SubCell by 13% in CellPHIE and 15% in RBC-MC. This result suggests that the scale and diversity of `CHAMMI-75` are important factors for achieving robust channel and domain generalization.

## 5.3 Factors impacting representation learning

To understand which factors of variation in `CHAMMI-75` drive robust representation learning, we performed six targeted data ablation studies, systematically separating training images according to variables in the metadata. In each ablation, we train a DINO-BoC model on each subset and then measure their relative difference in CHAMMI scores (Figure 7). The results reveal a hierarchy of factors that influence model performance. First, specialized vs. heterogeneous data: models trained only with the target data under-perform by 27% relative difference, while models trained with `CHAMMI-75` can improve performance up to 38%. This indicates that heterogeneous data facilitates learning representations that are more expressive and are less prone to overfitting.

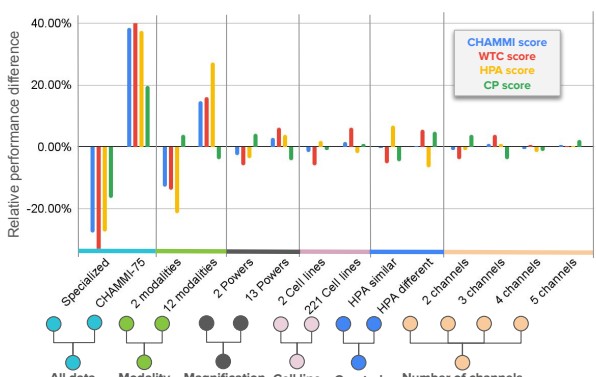

Figure 7: Data ablation study with six factors of variation. Heterogeneity and microscopy modality are major drivers of performance improvement in `CHAMMI-75` (Appendix B).

The second most influential factor is the diversity of imaging modality (microscopy type). A model trained only on the two dominant modalities (fluorescence and epi-fluorescence) underperforms by 13% relative to a model trained on the other 12 less-represented modalities, which improves performance by 15% relative. The next most impactful factor is microscope magnification with relative performance difference of ∼ 3%. These results indicate that broad exposure to technical variation makes a model more generalizable. Other factors, such as training without the most common cell lines (U2OS and A549), or limiting the number of channels (e.g., training only on studies with up to two, three, or four channels), resulted in minor performance changes (∼ 1%). This indicates that while highly diverse cell lines and varied channel counts contribute to overall robustness, the diversity of the physical imaging process (modality) is the main variable that improves representation learning.

## 5.4 Scalability analysis

We evaluate `CHAMMI-75` as a pre-training resource through three main evaluations. (a) Dataset scaling: we assess performance on the CHAMMI benchmark (six out-of-distribution,

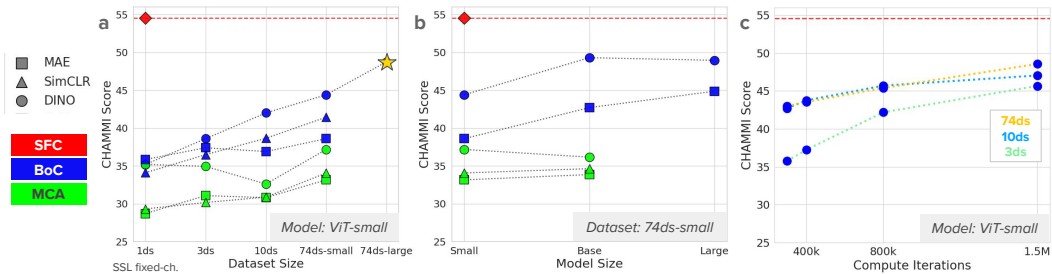

Figure 8: Scaling properties of self-supervised, multi-channel models. (a) Dataset, (b) model, and (c) compute scaling results on the CHAMMI benchmark. SFC supervised fixed-channels, BoC bag of channels, MCA multi-channel attention, ⭐ MorphEm model, scaled with best configuration.

phenotype matching tasks) by increasing the amount of multi-channel images used for training. The baseline is the average of three specialized models (1ds, ∼33K images, 3, 4, or 5 channels). We then combine sets: 3 CHAMMI subsets (3ds, 100K images, 8 channels), 10 datasets (10ds, 178K images, 14 channels), and a sample of CHAMMI-75 (74ds-small, 560K images, 25 channels). The full CHAMMI-75 set (74ds-large, 2.8M images, 25 channels) was only used to train MorphEm for final benchmarking. (b) Model scaling: we fix the dataset size (74ds-small) and increase model parameters. In all cases, we train ViT models, BoC (single channel) and MCA (variable channels), with 224x224 input size using three SSL algorithms (SimCLR, MAE, DINO). (c) Compute iterations: we keep dataset and model sizes fixed and evaluate the impact of using more compute to train models. Hyperparameters are kept consistent where possible. After SSL training, features are extracted from the CHAMMI test sets with frozen weights and no finetuning. See Appendix C and G.2.

Consistent with prior work, models trained with SSL benefit from more data. Comparing SSL performance against fully-supervised, specialized models (an upper bound), our results (Figure 8a,b) confirm that as data and model size increase, SSL models approach the specialized supervised performance. This shows that a single, scaled, unsupervised multi-channel model can be highly competitive. The primary performance determinant is the multi-channel strategy (BoC or MCA), followed by the SSL algorithm and model size. BoC models outperform MCA in the SSL regime, suggesting difficulty in learning cross-channel correlations in an unsupervised way. BoC models are also easier to scale, as MCA requires 3X to 5X more GPU hours due to longer sequences (Figure G23). We also observe that when keeping compute constant (total iterations per epoch, Figure 8c), BoC models with access to more data can yield improved performance. Based on these findings, we trained MorphEm, a BoC ViT-small model using DINO on the full CHAMMI-75 dataset (5X larger), requiring 2,352 GPU hours. This yielded a 9.8% relative improvement over the best result from the dataset scaling evaluation, with potential for further gains using a larger ViT in future work.

## 5.5 Model type and disentanglement analysis

**Model design choices.** The experimental results suggest the following factors impacting performance: (1) Multi-channel strategy: the BoC approach was more effective and scalable under the SSL regime, yielding up to 19% relative improvement over MCA, at a fraction of the computational cost. (2) SSL method: among the tested methods, DINO was the top-performing algorithm, showing a ∼ 15% relative improvement over MAE and a ∼ 7% relative improvement over SimCLR (Figure 8a). (3) Model size: we observed consistent performance gains from model scaling. Moving from ViT-small to ViT-large resulted in a relative improvement of 10% (Figure 8b). (4) The use of weak supervision, even with noisy labels, improved performance by 1-19% relative to the baseline in the MCA-SSL setting. These results provide guidance for future research on multi-channel models as the field continues to bridge the gap between self-supervised and supervised performance.

**Disentanglement and batch correction.** We adopted the batch correction benchmark of JUMP-CP (Arevalo et al., 2024) to evaluate the ability of models to separate technical from biological variation. The results indicate that features learned with `CHAMMI-75` are more robust to technical variation (requiring less batch correction) and maximize biological signal. Following correction with the Seurat CCA algorithm (Stuart et al., 2019), our features achieve high separation of biological clusters from technical batches, performing better than features extracted with CellProfiler and those learned from the IDRCell100K dataset. A qualitative analysis of the feature space (Appendix D) using UMAP visualizations of single-channel features confirm that the primary clustering factor is the source study (technical domain). This shows that models are sensitive to technical variation, and confirms the need to apply batch correction to maximize biological signal (Appendix E).

## 5.6 WEAKLY SUPERVISED LEARNING

The state-of-the-art in image-based profiling introduces prior knowledge of biological conditions using weakly supervised learning (WSL) with objectives such as treatment or protein classification (Gupta et al., 2024; Moshkov et al., 2024). While these pretext tasks are not focused on the primary model use, they improve performance as long as the labels are clean and consistent enough to facilitate learning. Table 2 evaluates two WSL models trained with 74ds-small using the reagent identifier as a target in the loss function (see Appendix A.7). The models evaluated here implement multi-channel attention because biological guidance may require a combination of channels to learn meaningful representations. The WSL method ChA-MAEViT (Pham et al., 2025) obtains best performance, demonstrating the ability of `CHAMMI-75` to benefit multiple learning strategies.

Table 2: Performance of WSL models ChA-MAEViT (Pham et al., 2025) and MCA-SupC (Khosla et al., 2020) across benchmarks, with a DINO-based SSL model (MCA-SSL) included for comparison.

| Model | Benchmarks ↑ | | | | | |
|---|---|---|---|---|---|---|
| | **CM** | **H** | **J1** | **J2** | **I** | **CP** |
| MCA-SSL | 37.18 | 37.46 | 61.72 | 01.85 | 24.40 | 67.93 |
| MCA-SupC | 36.15 | 51.81 | 72.72 | 05.20 | **25.21** | 75.59 |
| ChA-MAEViT | **38.72** | **56.92** | **74.99** | **05.53** | 24.99 | **76.12** |

## 6 CONCLUSION AND LIMITATIONS

**Conclusion.** We introduced `CHAMMI-75`, a dataset of heterogeneous multi-channel microscopy images for pre-training cellular image analysis models. The dataset combined images from 75 sources, resulting in a curated, high-quality resource that has more biological and technical variation than other microscopy datasets used in prior work. Our dataset paves the way to investigate multi-channel imaging models at large scale, and can facilitate the development of models that work seamlessly across biological labs and image configurations. We adopted existing benchmarks and introduced new ones to continue evaluating progress in the ever-changing world of microscopy. Our experimental results show that `CHAMMI-75` can be used to scale models that yield strong performance across the benchmarks. Importantly, improved generalization can be achieved when training models with heterogeneous imaging modalities, resulting in models that perform well in novel channel combination settings, novel modalities, and that learn to disentangle biological and technical variation. We make our top-performing model, MorphEm, publicly available.

**Limitations.** The experimental evaluation was limited by the computational resources available in academic institutions. This work was focused on high-quality data curation and leaves the investigation of novel methods for multi-channel modeling for future research. The collected metadata is informative but noisy despite best efforts to standardize all sources. This represents a real-world condition of imaging data, and generates challenges for supervised methods or similar types of studies. While the presented dataset is large and representative of many microscopy imaging types and biological conditions, it is also sparse and does not cover all relevant variables in balanced way.

ETHICS STATEMENT

We have no ethical concerns with our work. All the datasets sampled in our 75 studies have been cited with their licenses. The licensing information can be obtained in the Appendix A5. All these datasets can be used for furthering scientific research, and understanding foundation models in microscopy but not for commercial purposes.

REPRODUCIBILITY STATEMENT

For reproducibility purposes, we will release our best model weights, dataset and the code on a suitable platform upon acceptance of the paper. We hope that the community can use this work as a resource for further research into foundation models for cellular microscopy.

CODE AND DATA AVAILABILITY

The code to reproduce the results of is available on GitHub:
https://github.com/CaicedoLab/CHAMMI-75.

The data can be downloaded from the following S3 bucket through AWS Open Data:
https://registry.opendata.aws/chammi/.

The best ViT-small model trained on `CHAMMI-75` is available on Hugging Face:
https://huggingface.co/CaicedoLab/MorphEm.

The tutorials on how to use the code and model are available at:
https://github.com/CaicedoLab/CHAMMI-75/blob/main/aws-tutorials

ACKNOWLEDGEMENTS

The authors are grateful to Gantugs Atarsaikhan and Zayn Kayali for numerous contributions to the early stages of this project. We thank the CHTC team at the University of Wisconsin–Madison for their guidance, support, and access to compute infrastructure. The authors also wish to acknowledge CSC - IT Center for Science, Finland, for awarding this project access to the LUMI supercomputer, owned by the EuroHPC Joint Undertaking, hosted by CSC (Finland) and the LUMI consortium. This work was supported by the National Science Foundation under Award No 2348683 (to JCC) and 2134696 (to BAP), and by the National Library of Medicine, NIH Grant 5T15LM007359. Funding was provided by the Research Council of Finland with awards 340273, 346604, 359907 (to LP, MVS), and the University of Helsinki Doctoral Programme in Computer Science (to MVS). This work was also supported by an unrestricted grant from Meta AI, the AWS Open Data Sponsorship Program, and the BU AI Research Resource Program.

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

## A  DATASET

The `CHAMMI-75` dataset contains two versions for pre-training: a small dataset for development and ablation studies, and a large dataset for scaling models. Table A3 reports the number of images in each set.

Table A3: Number of multi-channel images in the small and large pretraining sets

| Set | # Single Channel Images | # Multi-Channel Images |
|---|---|---|
| Small | 1,679,765 | 560,558 |
| Large | 8,029,583 | 2,849,483 |

### A.1  DATA SOURCES

The following are the official data sources or hosting platforms where we obtained microscopy images to curate the `CHAMMI-75` dataset in alphabetical order: Allen Institute (Viana et al., 2023), AMSActa (of Bologna, 2025), BBBC (Ljosa et al., 2012), BioImage Archive (Hartley et al., 2022), BioStudies, Cell Painting Gallery (Weisbart et al., 2024), Edmond (Society, 2025), EMPIAR (Iudin et al., 2023), Figshare (Singh, 2011), GitHub (Kaggle, 2025), HPA (Thul et al., 2017), IDR (Williams et al., 2017), Mendeley Data (Elsevier, 2025), OSF (OSF, 2025), Recursion (Recursion, 2025), University of Reading Research Data Archive (of Reading, 2025), Zenodo (CERN & OpenAIRE, 2013). Each of the 75 datasets are listed in Table A5 with their corresponding licenses, number of channels per image, and the number of multi-channel images sampled from the original dataset.

Table A4: Pre-training Image Dataset Information

| ID | Dataset | License | Channels | Images |
|---|---|---|---|---|
| hpa0018 | HPAv18 (HPA) (Thul et al., 2017) | CC BY-SA 4.0 | 4 | 113,545 |
| hpa0023 | HPAv23 (HPA) (Thul et al., 2017) | CC BY-SA 4.0 | 4 | 249,999 |
| idr0001 | IDR0001 (IDR) (Graml et al., 2014) | CC BY 4.0 | 2 | 112,476 |
| idr0002 | IDR0002 (IDR) (Hériché et al., 2014) | CC BY 4.0 | 2 | 27,092 |
| idr0003 | IDR0003 (IDR) (Breker et al., 2013) | CC BY-NC-SA 3.0 | 3 | 43,072 |
| idr0005 | IDR0005 (IDR) (Toret et al., 2014) | CC BY-NC-SA 3.0 | 1 | 34,698 |
| idr0006 | IDR0006 (IDR) (Fong et al., 2013) | CC BY-NC-SA 3.0 | 2 | 125,508 |
| idr0007 | IDR0007 (IDR) (Srikumar et al., 2013) | CC BY-NC-SA 3.0 | 2 | 3,213 |
| idr0008 | IDR0008 (IDR) (Rohn et al., 2011) | CC BY-NC-SA 3.0 | 4 | 23,159 |
| idr0009 | IDR0009 (IDR) (Simpson et al., 2012) | CC BY 4.0 | 3 | 86,116 |
| idr0010 | IDR0010 (IDR) (Doil et al., 2009) | CC BY 4.0 | 2 | 44,761 |
| idr0011 | IDR0011 (IDR) (Ledesma-Fernández & Thorpe, 2015) | CC BY 4.0 | 3 | 17,893 |
| idr0012 | IDR0012 (IDR) (Fuchs et al., 2010) | CC BY-NC-ND 4.0 | 3 | 22,182 |
| idr0013 | IDR0013 (IDR) (Neumann et al., 2010) | CC0 1.0 | 1 | 208,474 |
| idr0017 | IDR0017 (IDR) (Breinig et al., 2015) | CC BY-NC-ND 4.0 | 2 | 65,048 |
| idr0022 | IDR0022 (IDR) (Koedoot et al., 2019) | CC BY 4.0 | 1 | 54,690 |
| idr0025 | IDR0025 (IDR) (Stadler et al., 2012) | CC BY-SA 3.0 | 4 | 564 |
| idr0028 | IDR0028 (IDR) (Pascual-Vargas et al., 2017) | CC BY 4.0 | 4 | 33,707 |
| idr0030 | IDR0030 (IDR) (Sero & Bakal, 2017) | CC BY 4.0 | 4 | 38,017 |
| idr0033 | IDR0033 (IDR) (Rohban et al., 2017) | CC BY 4.0 | 5 | 16,024 |
| idr0035 | IDR0035 (IDR) (Ljosa et al., 2013) | CC BY 4.0 | 3 | 11,403 |
| idr0037 | IDR0037 (IDR) (Vigilante et al., 2019) | CC BY 4.0 | 5 | 17,617 |
| idr0056 | IDR0056 (IDR) (Stojic et al., 2020) | CC BY 4.0 | 5 | 50,177 |
| idr0069 | IDR0069 (IDR) (Caldera et al., 2019) | CC BY-NC 4.0 | 3 | 82,812 |
| idr0072 | IDR0072 (IDR) (Schormann et al., 2020) | CC BY 4.0 | 2 | 68,642 |
| idr0080 | IDR0080 (IDR) (Way et al., 2021) | CC0 1.0 | 5 | 11,425 |
| idr0081 | IDR0081 (IDR) (Georgi et al., 2020) | CC BY 4.0 | 2 | 10,040 |
| idr0086 | IDR0086 (IDR) (Miron et al., 2020) | CC BY 4.0 | 6 | 15,283 |
| idr0088 | IDR0088 (IDR) (Cox et al., 2020) | CC BY-NC 4.0 | 3 | 151,021 |
| idr0089 | IDR0089 (IDR) (Fischl et al., 2020) | CC BY 4.0 | 3 | 8,077 |
| idr0093 | IDR0093 (IDR) (Müller et al., 2021) | CC BY 4.0 | 5 | 44,858 |
| idr0094 | IDR0094 (IDR) (Ellinger et al., 2021) | CC0 1.0 | 1 | 65,828 |
| idr0115 | IDR0115 (IDR) (Otsuka et al., 2023) | CC BY 4.0 | 2 | 13,273 |
| idr0120 | IDR0120 (IDR) (German et al., 2021) | CC BY 4.0 | 5 | 33,390 |
| idr0123 | IDR0123 (IDR) (Mota et al., 2022) | CC BY 4.0 | 7 | 3,157 |
| idr0128 | IDR0128 (IDR) (Olszewski et al., 2022) | CC BY 4.0 | 2 | 9,539 |
| idr0129 | IDR0129 (IDR) (Olszewski et al., 2022) | CC BY 4.0 | 2 | 10,441 |
| idr0130 | IDR0130 (IDR) (Olszewski et al., 2022) | CC BY 4.0 | 2 | 3,785 |
| idr0133 | IDR0133 (IDR) (Dahlin et al., 2023) | CC BY 4.0 | 5 | 23,149 |
| idr0140 | IDR0140 (IDR) (Ho et al., 2022) | CC BY 4.0 | 2 | 16,721 |
| idr0145 | IDR0145 (IDR) (Ho et al., 2023) | CC BY 4.0 | 2 | 16,338 |
| jump0001 | cpg0016-jump (Cell Painting Gallery) (Chandrasekaran et al., 2023b) | CC0 1.0 | 5 | 146,741 |
| nidr0001 | ALFI (Figshare) (Antonelli et al., 2023) | CC BY 4.0 | 1 | 2,146 |
| nidr0002 | Stomata (Mendeley Data) (Dey et al., 2023) | CC BY 4.0 | 3 | 1,004 |
| nidr0003 | S-BIAD531 and S-BIAD840 (BioImage Archive) (Jones et al., 2024) | CC0 1.0 | 1 | 15,056 |

| ID | Dataset | License | Channels | Images |
|---|---|---|---|---|
| | Continued from previous page | | | |
| nidr0004 | White blood cells (Figshare) (Bodzas et al., 2023) | CC BY 4.0 | 3 | 14,565 |
| nidr0005 | BriFiSeg (Zenodo) (Mathieu et al., 2022) | CC BY 4.0 | 1 | 1,029 |
| nidr0006 | VirtualStaining (Figshare) (Trizna et al., 2023) | CC BY 4.0 | 4 | 252 |
| nidr0007 | S-BIAD300 (BioImage Archive) (Yakimovich & Galimov, 2021) | CC0 | 1 | 31,558 |
| nidr0008 | BBBC030 (BBBC) (Koos et al., 2016) | CC BY 4.0 | 1 | 60 |
| nidr0009 | Parasites (Mendeley Data) (Zhang et al., 2022) | CC BY 4.0 | 3 | 297 |
| nidr0010 | DICimages (University of Reading Research Data Archive) (Kempster et al., 2022) | CC BY 4.0 | 1 | 132 |
| nidr0011 | PerceptiLabs/bacteria (GitHub) | CC0 1.0 | 1 | 366 |
| nidr0012 | DeepBacs1 (Zenodo) (Spahn et al., 2022) | CC BY 4.0 | 1 | 99 |
| nidr0013 | DeepBacs2 (Zenodo) (Spahn et al., 2022) | CC BY 4.0 | 1 | 60 |
| nidr0014 | DeepBacs3 (Zenodo) (Spahn et al., 2022) | CC BY 4.0 | 1 | 34 |
| nidr0015 | EVICAN (Edmond) (Schwendy et al., 2020) | CC BY 4.0 | 1 | 4,361 |
| nidr0016 | Fluorescent Neuronal Cells v2 (AMSActa) (Clissa et al., 2024) | CC BY 4.0 | 2 | 1,809 |
| nidr0017 | LIVECell (Figshare) (Edlund et al., 2021) | CC BY 4.0 | 1 | 5,165 |
| nidr0018 | Omnipose (OSF) (Cutler et al., 2022) | CC BY-NC 3.0 | 1 | 791 |
| nidr0019 | BBBC042 (BBBC) (Suleymanova et al., 2018) | CC BY 4.0 | 1 | 1,054 |
| nidr0020 | VISEM-Tracking (Zenodo) (Thambawita et al., 2023) | CC BY 4.0 | 3 | 13,608 |
| nidr0021 | WBC1 (Mendeley Data) (Zheng et al., 2018) | CC BY-NC 3.0 | 3 | 300 |
| nidr0022 | WBC2 (Mendeley Data) (Zheng et al., 2018) | CC BY-NC 3.0 | 3 | 100 |
| nidr0023 | S-BSST265 (BioImage Archive) (Kromp et al., 2020) | CC0 1.0 | 1 | 79 |
| nidr0024 | CEM500K (EMPIAR) (Conrad & Narayan, 2021) | CC0 1.0 | 1 | 264,187 |
| nidr0025 | S-EPMC8322260 (BioStudies) (Khoshkenar et al., 2021) | CC BY 4.0 | 2 | 81 |
| nidr0027 | Microscopic peripheral blood (Mendeley Data) (Acevedo et al., 2020) | CC BY 4.0 | 3 | 15,710 |
| nidr0028 | Three fold annotated potato dataset (Figshare) (Biswas & Barma, 2020) | CC BY 4.0 | 3 | 14,269 |
| nidr0029 | RxRx19a (Recursion) (Heiser et al., 2020) | CC BY 4.0 | 5 | 60,594 |
| nidr0030 | RxRx19b (Recursion) (Cuccarese et al., 2020) | CC BY 4.0 | 6 | 34,787 |
| nidr0031 | RxRx1 (Recursion) (Sypetkowski et al., 2023) | Recursion Custom Commercial License | 6 | 26,328 |
| nidr0032 | RxRx2 (Recursion) (Cuccarese et al., 2020) | CC BY-NC-SA 4.0 | 6 | 30,744 |
| wtc0001 | WTC-11 (Allen Institute) (Viana et al., 2023) | Allen Institute Terms of Use | 4 | 117,882 |
| | | | **Total** | **2,792,462** |

Table A5: Testing Image Dataset Information

| ID | Dataset | License | Channels | Images |
|---|---|---|---|---|
| CHAMMI-WTC | WTC Chen et al. (2023) | CC BY-NC 4.0 | 3 | 65,103 |
| CHAMMI-HPA | HPA Chen et al. (2023) | CC BY-NC 4.0 | 4 | 66,936 |
| CHAMMI-CP | CP Chen et al. (2023) | CC BY-NC 4.0 | 5 | 88,245 |
| CellPHIE | CellPHIE (Broad Institute) (Kang et al., 2025) | CC BY-NC 4.0 | 14 | 57,021 |
| idr0017 | IDR0017 (IDR) (Breinig et al., 2015) | CC BY 4.0 | 2 | 122,880 |
| HPAv23@256x256 | HPAv23 Chen et al. (2023) | CC BY-SA 4.0 | 4 | 1,138,378 |
| Jump-CP | Jump-CP Chandrasekaran et al. (2023a) | CC0 1.0 | 5 | 813,040 |
| RBC-MC | RBC-MC Doan et al. (2020) | CC BY-NC 4.0 | 1 | 123,272 |
| | | | **Total** | **2,474,875** |

## A.2 DATA PREPARATION

Before sampling and data curation, all the data from the original 75 studies was down-loaded to our local servers in their original TIFF/OME 16-bit formats. We used a high-throughput computing cluster (Center for High Throughput Computing, 2006), Condor workflows (Livny et al., 1997), and GNU Parallel (Tange, 2024) for pre-processing and standardization of these datasets, bringing them down to ∼50TB of compressed images. First, we decoded the original image format (e.g., TIFF, flex, etc) and separated individual channels, individual z-planes, and individual temporal frames into separate files. Next, we standardized pixel depth from the original (e.g., 12 or 16 bits) to 8 bits after rescaling illumination values. Images intensities were normalized between 0 and 255 after trimming the tail end distributions of the initial histogram at the 0.1% and 99.9% percentiles. Finally, each single file was independently stored in PNG format with lossless compression, resulting in ∼42M individual channel files. Note that no spatial rescaling or cropping was used to reduce storage size — images in our dataset preserve their original resolution.

## A.3 METADATA PARSING

The resulting metadata file is useful to understand the distribution of images and to sample a representative subset. Figure 4 illustrates the diversity of images after selection and sampling

according to a few relevant annotations, which could be leveraged for learning. Importantly, while there are many types of images represented, the specific subgroups in the training set are sparse and have long-tail distributions (Fig. 4b). This is a known challenge of real world data and an opportunity to investigate robust learning strategies despite the natural biases. Also, note that the original sources may not have all the information of the 22 columns that we tried to parse, due to lack of standards for imaging metadata (Schmied et al., 2024). Therefore, the final metadata file may have missing information for some studies.

**Techniques Used to Reduce the Number of Reagents in the Metadata**  After merging all metadata sources the resulting file contained 145k reagents, which were reduced to 92k using multiple alignment techniques. We performed metadata harmonization to address heterogeneous identifiers and naming conventions in the candidate list. Using a programmatic gene-ID mapping step (mygene) we mapped ENSEMBL identifiers to HGNC symbols and removed records that were duplicates by virtue of having both identifiers; this eliminated 16,029 ENSEMBL-only redundancies and reduced the working set to 114,748 reagents. We then applied a sequence of regex- and rule-based sanitization passes: we discarded tokens of two characters or fewer, removed purely numeric tokens of three characters or fewer, and normalized hyphenation to collapse format mismatches (for example, MK2206 versus MK-2206) while explicitly preserving legitimate hyphens that encode stereochemistry, numbered loci distinctions (gene1-1 versus gene11), and organism-specific conventions (for example, fly/worm par-1 versus human PAR1). We also stripped timestamp-like experimental annotations appended to gene symbols (for example, HU-90-min-ILK6) and, when a canonical symbol existed elsewhere in the list, removed the annotated variant entirely. We also used some databases such as Flybase, and Pombase datasets to rename Drosophila and Yeast related genes.

**Ground Truth Extraction for IDR Studies**  We developed a reproducible, metadata-paper pipeline to adjudicate "hit" versus "no hit" labels for candidate metadata reagents using the content present in each paper's reconstructed text, tables, and machine-generated figure descriptions. A dataset-specific prompt encodes the inclusion criteria for what constitutes a hit in that study.

The pipeline begins by loading the reconstructed text for a selected study. If image links are present in the text, we apply a two-stage figure augmentation pass using OpenAI's o1 model (Jaech et al., 2024). First, we download each image and produce a transcription that captures the image, visible labels and legend text. Second, we use the figure transcription to summarize roles, interactions, conditions, and quantitative readouts described in the figure. For each image, the URL in the text is replaced with the generated description so that subsequent steps operate on a text representation that incorporates figure content.

Candidate entity construction is metadata-driven. We read the study's reagent or gene list line-by-line and derive name variants by lowercasing, extracting parenthetical synonyms, and stripping common suffix words to produce base names. We compile a word-bounded regular-expression union of all variants, exclude very short tokens and a small stoplist to reduce spurious matches, and scan the augmented text to recover a de-duplicated set of candidates that are explicitly mentioned in the paper. Because candidates originate from the metadata list and are matched with word boundaries, we limit substring bleed (for example, avoiding matches of "ER" within "ERK").

Hit adjudication is performed by invoking paper specific "hit" definitions for each candidate against the full augmented text under a text-only evidence policy. The adjudicator, OpenAI's o1 model, must both assert whether the candidate meets the inclusion criteria and return a concise rationale together with a short, verbatim in-text snippet that substantiates the decision; labels without a snippet do not qualify as positive.

Inclusion criteria are encoded per dataset to reflect the study's stated endpoints. For IDR0017 specifically, a hit is a reagent that exhibits interaction activity with any screened cell line. In all cases, external knowledge is disallowed and every positive must be justified by an in-paper snippet. Outputs consist of a per-reagent record written to a study-specific text file that captures the final label, a brief rationale, and a snippet of at most two sen-

tences that provides the evidentiary anchor. We additionally persist the full processing log, the augmented text file where applicable, and a JSON file with per-figure transcriptions and relationship summaries. Each record includes provenance fields such as the paper identifier, the exact matched string for the entity, model name and version, timestamps, and prompt versions to facilitate audits and downstream benchmarking.

## A.4 DATASET CURATION

**3D sampling.** Seven of the 75 studies involve imaging data acquired in three dimensions, i.e., z-planes recorded at various depths. Given that our focus is 2D imaging, we sampled 20% of the z-planes from the central region of the 3D stack. This follows two observations: first, images in the extremes of the stack appear to be empty or out of focus in most cases. Second, z-planes from the same 3D stack are highly correlated and may not bring new information from the 2D perspective. After applying this filtering, we reduced the number of multi-channel images from 26.7M to 24.2M. A6 reports the sampling information for all the 3D studies.

Table A6: 3D Sampling Information for Multi-Channel Images

| Dataset Name | Multi-Channel Images | | |
|---|---|---|---|
| | **After** | **Before** | **Ratio Preserved** |
| idr0001 | 741,960 | 2,374,272 | 0.3125 |
| idr0011 | 39,890 | 167,551 | 0.2381 |
| idr0086 | 16,705 | 76,851 | 0.2174 |
| idr0089 | 8,857 | 40,565 | 0.2183 |
| idr0115 | 138,035 | 579,747 | 0.2381 |
| idr0120 | 75,828 | 192,908 | 0.3931 |
| idr0123 | 3,394 | 13,342 | 0.2544 |
| wtc0001 | 39,294 | 117,882 | 0.3333 |

**Temporal sampling.** Nine of the 75 studies involve time-lapse imaging, i.e., images acquired through time. In live microscopy, the camera does not move and the cells do not move too quickly, resulting in highly correlated 2D frames. Following the same intuitions from 3D sampling, we sampled a fraction of the frames by defining evenly distributed time points in a given sequence. With this, we reduced the number of multi-channel images from 24.2M to 6.8M. A7 reports the sampling information for all the temporal studies.

Table A7: Temporal Sampling of Studies for Multi-Channel Images

| Dataset Name | Multi-Channel Images | | |
|---|---|---|---|
| | **After** | **Before** | **Percent Preserved** |
| idr0002 | 43,392 | 386,884 | 0.1122 |
| idr0011 | 39,890 | 39,890 | 1 |
| idr0013 | 959,795 | 17,848,117 | 0.0538 |
| idr0115 | 14,815 | 138,035 | 0.1073 |
| nidr0003 | 16,699 | 33,403 | 0.4999 |
| nidr0020 | 14,685 | 29,370 | 0.5 |

**Control-based sampling.** The majority of studies in the collection involve experiments with control conditions used to compare against treatments. Control samples are usually abundant and replicated multiple times to improve the ability to make meaningful comparisons. To reduce redundancy, we sampled a fraction of the control images to balance its representation with respect to treatment samples. This reduced the number of multi-channel images from 6.8M to 5.9M.

**Intensity filtering.** The pixel intensity statistics of microscopy images tend to have a low mean and a long tail distribution. We filtered channels whose mean pixel intensity is too dark with respect to the distribution of all channels in the same study, under the assumption that these images are mostly empty FoV or have too small or too few interesting objects.

**K-means based sampling.** Images in the same study may be too similar to each other because the morphological differences of certain perturbations is too subtle or because some treatments do not have a strong effect. In order to sample a few representative images from each study we use k-means clustering based on the histogram of intensities. First, we compute the gray scale histogram for each channel with 256 bins of intensity values. Next, we load the histograms of all images per plate and use kernel k-means with the histogram intersection kernel to group the data in K groups, where K is determined by a sampling factor (percentage of desired images). For sampling we keep only one random sample from each cluster. This approach ensures diversity in the intensity values of the images, which is a proxy indicator of image content. The sampling factor $\alpha$ that determine the number of clusters depends on the total amount of images in one study: $\alpha = 1$ for studies with less than 100K multi-channel images, $\alpha = 0.5$ for studies with images between 100K and 500K images, and $\alpha = 0.2$ for studies with more than 500K images.

## A.5  CELL SEGMENTATION AND CELLULAR SCALES

We used the Cyto3 model in Cellpose (Pachitariu & Stringer, 2022) to segment all the different 75 studies in our dataset to obtain centroid coordinates of all these different microscopy images at the single-cell level. Our goal was to segment the nucleus of the cells as that was the most consistent channel type present in all the studies. We manually configured Cellpose parameters such as size, and channels used for segmentation to improve segmentation quality. We have found 300 million cells in the small version of our dataset and almost 1.8 billion cells (1,791,151,533 cells) in the large version of our dataset. Figure A9 provides a quantitative background about the segmentation results.

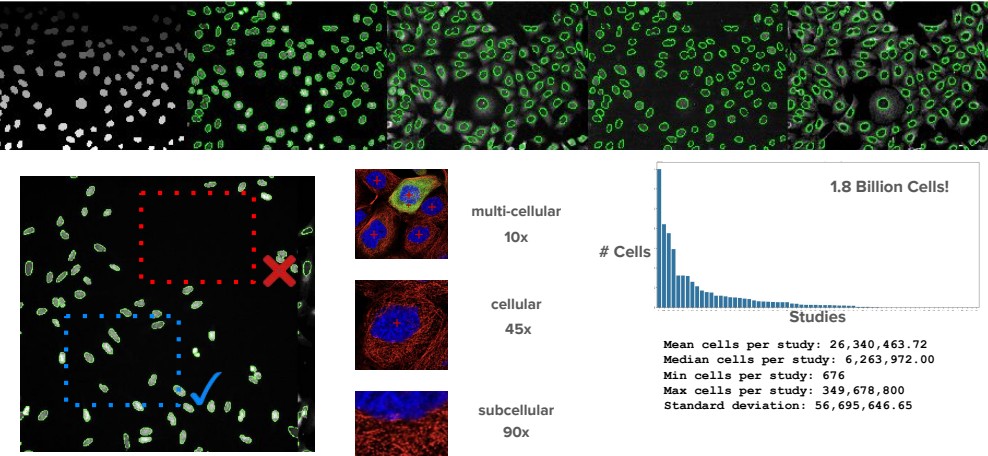

Figure A9: Segmentation pipeline examples.

It takes 7 days on 8 NVIDIA L40s GPUs to segment all multi-channel images in `CHAMMI-75`. Once all images were segmented, we found that the area taken by the segmented nucleus on microscopy images is 19.12%, which means that 80.88% of the nucleus image is empty. To counteract this issue, we used our single cell coordinates to do guided crops during our model training. We also annotated average cell sizes in all 75 studies, which helped us observe the cellular structures at different scale levels similar to how magnification levels are discrete in microscopy.

**Cell scales.** We manually determined crop sizes that contain subcellular, cellular, or multi-cellular views and created study-level annotations for the 75 sources. The resolution of some studies may only allow access to one or two of the three scales. These annotations are helpful to crop and resize images in a predictable way.

We have developed cellular scale annotations at the multi-cellular scale and the cellular scale. These annotations were obtained manually for all the different cellular scale levels. We set the cell centroid for an image in the middle and crop the scale accordingly.

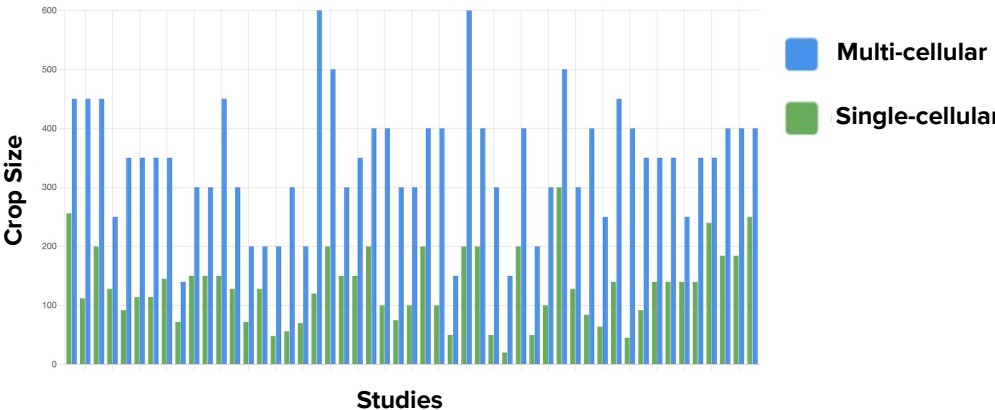

Figure A10: Histogram of cellular-level annotations

**Data loader.** In combination, the single-cell coordinates and the scale annotations are useful information to design data loaders that generate fixed-size crops with the desired properties. We implemented a data loader that samples image content hierarchically in the following way: 1) randomly select a multi-channel image, 2) randomly select a single cell from the coordinates table, 3) randomly select an available scale size, 4) crop a region around the cell location with the selected scale size and then resize to a standard model input size, 5) finally, apply any additional transformations or augmentations. This design can be modified or expanded to explore other multi-scale learning approaches.

**Content annotations.** The heterogeneous images in `CHAMMI-75` have different sizes, magnifications, and resolutions, which departs from the standard practice of assuming a fixed configuration for all images. This means that a random crop from an image in `CHAMMI-75` may contain multiple cells, a single cell, or a subcellular structure. This lack of alignment in scales can be a challenge for representation learning algorithms. We created study-level, scale annotations for the 75 sources to determine the views available in an image (subcellular, cellular, multi-cellular). We implemented a content-aware, hierarchical data loader for training that leverages cell coordinates and scale annotations.

A.6 Evaluation Datasets and Benchmarks

**IDR-0017.** This benchmark aims to reproduce the findings of the IDR017 study: A Chemical-Genetic Interaction Map of Small Molecules (Breinig et al., 2015). The study evaluates gene-compound interactions by treating 1,280 compounds on 12 isogenic cell-lines using high-throughput imaging techniques. It extracts image-based features at the single-cell level to quantify phenotypic changes under a combinatorial study that aims to understand drug mechanisms and target effects.

The biological experiment behind the images aims to determine whether a cell-line (with a mutated gene) responds to specific chemical treatments. A control experiment consists of a cell-line without chemical treatment, and then other treatments are applied to evaluate if there is any difference between using the treatments or not. When a difference between the control experiment and the effect of a treatment is large enough, the chemical treatment is labeled as a hit. In this study, there are a total of $1,280 \times 12 = 15,360$ combinatorial experiments which identified 193 hits. To identify hits, the original study used 20 manually engineered features at the single-cell level, including cell size, actin intensity, nucleus intensity, cell shape and nuclear shape.

The dataset consists of approximately 122,880 two-channel images across 96 384-multi-well plates. The two channels are: a nucleus marker and a cytoskeleton marker, and 4 fields of view are acquired per well at 10X magnification in 2048x2048 pixel images. Each plate contains controls followed by treated cells, and the study is performed across 2 replicates on 2 different set of plates. The original dataset was obtained from IDR (Williams et al., 2017),

and we pre-processed the images to facilitate standardized analysis as follows: 1) Performed cell segmentation using cellpose model using nucleus and cytoskeleton channel. 2) sampled 512 x 512 patch from every image based on cell density using segmentation maps to reduce the number of cells to analyze. This method help us to reduce the average cell count from 2300 cells to 200 cells per image. 3) Patches of 100 x 100 are sampled across single cells for feature extraction.

To transform the original analysis into an evaluation benchmark, we start computing features using trained models to approximate the representation of the phenotypic changes at the single-cell level. To identify hits, we use the model embeddings from images of the same treatment to calculate the effect size with respect to control cells. The single-cell embeddings are aggregated by taking the mean per image and two Eucledian distance matrices are constructed: one for the treatment vs control images, and another one for the control vs control images. The effect size of a compound is estimated using the Wasserstein distance between the distance matrices of control and treatment image embeddings. To avoid batch effects, we aggregate both replicates for each compound using PCA whitening normalization. The compounds for each cell line are ranked on the basis of the effect score. We use Recall@50, Recall@100, and AUROC as metrics to measure how well a model ranks high-interaction compounds above low-interaction compounds.

**HPAv23 at 256x256.** The HPAv23 subcellular benchmark was released with the SubCell model(Gupta et al., 2024). The dataset was originally curated in the 23rd version of the Human Protein Atlas Project (Thul et al., 2017). The authors of (Gupta et al., 2024) made crops of the images to allow machine ingestion, and develop a benchmark for the same. This benchmark provides a granular detail needed to test whether current models can identify meaningful subcellular differences in cellular organization.

The dataset is a collection of immunofluorescence images encoding the expression and spatiotemporal distribution of 13,141 genes in 37 cell lines. The images were stained with DAPI, a fluorescent dye that labels the nucleus, and antibodies labeling endoplasmic reticulum (ER), microtubules (MT) and protein of interest (protein). The images were cropped from FOV images to single cell images that contained 1,138,378 cells. The data set was split by the original authors based on antibodies with a ratio of 7: 1: 2 into training, validation, and test sets, respectively. (Gupta et al., 2024) used a multilabel stratification strategy to ensure a similar multilabel distribution between the sets.

The protein localization task we have adapted paper is a supervised learning task using a multi-layer perceptron (MLP) on the features obtained from a frozen backbone of all the models. The MLP classifier uses the same three-layer classifier architecture as (Kraus et al., 2024) and focal loss (Cho et al., 2022) to address class imbalance in the dataset. The classifiers were trained on features extracted from the whole HPAv23 set including the pre-training set, and the rest of the images. Unlike the original authors, we downsampled all images from $1024 \times 1024$ to $256 \times 256$ to accelerate the benchmark, reduce its time complexity, and decrease storage requirements. We are re-releasing this reformatted set with our dataset to enable easy usage of this benchmark. The task has two sets of classifier: the first set us comprised of 19 categories specified in Kaggle challenge, and the second set has a broader range of 31 categories. We have reported the micro and macro average precision (AP) as the classification metrics and used multilabel ranking average error and coaverage error to evaluate multilabel performance on the test sets just like the original authors. The challenge category results are reported in the main text in 1, 2 and F16. The results of the unique category are reported in F17

**JUMP-CP1 Compounds.** In this benchmark, the quality of the replicate level and consensus treatment profiles is evaluated in a subset of the JUMP-CP1 dataset (Chandrasekaran et al., 2023b) for the compounds that were originally curated with Broad Institute's Drug Repurposing Hub. Cell Painting assay (Bray et al., 2016) was used, where six fluorescent dyes highlight eight cellular compartments that are imaged in five channels at 20X magnification.

From a biological perspective, the aim is to capture meaningful differences between populations of cells with respect to the perturbation and the target of this perturbation. Those

differences between cell states might be subtle and hard to detect for certain types of perturbations even against negative control (unperturbed) cells. The quality of the replicate level profiles is defined by the closeness of a given perturbation replicates against a set of negative control replicates. The quality of treatment-level profiles is evaluated by the closeness of profiles with the same biological label (in this case, gene target) against other perturbations. Both metrics are introduced and implemented in *copairs* benchmarking suite (Kalinin et al., 2025).

Originally, JUMP-CP1 includes many biological and experimental conditions: two cell lines (A549 and U2OS), three perturbation types (chemical compounds, gene open reading frame (ORF) overexpression and CRISPR-Cas9 knockouts) captured at two time points. For this benchmark, we used only the data from the U2OS cell line and chemical compound perturbations at both time points, eventually having a similar set of seven 384-well plates that matches the one used in the evaluation of SubCell (Gupta et al., 2024). The images from the evaluation plates are not used in the pre-training set. Original 16-bit TIFF images (24,192 fields of view – 120,960 single-channel images in total) were normalized and compressed to 8-bit PNG images with DeepProfiler (Moshkov et al., 2024) *prepare* option and then single-cell crops were exported with *export-sc* option, using the cell locations provided with the original CellProfiler features, resulting in 2M unmasked cell-centered crops ("cells in context").

In this benchmark, we start by computing features for single-cell crops that are saved at size 160x160 and that are further cropped to size 128x128. The final 128x128 input images are resized to accommodate to the expected input size of the particular model at use. Single-cell feature vectors are then aggregated using mean aggregation to the well-level (replicate) profiles. Well-level profiles are then batch corrected with ZCA-whitening from pyCytominer package (Serrano et al., 2025) relative to negative controls with $epsilon = 0.001$. This data is then used in the benchmark: *copairs* returns mAP and p-value for each perturbation (phenotypic activity) and target (phenotypic consistency). We report the mean mAP for phenotypic activity as *JUMP-CP1*. For treatments that did not pass the 0.05 p-value threshold, we assume $mAP = 0$. We report mean mAP for phenotypic consistency as *JUMP-CP2*. Similarly, for targets that did not pass 0.05 p-value threshold, we assign $mAP = 0$. As phenotypic consistency is calculated only for treatments that passed p-value threshold for phenotypic activity, some targets might not be represented in this step of benchmark. We also assign $mAP = 0$ to such missing targets.

**CellPHIE.** This study used iPSC-derived neurons to investigate genetic and morphological markers of Huntington's Disease using the latest generation of pooled genetic perturbations at the single-cell level with a multiplexed optical screen. The dataset contains a total of 57,021 images out of which 45,782 training images and 11,239 testing images. The original images from the dataset were filtered, and we kept images from DS28 time sample, and we removed two genes from the original set which were DGKE and GAS7 genes. Single-cell images segmented, cropped and masked at 64x64 pixels. The imaging panel consists of 5 Cell Painting channels and 9 protein markers obtained with immunofluorescence, for a total of 14 imaging channels. The 14 channels in the images are in the following order: DNA, NeuN, pRPS6, RANGAP1, NFKB, TOM20, LAMP1, TDP43, G3BP1, GM130, Golgin97, SYTO, ER, AGP. Channels 2 to 10 are protein channels.

Each single cell was perturbed with one of 19 genes, which was identified with optical barcoding. From the 19 perturbations, one is a non-targetting control used as a reference to determine the effect of other preturbations. The task in this benchmark is to classify single cells in a binary classification setting: non-targeting vs perturbed gene. The dataset has a training / validation split to evaluate performance using a linear probe. Note that none of the images in CellPHIE were included in the `CHAMMI-75` training set, representing a fully held-out set with novel channel combinations and the largest number of channels in the benchmark.

**RBC-MC.** This benchmark is based on bright-field microscopy imaging of red blood cells (RBCs) from Doan et al. (Doan et al., 2020) containing one channel. The dataset comprises RBC samples collected from two geographically distinct blood banks located in Switzerland and Canada. Following the experimental protocol, we train a logistic regression classifier on

one dataset and evaluate its performance on the other to assess cross-dataset generalization. The benchmark consists of 76,577 images from the Swiss dataset and 46,695 images from the Canadian dataset, with all images uniformly sized at 48×48 pixels. RBC morphology is categorized into seven distinct classes: Smooth Disc, Crenated Disc, Crenated Discoid, Crenated Spheroid, Crenated Sphere, Smooth Sphere, and Side View. Images originally labeled as "Undecided" by expert annotators, indicating uncertainty in morphological classification, were excluded from the benchmark to ensure label quality.

## A.7 Metadata Columns and Descriptions

Our metadata fields are divided into six different groups, where each group corresponds to a particular type of experiment. The metadata comes in six major groups: experiment, biology, imaging, microscopy, geometry, and storage information. Each record in the metadata file points to a single channel file. The metadata is designed to facilitate grouping of channel files according to the categories described before. For each category, we have several metadata columns described below. If the information for an image is missing or not known, the corresponding value will be labeled with the string "unknown". We try not to leave NaN or empty strings in the metadata file. If you see something, say something. A11 contains visualization of the six groups of metadata and the 22 fields present in the metadata.

We are going to provide a detailed list with descriptions for all the different columns present in the metadata:

1. **experiment.study**: Identifier of the study.
2. **experiment.plate**: Plate where the image was acquired. If images come from another format (not plate-based), this identifier can indicate a major group of experimental arrangements in the study.
3. **experiment.well**: Well position within the plate. The format of letter and number is preferred, but this is flexible.
4. **experiment.reagent**: Identifier or name of the treatment or reagent used to treat the cells. In many cases, this is a gene name, a compound name, or a protein name, while in other cases it may reflect other experimental interventions (e.g., temperature).
5. **experiment.control**: Whether the image comes from a control well or not, and what type of control they may be, for example, positive or negative control. If not a control, use the string "no".
6. **biology.organism**: Name of the organism where the cells come from. For example, humans, mice, plants, etc.
7. **biology.cell_line**: Name of the cell line. Many cell lines have well-known names (such as HeLa), other cell lines are from primary patients and have anonymized codes, and others are from genetically modified organisms.
8. **biology.cell_type**: The functional type of cell, regardless of the cell line. Examples include neurons, red blood cells, cancer cells, pancreatic cells, etc.
9. **imaging.multi_channel_id**: This is the field that ties together multiple channels. It is a consecutive number from the original database concatenated with the study number. A unique `multi_channel_id` connects the channels of an image.
10. **imaging.panel**: Names and dyes of the channels used to create the image. This gives context for where the observed channel file comes from. Example: "DNA, protein, cytoplasm".
11. **imaging.channel**: Numeric value of the channel according to the panel. This value is one-based.
12. **imaging.channel_type**: Biological compartment of the cell that is visible in the channel. This is a list of standardized values that include: nucleus, cell body, bright-field, etc.
13. **microscopy.type**: Name of the type of microscopy used for acquisition of the channel file. Examples include: fluorescence, bright-field, confocal, cryoEM, etc.

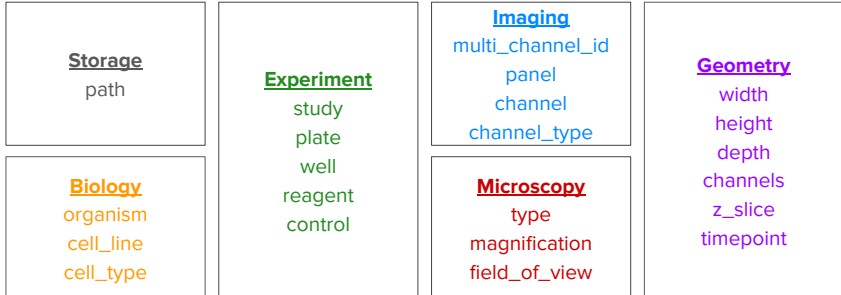

Figure A11: Visualization of metadata fields in six groups and 22 fields.

14. **microscopy.magnification**: Numeric value of the magnification used to acquire the image.

15. **microscopy.fov**: Field of view, well site, or microscope position in the well when the channel was captured.

16. **geometry.width**: Channel width in pixels.

17. **geometry.height**: Channel height in pixels.

18. **geometry.depth**: Total number of z-planes this channel belongs to, if the study is a 3D imaging assay.

19. **geometry.channels**: Total number of sibling channels in the same image.

20. **geometry.z__slice**: Number of the z-plane for this channel. It is a numerical value.

21. **geometry.timepoint**: Number of the frame in the timelapse sequence, if applicable.

22. **storage.path**: File path of the PNG file in the dataset containing this channel

## A.8 METADATA QUALITY AND HANDLING INCONSISTENCIES

The construction of `CHAMMI-75` involved integrating 75 heterogeneous studies, many of which lacked standardized or complete metadata, reflecting a general deficiency in data sharing practices across biology. Our approach to handling missing, inconsistent, and ambiguous data was rigorous yet pragmatic, acknowledging that perfect retrospective standardization is often impossible.

**Handling missing data.** The vast majority of studies provided partial metadata. For fields where information was unequivocally absent (e.g., biology.cell_type or microscopy.fov), the entry was explicitly and consistently annotated with the string "unknown". This approach avoids using null values or empty strings, ensuring consistency for subsequent programmatic access and filtering.

**Resolving ambiguity and inconsistency.** In cases where metadata was ambiguous (e.g., multiple spellings for a cell line, or an overly verbose stain identity), we employed a multi-step resolution pipeline: (1) Cross-reference: values were validated against other metadata columns (e.g., cross-checking organism and cell line). (2) External validation: we performed targeted web searches and reviewed the associated scientific publications to infer the most likely accurate value. (3) LLM-assisted parsing: As noted in Appendix A.3, we used LLMs to systematically extract and organize certain information, providing a rapid, initial pass at resolving naming variants and extracting structured data from free-form text descriptions. (4) Manual curation and normalization: to minimize inconsistencies, all resulting values were manually reviewed, low-cased, and mapped to a simplified vocabulary or ontology where possible (e.g., different fluorescence channels like 'DAPI', 'Hoechst 33342', and 'Nuclear stain' were often mapped under the canonical channel_type of 'nucleus').

**Last resource.** If, after all these steps, the interpretation remained uncertain (e.g., a stain was ambiguously described and the original publication offered no clarity), we defaulted to preserving the original value or, more conservatively, labeling the entry as "unknown".

The resulting metadata table, while significantly cleaned, remains inherently noisy. We view this noise and high sparsity (Figure 4b) not as a limitation, but as a defining feature of the resource. Unlike highly standardized datasets produced synchronously by a single consortium, `CHAMMI-75` attempted to align data that were never intended for cross-study integration. This noisy complexity presents the real-world challenge foundation models for bioimaging must solve –to learn representations that are robust despite inconsistent. Future efforts should focus on promoting standardized metadata acquisition for new studies (Schmied et al., 2024), as retrospectively fixing metadata is computationally infeasible and prone to subjective error. Furthermore, our dataset provides a compelling target for future research into semi-automated, multi-modal systems capable of resolving these remaining metadata inconsistencies.

### A.9 Comparison to other datasets

These datasets were chosen for comparision as our dataset is not a biological study but rather an AI-ready dataset to investigate single-cellular morphology foundation models. Our data set is not a reference set for interactive querying because it contains diverse samples from heterogeneous studies in a way optimized for machine learning, not for biological discoveries. We have compared our data set against similar imaging data sets that are ready for machine ingestion and have been used in relevant work to build foundation models in cellular morphology. All the numbers used in Figure 2 are reported in the table A8.

Table A8: Overview of Multi-Channel Image Datasets.

| Name | Images | Src | Ch. | Org. | Access | Cite |
|---|---|---|---|---|---|---|
| CHAMMI-75 | 2,849,483 | 75 | 25 | Multi-channel | Public | **Ours** |
| CHAMMI | 220,284 | 3 | 8 | Multi-channel | Public | (Chen et al., 2023) |
| RxRx | 4,168,973 | 6 | 6 | Fixed-channel | Public | (Recursion, 2025) |
| Jump-CP | 8,109,884 | 12 | 5 | Fixed-channel | Public | (Chandrasekaran et al., 2023a) |
| HPAv23 | 1,138,378 | 1 | 4 | Fixed-channel | Public | (Gupta et al., 2024) |
| IDRCell100k | 104,093 | 79 | 10 | Multi-channel | Public | (Bourriez et al., 2024) |
| Phenoprints-16M | 16,000,000 | 1 | 6 | Fixed-channel | Private | (Kenyon-Dean et al., 2024) |
| CytoImageNet | 890,737 | 40 | 12 | Mixed-channel | Public | (Hua et al., 2021) |
| Microsnoop | 2,230,000 | 7 | 3 | Mixed-channel | Public | (Xun et al., 2024) |

Src refers to Source. Org. refers to the type of organization of the dataset and whether it was multi-channel, fixed-channel or mixed-channel. Multi-channel means that it has images with images having varied number of channelled images, Fixed-channel means that the images only have one configuration of channels. Mixed channel mean if they originally had images with varied number of channels but the channel order and information was not preserved properly.

## B Factors Affecting Representation Learning

### B.1 Motivation

CHAMMI-75 is highly heterogeneous (Figure 4). In this section, we aim to understand how different factors of variation drive generalization in representation learning. We perform five ablations using metadata entries as a signal to separate data with for training. The factors are the following:

- Specialization
- Modality
- Magnification
- Cell Line
- Similarity to HPA
- Number of Channels

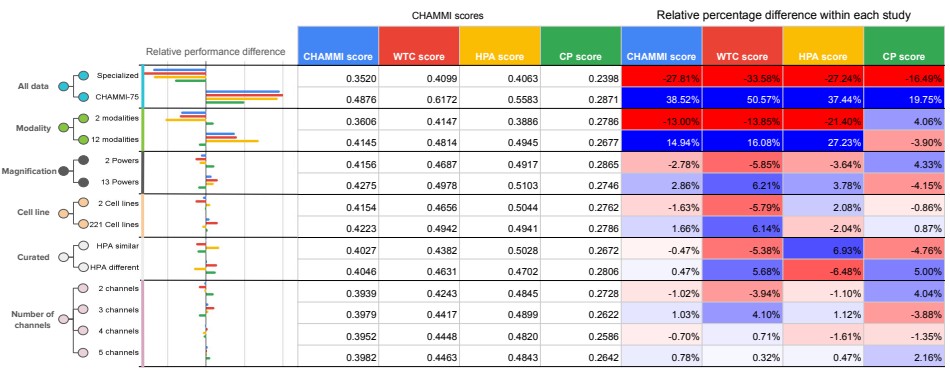

| | | Relative performance difference | CHAMMI scores | | | | Relative percentage difference within each study | | | |
| | | | CHAMMI score | WTC score | HPA score | CP score | CHAMMI score | WTC score | HPA score | CP score |
|---|---|---|---|---|---|---|---|---|---|---|
| All data | Specialized | | 0.3520 | 0.4099 | 0.4063 | 0.2398 | -27.81% | -33.58% | -27.24% | -16.49% |
| | CHAMMI-75 | | 0.4876 | 0.6172 | 0.5583 | 0.2871 | 38.52% | 50.57% | 37.44% | 19.75% |
| Modality | 2 modalities | | 0.3606 | 0.4147 | 0.3886 | 0.2786 | -13.00% | -13.85% | -21.40% | 4.06% |
| | 12 modalities | | 0.4145 | 0.4814 | 0.4945 | 0.2677 | 14.94% | 16.08% | 27.23% | -3.90% |
| Magnification | 2 Powers | | 0.4156 | 0.4687 | 0.4917 | 0.2865 | -2.78% | -5.85% | -3.64% | 4.33% |
| | 13 Powers | | 0.4275 | 0.4978 | 0.5103 | 0.2746 | 2.86% | 6.21% | 3.78% | -4.15% |
| Cell line | 2 Cell lines | | 0.4154 | 0.4656 | 0.5044 | 0.2762 | -1.63% | -5.79% | 2.08% | -0.86% |
| | 221 Cell lines | | 0.4223 | 0.4942 | 0.4941 | 0.2786 | 1.66% | 6.14% | -2.04% | 0.87% |
| Curated | HPA similar | | 0.4027 | 0.4382 | 0.5028 | 0.2672 | -0.47% | -5.38% | 6.93% | -4.76% |
| | HPA different | | 0.4046 | 0.4631 | 0.4702 | 0.2806 | 0.47% | 5.68% | -6.48% | 5.00% |
| Number of channels | 2 channels | | 0.3939 | 0.4243 | 0.4845 | 0.2728 | -1.02% | -3.94% | -1.10% | 4.04% |
| | 3 channels | | 0.3979 | 0.4417 | 0.4899 | 0.2622 | 1.03% | 4.10% | 1.12% | -3.88% |
| | 4 channels | | 0.3952 | 0.4448 | 0.4820 | 0.2586 | -0.70% | 0.71% | -1.61% | -1.35% |
| | 5 channels | | 0.3982 | 0.4463 | 0.4843 | 0.2642 | 0.78% | 0.32% | 0.47% | 2.16% |

Figure B12: Expanded version of Figure 7 where we show CHAMMI scores numbers and relative percentage differences within each ablation

All these experiments have been conducted on the best performing model configuration we found in our scalability analysis (Figure 8): DINO BoC (Bag of Channels). In our ablation experiments, we compare the two most dominant metadata entries for each column removed against the remaining diverse set of entries left.

## B.2 SPECIALIZED MODELS ABLATION

We compared our CHAMMI-75 pre-trained BoC model against specialized self-supervised models trained on training sets (as reported in (Chen et al., 2023)) of all the three datasets: WTC, HPA, and Jump-CP.

According to Figure B12, we see a 38% increase when we use the diverse CHAMMI-75 to train a self-supervised learning model. This result inspired us to conduct the ablation study to better understand what factors enable this significant boost in performance. We have found variability in the amount of effect these different metadata parameters have.

## B.3 MODALITY ABLATION

We separated the data into two separate sets: the two most dominant modalities (fluorescence and epifluorescence modalities) versus all of the rest. The pre-trained model on fluorescence imaging consists of 278,735 multi-channel images (880,000 single channel images), whereas the pre-trained model of other modalities consists of 281,823 multi-channel images (790,000 single channel images). We see the distribution of multi-channel images in Figure B13 where the other 12 modalities have the majority of studies with 4 channel types.

According to our findings in Figure 8, one would expect the fluorescence imaging model to perform better than the other modalities model due to having more number of images but our findings contradict this expectation. We see that the other modalities model performs 13% better than the two modalities model, demonstrating that modalities play a huge role in affecting representation learning. We can infer from this result that having diverse microscopy types helps the model to learn better representations and hence is an integral feature of the dataset. These results align with previous work by (Arevalo et al., 2024), who identified microscopy type as the dominant confounding factor in batch correction studies. This convergent evidence suggests that exposure to diverse imaging modalities drives the learning of robust, generalizable representations that transfer effectively across different imaging contexts

## B.4 MAGNIFICATION ABLATION

We separated the data into two separate sets: the two most dominant magnifications (20 and 63) versus all of the rest. We pre-trained DINO bag of channels at a fixed compute budget on these two sets of data to understand how different sets of data affect representation

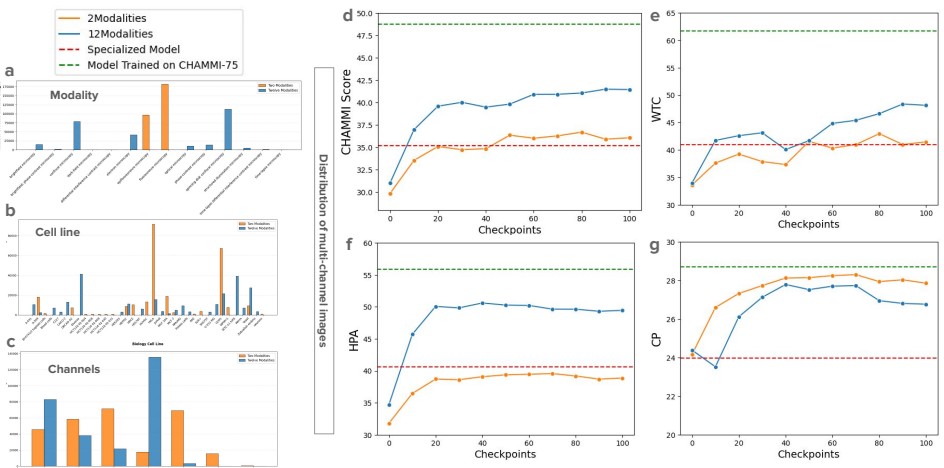

Figure B13: Showcasing Modality Ablations of having 2 modalities versus 13 modalities compared to the Specialized Models and model trained on `CHAMMI-75`

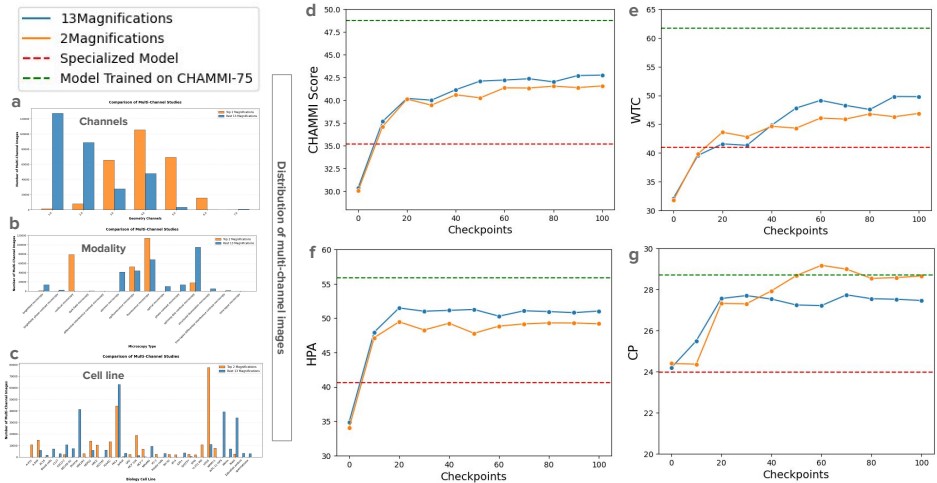

Figure B14: Showcasing Magnification Ablations of 20 and 63 compared to the Specialized Models and model trained on `CHAMMI-75`

learning. The distribution of channels (shown in Figure B14a) in the top 2 magnifications is skewed towards studies with ($\geq 3$) higher channel types whereas it is the latter in the 13 magnifications. In Figure B14b, 13 magnifications had different modalities, but top 2 magnifications were dominated by two modalities: fluorescence microscopy and confocal microscopy.

We compared the CHAMMI scores and observed in Figure B14 that we see marginally better performance in the case of diverse magnifications by 2.5%. These results indicate that diverse magnification slightly improves performance.

## B.5 CELL LINE ABLATION

We separated the data into two separate sets: the two most dominant cell lines (HeLA and U2OS) versus all of the rest. We pre-trained DINO bag of channels at a fixed compute budget on these two sets of data to understand how different sets of data affect representation learning. In Figure B15, we see what the distribution of multi-channel images looks like in terms of channels, modality, and cell lines. We can see that in terms of the modality distri-

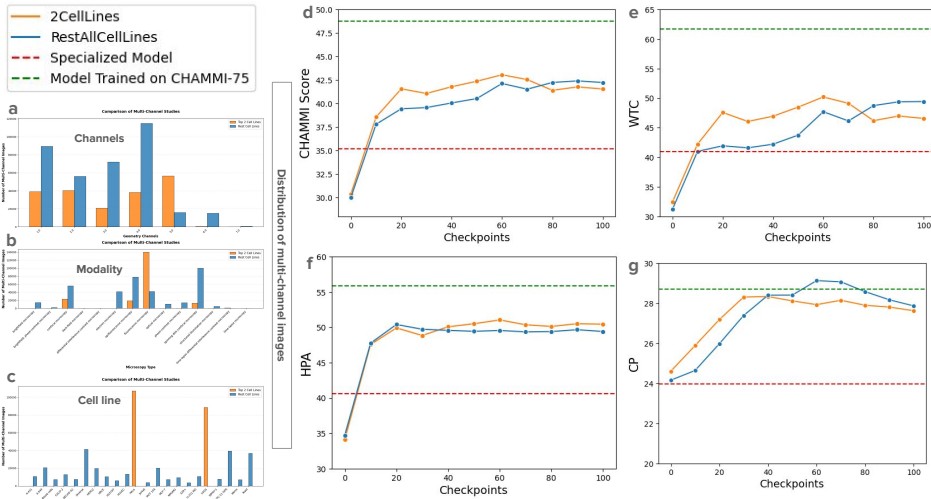

Figure B15: Showcasing Cell Line Ablations of U2OS and A-459 compared to the Specialized Models and model trained on `CHAMMI-75`

bution in Figure B15b, two cell lines consist mainly of fluorescence and confocal microscopy. The distribution of the number of channels in Figure B15a these studies seems to be fairly equally distributed as the rest of the cell lines.

We compared the CHAMMI scores and observed in Figure B15 that we see marginally better performance in the case of diverse cell lines by 1%. These results indicate that cell line variation does not affect performance.

### B.6 HPA Curated Ablation

We investigated how training data similarity to the target evaluation set affects representation learning. Our methodology involved extracting features from 1,000 single cells per study and channel type, totaling 75,000 cells across our dataset. We performed identical feature extraction on the CHAMMI-HPA test set using our best-performing DINO BoC model. We quantified similarity between different channel type studies against the test day. The average feature vectors were computed for each study and channel combination and then the top 10 most similar studies and bottom 10 most dissimilar studies were selected. We trained separate self-supervised DINO BoC models on each subset on a fixed compute budget.

In Figure B16a, we can see that the most similar studies arise from studies with 4 channels which makes sense as HPA also contains 4 channels. We see clear demarcations between different modalities and cell lines in B16b,c into two separate succint groups: one similar to HPA dataset, and one dissimilar to it. We compare the CHAMMI score results and see improvements in the HPA benchmark suggesting some benefit in training with similar data but overall the performance decreases in WTC, CP resulting in an overall net zero affect on CHAMMI score. This finding indicates that diverse, dissimilar training data promotes learning of more generalizable representations, even if it means sacrificing some degree of specialization to the target domain.

This result reinforces our broader finding that diversity in training data is a key driver of robust representation learning. Models trained on dissimilar data are exposed to a wider range of biological and imaging contexts, which appears to facilitate the learning of features that transfer more effectively across diverse microscopy tasks. While domain-specific fine-tuning on similar data may benefit specialized applications, diverse pre-training appears essential for building generally applicable microscopy foundation models.

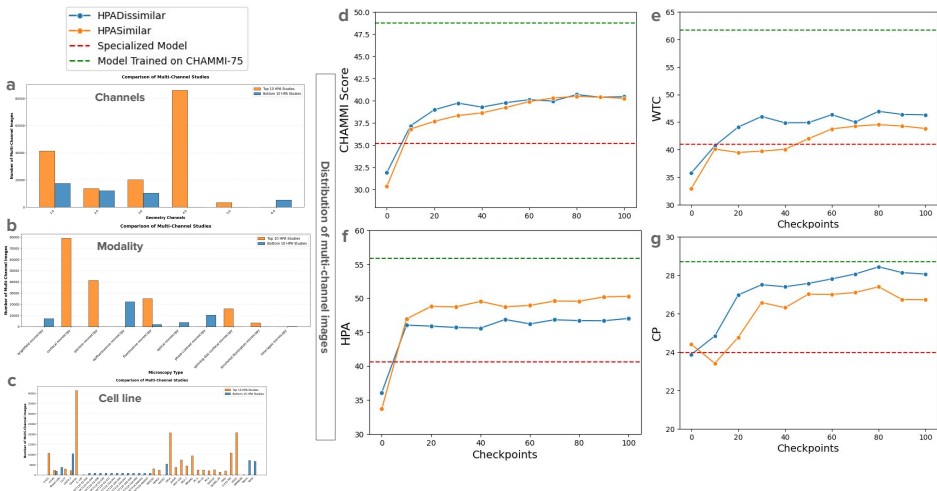

Figure B16: Showcasing Similar Data vs Dissimilar Data compared to the Specialized Models and model trained on `CHAMMI-75`

### B.7 Channel Number Ablation

We investigated whether the number of imaging channels present per study influences representation quality. We trained DINO BoC models on study subsets with varying channel counts (2, 3, 4, and 5 channels). Data volume and Channel count are two confounded variables as when you add studies with more channels, total data volume also increases. This results in seeing similar results as dataset scaling shown in Figure 8 if you don't control for this confounding variable.

To control for this confounding variable, we have a fixed compute budget for all models at 500,000 training iterations. Under these controlled conditions, we observed no substantial performance variations attributed to channel count (Figure B17). This result is expected given that the BoC architecture does not explicitly model inter-channel interactions, treating each channel independently.

Whether modeling inter-channel dependencies could yield improvements over BoC approaches remains an open research question. While several methods have explored this direction, including (Bao et al., 2023), (Bourriez et al., 2024), and (De Lorenci et al., 2025) approaches, none have yet demonstrated clear superiority over BoC methods. Our dataset, with its diverse channel configurations across studies, provides a valuable resource for future investigations into this question.

## C Scalability Analysis Details

To train multi-channel ViTs, prior work has explored supervised (e.g., Channel-ViT (Bao et al., 2023)), self-supervised (e.g., CA-MAE (Kraus et al., 2024), ChADa-ViT (Bourriez et al., 2024)), or hybrid multi-task objectives (e.g., ChA-MAEViT (Pham et al., 2025)). `CHAMMI-75` is a pre-training dataset primarily useful for self-supervised, potentially useful for weakly supervised, and not useful for fully supervised learning. We explore SSL algorithms to train a multi-channel model using `CHAMMI-75` with the goal of learning cellular morphology representations for solving diverse downstream tasks. Here, we evaluate well-established, representative algorithms from three major families of self-supervised learning methods for images: (1) SimCLR (contrastive) (Chen et al., 2020), (2) Masked Autoencoders (reconstructive) (He et al., 2022), and (3) DINO (self-distillation) (Caron et al., 2021).

To evaluate `CHAMMI-75` as a pre-training resource, we compare performance by training with varying configurations. First, we evaluate the performance of models trained with an increasing amount of multi-channel images, which we call "*dataset scaling*" evaluation.

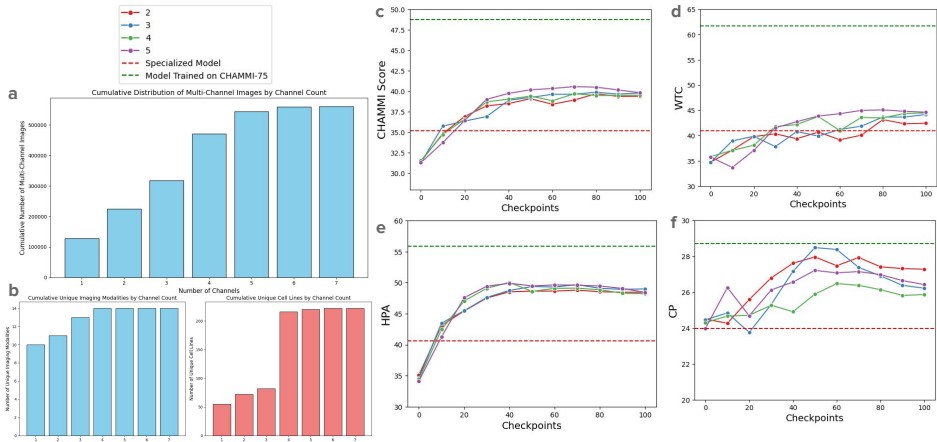

Figure B17: Showcasing Channel number variations with fixed compute of 500k compared to the Specialized Models and model trained on `CHAMMI-75`

Performance is evaluated on the CHAMMI benchmark (Chen et al., 2023), which has six out-of-distribution, phenotype matching tasks. We train models specialized on a fixed channel configuration as a baseline (1ds, 33K images per dataset; 3,4,&5 channels), and then combine sets to increase channel types and source variation, starting with the 3 CHAMMI subsets (3ds, 100K images, 8 channels), 10 datasets (10ds, 178K images, 14 channels), and a sample of the images in the 74 pre-training datasets from `CHAMMI-75` (74ds-small, 560K images, 25 channels). We leave the full `CHAMMI-75`-set (74ds-large, 2.8M images, 25 channels) out of the scaling evaluation; we only used it for the main benchmarking experiment following the best configuration in our analysis. We also conduct a "*model scaling*" evaluation, where we keep the dataset size fixed (74ds-small) and grow model parameters.

**Experimental and implementation details.** We train two types of ViT models: bag of channels (BoC) and multi-channel attention (MCA). The input size for all models is 224x224 pixels, and the number of channels is either one (BoC) or variable (MCA). Models are trained with one of the three selected SSL algorithms (SimCLR, MAE, DINO), and we keep hyper-parameters as constant across experiments as possible and apply relevant tweaks to avoid optimization divergence or collapse. After a model is trained with SSL, we keep the weights frozen and extract features in the three test sets of the CHAMMI benchmark without any finetuning. We use the PyTorch framework and we run experiments using multi-GPU training with 4 to 8 GPUs typically in a single node. All experiments were conducted in academic compute clusters (Appendix G.2).

**Results.** Consistent with previous studies, we find that models trained with SSL benefit from having access to more data (He et al., 2022; Oquab et al., 2023; Siméoni et al., 2025). We compare performance against the top-line of specialized models trained for each of the three subsets with fixed channels and with full supervision. This serves as an upper-bound reference to assess performance of models trained with SSL. The results are presented in Figure 8, and confirm that as we increase the dataset size and the model size, SSL models approach the performance of specialized, supervised models. This indicates that a single, multi-channel model trained at scale without supervision can be highly competitive in various downstream tasks.

We observe that the top factor that determines performance is the multi-channel strategy (BoC or MCA), followed by type of SSL algorithm, and model size. BoC models yield better performance than MCA models in the SSL regime, indicating that cross-channel correlations remain difficult to learn without supervision. We find BoC models easier to scale than MCA models because the latter has longer sequences that have high memory and compute requirements. MCA models required between 3X and 5X more GPU hours than BoC to complete a training session with the same amount of data (Figure G23). In addition,

SSL algorithms perform comparably within multi-channel strategies (BoC or MCA), with DINO consistently outperforming the others.

Based on these results, we trained a BoC ViT-small model with the DINO algorithm using the full `CHAMMI-75` dataset, which has 5X more data than the 74ds small set, and required 2,352 GPU hours to complete (7 days with 2x7-GPU servers). The result of this experiment followed the performance improvement trend obtaining a 9.8% relative improvement over the best result in the dataset scaling evaluation. The model scaling results suggest that additional performance gains could be obtained with larger ViT architectures, which we did not explore in this work.

# D    QUALITATIVE ANALYSIS OF FEATURE SPACE

To visualize the global structure of the feature space learned by models trained on `CHAMMI-75`, we performed a qualitative analysis using UMAP projection on single-channel features extracted from a balanced, single-cell sample of all 75 source studies.

## D.1    EXPERIMENTAL PROTOCOL

We sampled approximately 1,000 single-cell, multi-channel crops from imges in each of the 75 source studies, resulting in 196,660 individual single-channel images. We then used three representative pre-trained ViT models to compute fixed-length feature representations for each individual single-channel image: (1) `CHAMMI-75` DINO-BoC: our proposed model trained with `CHAMMI-75`. (2) DINOv2: the generalist vision model adapted for microscopy features. (3) OpenPhenom: the channel-adaptive model trained on fixed-channel Cell Painting data. Given the difference in number of channels, we do not display multi-channel images in the visualizations because the bag-of-channels approach yields a variable-length feature representation across studies. We collect single-channel features with all models, including the channel-adaptive OpenPhenom model.

## D.2    RESULTS AND DISCUSSION

The UMAP visualizations for all three models (Figure D18) show that the feature space is primarily segregated by the source study ID (technical domain), rather than being uniformly mixed. This result leads to two key interpretations: (1) domain sensitivity vs. invariance: self-supervised models, even when trained on highly diverse data, do not produce strictly domain-invariant representations in the raw feature space. Instead, they are domain-sensitive, clustering the input based on technical factors (e.g., image acquisition parameters, microscopy type, local processing) unique to each of the 75 source studies. This strong inter-study differentiation highlights the genuine heterogeneity of `CHAMMI-75`. (2) Consistency with quantitative results: this qualitative observation aligns with our findings in the Disentanglement Experiments (Appendix ): since the technical signature remains present, a post-processing step like batch correction is necessary to remove this study-specific bias and maximize the detection of subtle biological signals. The strong clustering visible in the UMAP indicates that the high performance of `CHAMMI-75` features (Section 5.1) is achieved not by erasing the domain difference entirely, but by encoding these differences in a way that is highly distinguishable and amenable to correction. Further qualitative exploration by coloring the UMAP plots with other metadata fields (such as channel configuration or biological condition) reveals strong intra-study consistency, but the inter-study structure remains dominated by the technical source.

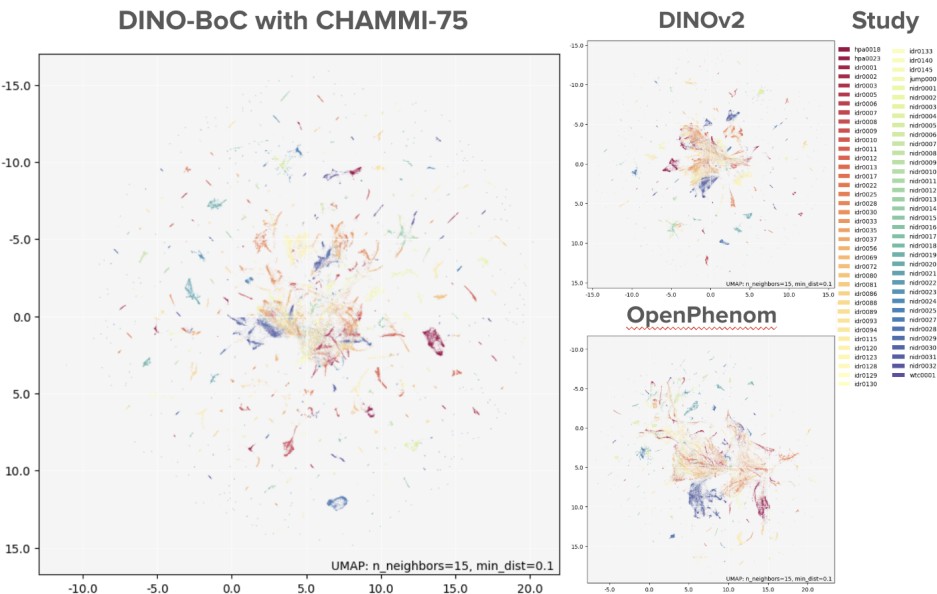

Figure D18: UMAP visualizations of the feature spaces obtained with three ViT-small models: DINO-BoC trained with `CHAMMI-75`, DINOv2, and OpenPhenom. The colors indicate source study, which is the dominant factor of variation in all three cases.

## E    DISENTANGLEMENT EXPERIMENTS

To quantitatively assess the quality of the learned representations in separating biological signal from technical noise, we adopt the batch correction benchmarking framework from (Arevalo et al., 2024). This framework is designed to evaluate how well a feature set disentangles batch signal (confounding variable) from biological signal (compound identity) by measuring metrics before and after the application of state-of-the-art batch correction algorithms. A high-quality feature set should: (1) require minimal correction in its raw state, and (2) maximize biological signal post-correction.

### E.1    EXPERIMENTAL PROTOCOL

We evaluate performance using Scenario 2 from (Arevalo et al., 2024): the classification of 302 landmark compounds generated by three different laboratories using the same microscope type. In this scenario, the batch variable is the Laboratory ID and the biological variable is the Compound. We assess the profiles using the population-averaged well-level approach to compute profiles. We compare three feature models: (1) CellProfiler Baseline: The original raw morphological features extracted using the conventional image analysis pipeline. (2) IDRCell Features: Features extracted using a ViT-Small DINO-BoC model trained on the smaller, heterogeneous IDRCell100K dataset ($\sim$ 100K images). (3) `CHAMMI-75` Features: Features extracted using our proposed ViT-Small DINO-BoC model trained on the full `CHAMMI-75` Large dataset. Based on the original study's findings, we focus on Seurat CCA as the optimal batch correction method for this scenario, which we apply to correct the features from all three models. Following standard practice, we report the aggregate score of four batch correction metrics, and six biological metrics, and the detail of these metrics is also reported for all three evaluated methods.

### E.2    RESULTS

The main results reported in Figure E19 highlight the following trend: (1) baseline state (no correction): `CHAMMI-75` features exhibit the best performance at the baseline level, yielding a higher raw biological signal, demonstrating better initial disentanglement of batch effects

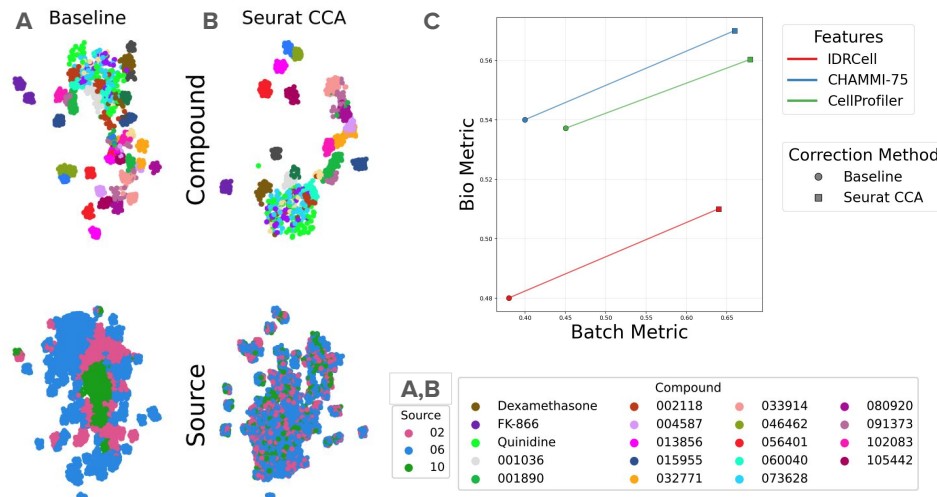

Figure E19: Disentanglement of feature representations using batch-correction on the JUMP-CP dataset, following Scenario 2 from the framework by (Arevalo et al., 2024). A) UMAP visualization of raw DINO-BoC features learned from `CHAMMI-75` and aggregated at the well-level. B) UMAP visualization of corrected DINO-BoC features from A corrected with the Seurat CCA algorithm. C) Scatter plot of batch and biological scores (x and y axes, respectively) for three feature representations: IDRCell in red (ViT-small trained with IDRCell dataset), `CHAMMI-75` in blue (ViT-small trained with `CHAMMI-75`), and CellProfiler in green (classical features). In both axis, higher performance is better.

compared to both baselines. This indicates that pre-training on the highly diverse and curated `CHAMMI-75` dataset yields representations that are inherently more robust to cross-site technical variation. (2) Post-correction state: after applying the Seurat CCA batch correction method, `CHAMMI-75` features continue to facilitate the best performance, resulting in the highest biological signal among all feature sets. The performance of `CHAMMI-75` features is 1.7% better than CellProfiler features, and 11.8% better than IDRCell features, all after correction. Detailed results of batch correction metrics are reported in Figure E20 for the `CHAMMI-75` features, in Figure E21 for IDRCell features, and in (Arevalo et al., 2024) for CellProfiler features.

This confirms that the learned `CHAMMI-75` feature space is structured in a way that is amenable to correction, allowing the batch correction algorithm to remove confounding noise without destroying the underlying biological coherence. Importantly, features learned with the IDRCell dataset consistently perform worse than the CellProfiler baseline in both the raw and corrected states. This contrast highlights that simply aggregating diverse data is insufficient; the scale, comprehensive heterogeneity, and careful curation embedded in `CHAMMI-75` are necessary factors for learning disentangled, biologically meaningful representations.

# F  BENCHMARK RESULTS

We will take a look at a larger resolution of numbers for all the benchmarks with more related results for better analysis.

## F.1  CHAMMI

CHAMMI is the benchmark for multi-channel microscopy imaging presented in (Chen et al., 2023). Here, we show all the values for CHAMMI showcased for dataset scaling laws in F9. We show sub-scores for WTC, HPA, and CP in F10, F11, and F12 respectively.

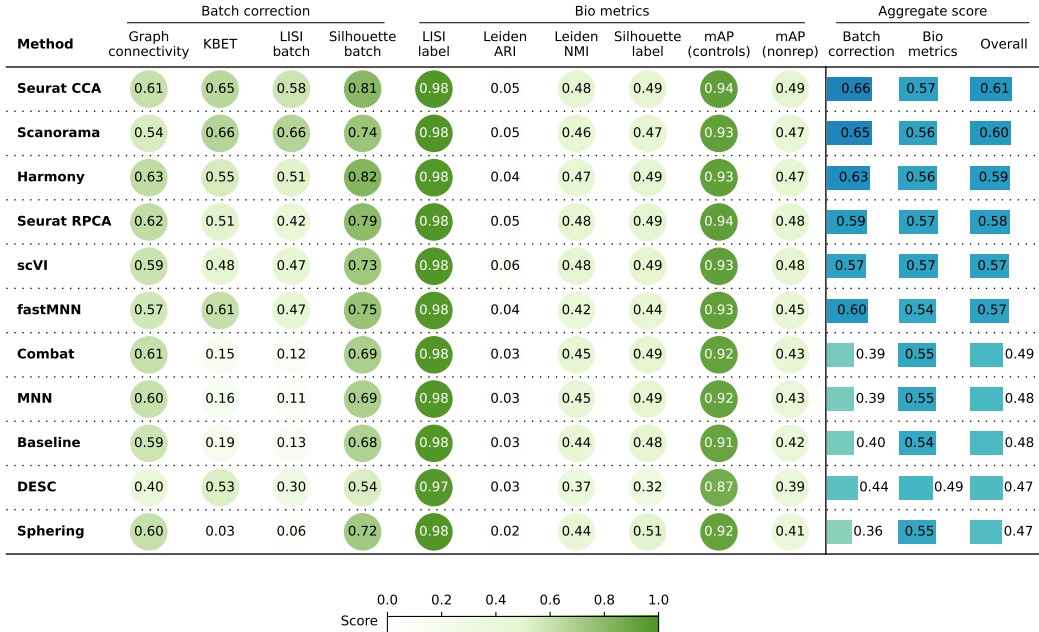

Figure E20: Detailed report of batch correction metrics for the DINO-BoC model trained with `CHAMMI-75`

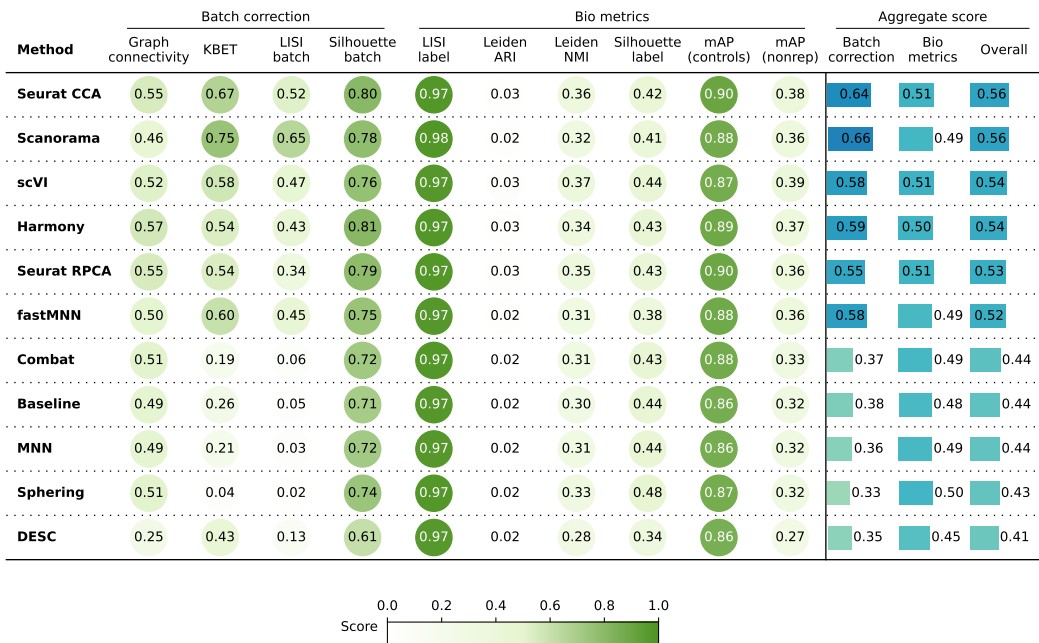

Figure E21: Detailed report of batch correction metrics for the DINO-BoC model trained with IDRCell

Table F9: Summary of IID Mean, CHAMMI Scores for WTC, HPA, and CP sets

| Model | IID Mean | CHAMMI Score | WTC | HPA | CP |
|---|---|---|---|---|---|
| MorphEm (ours) | 84.26 | 48.75 | 61.72 | 55.84 | 28.71 |
| OpenPhenom (ViT Small) | 75.55 | 38.23 | 42.80 | 43.15 | 28.74 |
| DINOv2 (ViT Small) | 73.95 | 37.93 | 46.27 | 39.66 | 27.87 |
| IDRCell (ViT Small) | 68.55 | 37.38 | 45.29 | 40.40 | 26.44 |

Table F10: WTC Dataset Benchmark Scores

| Model | Task 1 | Task 2 |
|---|---|---|
| MorphEm (ours) | 72.56 | 61.72 |
| OpenPhenom (ViT Small) | 59.72 | 42.80 |
| DINOv2 (ViT Small) | 63.56 | 46.27 |
| IDRCell | 56.33 | 45.29 |
| MCA-SSL (ViT Small) ours | 60.16 | 44.70 |

Table F11: HPA Dataset Benchmark Scores

| Model | Task 1 | Task 2 | Task 3 |
|---|---|---|---|
| MorphEm (ours) | 91.31 | 68.55 | 43.13 |
| OpenPhenom (ViT Small) | 78.84 | 50.91 | 35.40 |
| DINOv2 (ViT Small) | 70.07 | 53.00 | 26.33 |
| IDRCell (ViT Small) | 70.86 | 49.90 | 30.90 |

Table F12: CP Dataset Benchmark Scores

| Model | Task 1 | Task 2 | Task 3 | Task 4 |
|---|---|---|---|---|
| MorphEm (ours) | 88.90 | 51.73 | 21.83 | 12.57 |
| OpenPhenom (ViT Small) | 88.09 | 44.23 | 23.73 | 18.26 |
| DINOv2 (ViT Small) | 88.21 | 49.75 | 21.97 | 11.88 |
| IDRCell (ViT Small) | 79.29 | 51.43 | 20.63 | 07.25 |

### F.1.1 Tested Configurations for SubCell

Here, we showcase the number of SubCell configurations that were tested for the CHAMMI benchmark. Each row presents results with different models. The input channel configurations of four models are bg, rgb, ybg, and rybg, where each letter defines the input channel. The general protein of interest (g) that localizes into different cellular compartments is a mandatory input to all models together with nucleus (b) reference channel. The other two reference channels, microtubules (r) and endoplasmic reticulum (y), are present in two model configurations. The HPA benchmark uses the original channel configuration of the SubCell paper. For the WTC benchmark, the 'bg' model is run twice with different input channels ('g' as membrane or protein) followed by concatenation of these extracted features. A similar approach is utilized for the CP benchmark, where three different inputs (Syto, WGA, and Mitotracker) are used for 'g' input channel. We have reported all the configurations we tested for all the different models types for WTC, HPA, and CP in F13, F14, and F15 respectively.

Table F13: SubCell WTC Scores

| Model | WTC | Task 1 | Task 2 |
|---|---|---|---|
| MAE-CellS-ProtS-Pool bg | 55.06 | 78.22 | 55.06 |
| ViT-ProtS-Pool bg | 33.70 | 57.51 | 33.70 |

Table F14: SubCell HPA Scores

| Model | HPA | Task 1 | Task 2 | Task 3 |
|---|---|---|---|---|
| MAE-CellS-ProtS-Pool rybg | 64.58 | 96.26 | 84.01 | 45.15 |
| MAE-CellS-ProtS-Pool rbg | 71.63 | 95.59 | 88.07 | 55.19 |
| MAE-CellS-ProtS-Pool ybg | 66.93 | 95.03 | 79.73 | 54.12 |
| MAE-CellS-ProtS-Pool bg | 65.63 | 93.40 | 79.85 | 51.41 |
| ViT-ProtS-Pool rybg | 76.83 | 99.05 | 91.00 | 62.67 |
| ViT-ProtS-Pool rbg | 76.52 | 97.89 | 88.93 | 64.11 |
| ViT-ProtS-Pool ybg | 74.48 | 97.73 | 83.03 | 65.92 |
| ViT-ProtS-Pool bg | 72.96 | 96.06 | 82.70 | 63.23 |

Table F15: SubCell CP Scores

| Model | CP | Task 1 | Task 2 | Task 3 | Task 4 |
|---|---|---|---|---|---|
| MAE-CellS-ProtS-Pool ybg | 28.38 | 69.42 | 51.51 | 22.92 | 10.70 |
| ViT-ProtS-Pool ybg | 28.25 | 87.47 | 54.91 | 19.23 | 10.59 |

### F.2 HPAv23 at 256x256

HPAv23 at 256x256 is a version of the original SubCell training images where the resolution of the images has been reduced from 1024x1024 to 256x256 with a resize. This transformation was done to reduce the compute time, and storage taken by the test set. We ran protein localization with HPAv23 at 256x256 in both of its configurations: challenge_cats in F16 and all_unique_cats in F17.

Table F16: Protein Localization (Challenge Classification Labels)

| Model | Macro AP | Micro AP |
|---|---|---|
| MorphEm (ours) | 58.87 | 80.47 |
| OpenPhenom (ViT Small) | 49.13 | 75.75 |
| DINOv2 (ViT Small) | 53.76 | 77.01 |
| SubCell (ViT Base) | 69.33 | 84.79 |
| IDRCell (ViT Small) | 44.05 | 72.86 |

Table F17: Protein Localization (Unique Classification Labels)

| Model | Macro AP | Micro AP |
|---|---|---|
| MorphEm (ours) | 44.38 | 78.07 |
| OpenPhenom (ViT Small) | 35.98 | 73.13 |
| DINOv2 (ViT Small) | 39.67 | 74.84 |
| SubCell (ViT Base) | 52.60 | 82.58 |
| IDRCell (ViT Small) | 44.05 | 72.86 |

## F.3 JUMP-CP1 COMPOUNDS

Original JUMP-CP1 images were published with normalized (median absolute deviation *MAD-robustize*) well-level (replicate) CellProfiler features. Those features also include the ones that were extracted from brightfield images, for fair comparison those features were excluded. We also processed raw features by ofurselves: the same features were selected as in the paper version (except for features from brightfield channels) and performed ZCA-whitening (*spherize*) with pyCytominer in a similar way as we did for deep learning features. We also report the results obtained with Cell Painting CNN (Moshkov et al., 2024), feature post-processing was the same as for other deep learning features. Alongside the metrics *JUMP-CP1* and *JUMP-CP2*, we also report the *Active fraction*, that is a fraction of phenotypically active compounds versus negative controls and *mAP*-s from those active compounds contribute to *JUMP-CP1* result (Kalinin et al., 2025), otherwise *mAP* for non-active compounds would be zero. We reported our numbers with active fraction in F18.

Table F18: Phenotypic quality

| Model | Active fraction | JUMP-CP1 | JUMP-CP2 |
|---|---|---|---|
| MorphEm (ours) | 94.44% | 76.32 | 06.79 |
| CellProfiler (paper) | 81.70% | 58.71 | 04.00 |
| CellProfiler (ours) | 99.02% | 74.12 | 03.61 |
| Cell Painting CNN | 98.04% | 77.45 | 06.80 |
| OpenPhenom (ViT Small) | 96.08% | 74.26 | 04.99 |
| DINOv2 (ViT Small) | 94.44% | 75.84 | 07.03 |
| SubCell (ViT Base) | 95.42% | 77.60 | 07.44 |
| IDRCell (ViT Small) | 93.46% | 72.37 | 04.98 |

## F.4 IDR-17

IDR-17 is benchmarking based on a chemical–genetic interaction map of small molecules using high-throughput imaging in cancer cells. Add all the eight different cell lines, and the composite scores

We are going to be reporting all three metrics we have: ROC AUC scores in F21, Recall@50 in F20, Recall@100 in Table F19.

Table F19: Recall@100 Scores with Different Models

| Cell Line | DINOv2 | SubCell | OpenPhenom | MorphEm (ours) | IDRCell |
|---|---|---|---|---|---|
| HCT116 02-006 | 54.81 | 54.80 | 52.88 | 53.85 | 49.04 |
| HCT116 02-008 | 62.79 | 62.79 | 60.47 | 65.12 | 62.79 |
| HCT116 02-030 | 27.12 | 27.12 | 32.20 | 28.81 | 27.12 |
| HCT116 02-031 | 54.29 | 57.14 | 54.29 | 57.14 | 51.43 |
| HCT116 104-001 | 25.40 | 26.98 | 25.40 | 23.81 | 26.98 |
| HCT116 104-004 | 41.10 | 39.73 | 36.99 | 39.73 | 36.99 |
| HCT116 104-007 | 32.91 | 32.91 | 30.38 | 32.91 | 32.91 |
| HCT116 104-008 | 39.58 | 41.67 | 34.38 | 40.63 | 34.38 |
| **Average** | **42.24** | **42.89** | **40.87** | **42.75** | **40.20** |

Table F20: Recall@50 Scores for Different Models

| Cell Line | DINOv2 | SubCell | OpenPhenom | MorphEm (ours) | IDRCell |
|---|---|---|---|---|---|
| HCT116 02-006 | 29.80 | 28.85 | 28.85 | 29.81 | 28.85 |
| HCT116 02-008 | 46.51 | 44.19 | 46.51 | 46.15 | 44.19 |
| HCT116 02-030 | 15.25 | 15.25 | 15.25 | 15.25 | 15.25 |
| HCT116 02-031 | 31.43 | 28.57 | 28.57 | 31.43 | 31.43 |
| HCT116 104-001 | 17.46 | 15.87 | 17.46 | 15.87 | 15.87 |
| HCT116 104-004 | 23.29 | 20.54 | 21.92 | 23.28 | 20.55 |
| HCT116 104-007 | 20.25 | 20.25 | 20.25 | 20.25 | 21.52 |
| HCT116 104-008 | 20.83 | 18.75 | 18.75 | 20.83 | 17.71 |
| **Average** | **25.60** | **24.04** | **24.70** | **25.40** | **24.42** |

Table F21: AUROC Scores for Different Models

| Sample | DINOv2 | SubCell | OpenPhenom | MorphEm (ours) | IDRCell |
|---|---|---|---|---|---|
| HCT116 02-006 | 82.13 | 82.75 | 81.77 | 83.74 | 81.08 |
| HCT116 02-008 | 80.76 | 80.04 | 79.49 | 79.61 | 83.08 |
| HCT116 02-030 | 67.30 | 67.10 | 67.32 | 66.28 | 65.74 |
| HCT116 02-031 | 87.65 | 87.77 | 83.88 | 88.72 | 82.45 |
| HCT116 104-001 | 65.07 | 64.44 | 63.24 | 65.28 | 61.97 |
| HCT116 104-004 | 75.57 | 77.25 | 76.10 | 75.56 | 76.90 |
| HCT116 104-007 | 67.43 | 66.35 | 73.06 | 68.12 | 72.40 |
| HCT116 104-008 | 75.79 | 76.90 | 76.68 | 76.82 | 75.01 |
| **Average** | **75.21** | **75.37** | **75.19** | **75.52** | **74.83** |

### F.5 CELLPHIE

Here, we report all the statistics for the CellPHIE benchmark in Table F22

Table F22: Comparison of Neuron-Features with Different Models

| Neuron-Features | AUC | F1 | Precision | Recall |
|---|---|---|---|---|
| MorphEm (ours) | 80.51 | 77.45 | 79.91 | 75.54 |
| CellProfiler | 78.32 | 77.44 | 74.44 | 81.14 |
| DINOv1 (Pretrained on NF) | 73.72 | 72.30 | 74.76 | 70.44 |
| OpenPhenom (ViT Small) | 77.56 | 75.68 | 77.84 | 74.04 |
| DINOv2 (ViT Small) | 73.95 | 72.27 | 76.29 | 68.97 |
| SubCell (ViT Base) | 71.24 | 70.60 | 74.26 | 67.78 |

### F.6 RBC-MC

Here, we report all the statistics for the RBC-MC benchmark in Table F23, and Figure F23

Table F23: Comparision of RBC-MC with Different Models

| Model | Accuracy on Swiss | Accuracy on Canadian | Overall Accuracy |
|---|---|---|---|
| MorphEm (ours) | 68.34% | 65.06% | 66.70% |
| IDRCell | 55.85% | 51.18% | 53.50% |
| DINOv2 | 59.41% | 55.35% | 57.40% |
| OpenPhenom | 64.43% | 61.05% | 62.70% |
| SubCell | 59.10% | 53.50% | 56.30% |
| MCA-SSL | 59.50% | 56.90% | 58.20% |
| MCA-SupC | 55.40% | 51.20% | 53.30% |
| ChA-MAEViT | 61.5% | 56.60% | 59.00% |

## G  EXPERIMENTS

### G.1  MODEL CONFIGURATIONS AND PARAMETERS

All the models we trained have similar model configurations to have a fair comparison.

### G.1.1  MULTI-CHANNEL STRATEGIES

**Multi-Channel Attention (MCA).** Multi-channel attention models allow a ViT network to compute attention across channels. These networks work by unraveling the channel dimension into the sequence input of the vision transformer and use learned channel embeddings in order to denote which tokens belong to which channels (similar to positional encodings). These embeddings are learned per channel type (e.g. DNA and RNA encod-

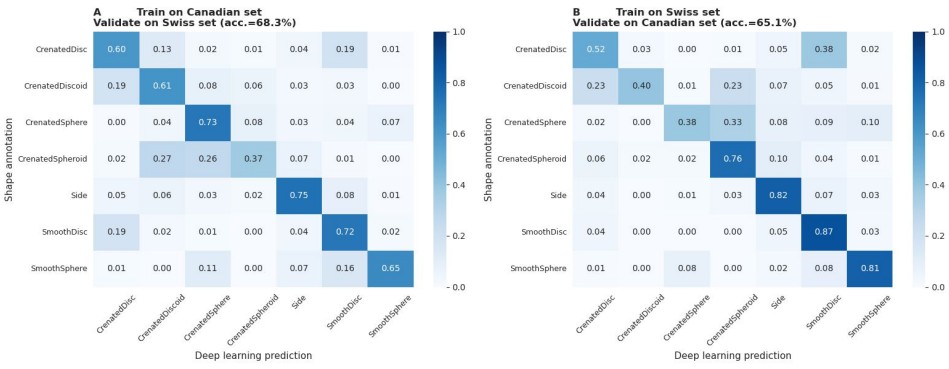

Figure F22: Confusion Matrices for RBC-MC Classification

ings) and effect all tokens in a given channel the same way. MCA was modified to use masked attention to handle mixed sequence lengths. Channels were padded with 0 tokens if sequences were of smaller length and an attention bias was used for the softmax to ignore these tokens. Models were trained with a patch size or 16, for 100 epochs in total.

**Bag of Channels (BoC).** Bag of Channels models is a strategy where we break all multi-channel images into single and the model learns from one single channel at a time. A drawback of this strategy is that the model does not learn inter-channel correlations as it only learns from one channel at a time. During evaluation, we usually concatenate the feature embedding space of all different test evaluation channels.

### G.1.2 Types of SSL Models

**DINO.** DINO is a self-supervised framework that employs a student-teacher network to learn representations in imaging (Caron et al., 2021).

**MAE.** MAE is a self-supervised framework that masks random patches and reconstructs the missing pixels (He et al., 2022).

**SimCLR.** SimCLR is a self-supervised framework that uses data augmentations to perform contrastive learning with images. (Chen et al., 2020).

### G.1.3 Model Parameters

**DINO BoC.** The default training parameters of the DINO model were used except - the learning rate changed to 5e-5. Batch sizes varied according to model sizes with 256, 128, and 32 used for ViT Small, ViT Base, and ViT Large respectively. We ran the model with horizontal and vertical flips for augmentations.

**MAE BoC.** The default training parameters of the MAE model were used except - learning rate changed to 5e-4, weight decay changed to 0.04, warm-up epochs changed to 10, and the number of epochs changed to 100 epochs. Batch sizes varied according to model sizes with 1024, 768, and 384 used for ViT Small, ViT Base, and ViT Large respectively. We ran the model with horizontal and vertical flips for augmentations.

**SimCLR BoC.** The default training parameters of the SimCLR model were used except - learning rate changed to 5e-5, weight decay changed to 0.04, warm-up epochs changed to 10, and the number of epochs changed to 100 epochs. The model was run with random resized crop with scale (0.2, 1.0), RandomHorizontalFlip, RandomVerticalFlip, and Gaussian blurring to develop two samples.

**Channel-ViT DINO.** Channel-ViT DINO replaces the fixed channel ViT backbone with a MCA ViT backbone and uses the standard DINO SSL algorithm to optimize the network. The multi channel id metadata was used to gather multi-channel images. DINO augmentations were than ran with horizontal and vertical flips with 8 local crops and 2 global crops. DINO global crops were 224x224 and local crops were 96x96 with crop ratios of 0.4-1.0 and 0.05-0.4 respectively. Learning rate was warmed up from 1e-6 to to 0.0001 for all models over 10 epochs. Teacher temperature was warmed up from 0.04 to 0.07 over 30 epochs. Weight Decay was warmed up from 0.04 to 0.04 over 10 epochs. The AdamW optimizer was used with default parameters other than those discussed previously. We trained with standard cosine annealing schedules as defined in (Caron et al., 2021). MCA DINO was modified to use masked attention to handle mixed sequence lengths. Channels were padded with 0 tokens if sequences were of smaller length and an attention bias was used for the softmax to ignore these tokens. Models were trained with a patch size or 16, for 100 epochs in total.

**Channel-ViT SimCLR.** Channel-ViT SimCLR replaces the fixed channel ViT backbone with an MCA ViT backbone and uses the standard SimCLR SSL algorithm. The multi_channel_id metadata was used to gather multi-channel images. The model was run with random resized crop with scale (0.2, 1.0), RandomHorizontalFlip, RandomVerticalFlip, and Gaussian blurring to develop two samples. The learning rate used was 5e-5.

**Channel-ViT MAE.** Channel-ViT MAE replaces the fixed channel ViT backbone with an MCA ViT backbone and uses the standard MAE SSL algorithm. The multi_channel_id metadata was used to gather multi-channel images. The model was run with RandomHorizontalFlip, and RandomVerticalFlip. The learning rate used was 5e-5.

### G.1.4 Supervised Baseline

ViT-small models were trained with a fixed number of channels per dataset in CHAMMI. A drop path rate of 0.2 was used for each block in the transformer. The AdamW optimizer was used with a learning rate of 0.001 and weight decay of 0.4. Augmentations of randomly resized crops, horizontal flips, rotations, gaussian blurring and self-normalization were used. A prediction head was used to go from the CLS token to the number of classes in each dataset. This was a linear head with L1 and L2 norms applied, with lambdas of 0.01. A cosign annealing along the learning rate was applied. No warmup was used. Standard cross entropy loss was used to train the network with 8 L40S GPUs for 60-100 epochs, depending on when scores stopped improving. The maximum score along the epochs was then used for the result of each CHAMMI dataset in the score G24.

Table G24: Summary of IID Mean, CHAMMI Score for Supervised Baseline)

| Model | IID Mean | CHAMMI Score | WTC | HPA | CP |
|---|---|---|---|---|---|
| Supervised Baseline | 78.76 | 54.56 | 65.21 | 71.84 | 26.64 |

### G.2 Resources Used

7-10 GPUs of NVidia A6000, L40S, and L40 were used with servers having 96 CPUs and server memory of 512 GB.

### G.3 Scaling Computations

Here, we describe how the computation varies for Bag of Channels and Multi-Channel Attention models. Figure G23 showcases how much more compute time Multi-channel models take as compared to Bag of Channels models.

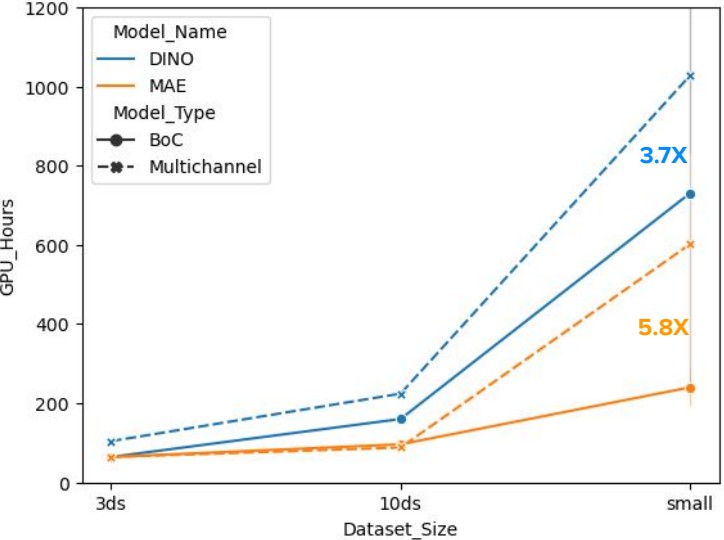

Figure G23: Compute time in GPU hours (vertical axis) of model training when the dataset size is increased (horizontal axis). The dataset sizes are 3ds (100K images), 10ds (150K images), and small (74ds with 560K images).

### G.4 Additional Experiments

#### G.4.1 Dataset Scaling with HPAv23 at 256x256

Our evaluation mainly used CHAMMI as an indicator of model performance in the scalability analysis where the number of sampled images used for training is increased. Table G25 reports the results of evaluating these models in the HPAv23 at 256x256 benchmark. The results show a trend similar to the CHAMMI score in Figure 8, where performance improves with more data.

Table G25: DINO Dataset Scaling Using HPAv23 at 256x256.

| Dataset Configuration | Macro AP |
|---|---|
| 3 Datasets | 41.00 |
| 10 Datasets | 53.40 |
| CHAMMI-75 Small | 54.92 |
| CHAMMI-75 Large | 59.17 |

#### G.4.2 Hi-res vs Low-res HPA experiments

We adopted the HPAv23 dataset for benchmarking protein localization classification. This dataset contains single-cell images at high resolution (1024x1024), which has high compute cost at test time. To accelerate evaluations and faciliate comparisons across models, we reduced the size of the images to 256x256 pixels, resulting in lower resolution.

Table G26 reports the differences in performances between high resolution classification and low resolution with images at 256x256 pixels. The model used in this evaluation is our best DINO-BoC model trained with CHAMMI-75. The results indicate that using low resolution images results in lower performance, as expected. However, the drop is very small; less than one point in the Kaggle Challenge task, and about 3 points in micro average precision for the unique categories task. The largest drop observed is in macro average precision for the unique categories task, which reveals that some of the classes with rare examples are missed at low resolution.

We conclude that testing at low resolution is still meaningful, while allowing for faster evaluation. The evaluation time is reduced from approx. 1 hour at high resolution to less than 10 minutes at low resolution, both using eight NVidia GPUs.

Table G26: Comparison of Metrics for Hi-Res (1024x1024 pixels) vs Low-Res (256x256 pixels).

| Metric | Task | Hi-Res | Low-Res | Drop |
|---|---|---|---|---|
| Macro AP | Kaggle Challenge (19 categories) | 59.59 | 58.87 | -0.72 |
| Micro AP | Kaggle Challenge (19 categories) | 80.84 | 80.47 | -0.37 |
| Macro AP | Unique (31 categories) | 59.59 | 44.38 | -15.21 |
| Micro AP | Unique (31 categories) | 80.84 | 78.07 | -2.77 |

### G.5 Fine-Tuning Models on Downstream Tasks

While our experiments successfully utilize the pre-trained CHAMMI-75 model with only a linear probe (frozen backbone) to achieve state-of-the-art results, researchers may explore fine-tuning for specific, complex downstream tasks. However, based on our experience, we caution that full fine-tuning of multi-channel models for cell phenotyping is often challenging due to the scarcity of large, high-quality supervised labels, leading frequently to poor generalization and rapid overfitting.

**When to avoid fine-tuning (default recommendation).** We strongly recommend avoiding fine-tuning when the downstream task relies on subtle phenotypic differences without clean, manually validated labels (e.g., weak treatment labels in small screens). Also when the available labeled data is small, as a ViT model (even small size) tends to quickly

overfit to the batch-specific signal in the training set. In such cases, extracting features with a frozen `CHAMMI-75` backbone and training a simple linear classifier (as done in our HPAv23 and RBC-MC benchmarks) is the most robust and computationally efficient approach.

**Best practices for fine-tuning (only if necessary).** If a researcher must fine-tune the backbone (e.g., if highly clean, validated annotation sets like those in HPAv23 are available), they should follow these steps: (1) Input Standardization: the input images must follow the same preprocessing steps applied during `CHAMMI-75` training, which include intensity normalization by applying 0.1% and 99.9% percentile clipping per channel, followed by per-channel self-normalization. (2) Spatial cropping: use single-cell coordinates to ensure crops are centered on relevant cellular content. Architecture adaptation (bag-of-channels): for a target task with $N$ channels, adapt the BoC architecture by fixing $N$. This involves replicating the pre-trained weights of the first layer (the convolution or patch embedding layer) $N$ times to accept the $N$-channel input image. The remainder of the model architecture (the transformer blocks) remains unchanged and loaded with the pre-trained weights. (3) Training protocol: employ conservative training schedules with a very low learning rate (e.g., $10^{-5}$ to $10^{-6}$) for the feature extractor backbone. Use strong regularization (e.g., high weight decay) to prevent catastrophic forgetting of the general knowledge learned from `CHAMMI-75`.

We reiterate that none of the results reported in this paper were obtained through fine-tuning; instead, we found that linear probing yielded the most stable and generalizable performance across all phenotyping tasks.

