# OpenReview forum: "CHAMMI-75: Pre-training multi-channel models with heterogeneous microscopy images"
_ICLR.cc/2026/Conference — ICLR 2026 Poster_

### Official Review · Reviewer_q2zP · 2025-10-25

**Soundness:** 3
**Presentation:** 3
**Contribution:** 3
**Rating:** 4
**Confidence:** 4

**Summary:**

This paper presents CHAMMI-75, a large-scale multi-channel microscopy dataset that aggregates 75 public biological studies and 18 imaging platforms. The authors provide detailed documentation of data collection, cleaning, redundancy control, and scaling studies comparing SSL frameworks on this dataset.
While CHAMMI-75 fills an important data gap and is technically well executed, the paper focuses primarily on dataset construction and benchmarking. It lacks methodological innovations addressing the intrinsic challenge of multi-channel adaptation and provides limited analysis on how dataset diversity affects model generalization.

**Strengths:**

1. The CHAMMI-75 dataset integrates 75 biological projects and 18 imaging platforms, providing unprecedented scale and diversity for microscopy SSL. This makes it a valuable benchmark for future research.
2. The authors conduct careful cleaning, redundancy reduction, and balanced sampling, improving representativeness and reproducibility.
3. The paper includes a thorough comparison of SSL methods and analyzes performance scaling with dataset size and channel configurations, validating the dataset’s utility.

**Weaknesses:**

1. The paper’s main contribution lies in data curation and benchmarking. The experimental pipeline relies heavily on existing models without proposing new architectures or training strategies tailored to multi-channel adaptation.
2. Although the dataset is multi-source and multi-channel, the study does not analyze how factors such as number of channels, organism diversity, or imaging modality affect representation learning. This limits understanding of what drives cross-domain generalization.
3. The reported SSL benchmarks focus on general downstream accuracy rather than the core motivation of channel generalization or cross-modality transfer, leaving a gap between dataset design and evaluation goals.

**Questions:**

The questions can be found in the weakness part. I understand this is a dataset and benchmark paper so the method part is not the research focus. However, additional validation experiments are essential for better understanding the benchmark. I am open to raise my score if more analysis is included in the rebuttal.

**Details Of Ethics Concerns:**

N.A.

---

> ### Author Response · Authors · 2025-11-26
> **Reply to Reviewer q2zP [Part 1]**
>
> We thank the reviewer for the constructive feedback, and for offering comments and suggestions to improve the clarity and experimental validations of our study. We have implemented all the recommendations, which has resulted in a higher quality manuscript. We discuss the detailed responses below.
>
> > **W1** The paper’s main contribution lies in data curation and benchmarking. The experimental pipeline relies heavily on existing models without proposing new architectures or training strategies tailored to multi-channel adaptation.
>
> We thank the reviewer for their fair assessment and recognition that the paper’s main contribution is in data curation and benchmarking. As you succinctly stated,
>
> > I understand this is a dataset and benchmark paper so the method part is not the research focus.
>
> We concur and would like to respectfully re-emphasize that this work was submitted to the Datasets and Benchmarks topic. The purpose of this submission category is to introduce resources &mdash;like CHAMMI-75&mdash; that are crucial for accelerating machine learning research and unlocking new research directions by providing standardized, large-scale data and evaluation protocols.
>
> The creation of CHAMMI-75 addresses a fundamental bottleneck in bioimage analysis: the lack of a large-scale, heterogeneous, and multi-channel dataset necessary to train and test generalizable "foundation models" for cellular imaging. By focusing on this challenging data curation effort, we aim to provide the essential prerequisite for the next wave of methodological research in channel-adaptive, multi-scale, and robust bioimage models. Our experimental pipeline, which relies on existing models like DINO-BOC and Channel-ViT, serves precisely to demonstrate the immediate and strong utility of CHAMMI-75 as a pre-training resource.
>
> We have added the following paragraph to the Introduction Section to clarify the contribution of our work (Section 1, Line numbers 90-101):
>
> *“This work addresses the data gap hindering the development of generalizable models for cellular imaging. Progress in machine learning is often catalyzed not only by algorithmic novelty but by the rigorous, large-scale data curation efforts that enable it. ImageNet [A] and LAION [B] are prominent examples that fundamentally shifted the focus of representation learning research and supported breakthroughs that would have been impossible without such data. In computational biology, challenging and realistic data is similarly necessary to advance the field. While significant progress has been achieved with fixed-channel image models, many open problems still exist to achieve a general understanding of cellular states regardless of the imaging technology. CHAMMI-75 is the first resource to integrate heterogeneous multi-channel imaging data at this scale in a way that directly facilitates the investigation of these challenges.”*
>
>
> [A] Jia Deng, Wei Dong, Richard Socher, Li-Jia Li, Kai Li, and Li Fei-Fei. Imagenet: A large-scale hierarchical image database. In 2009 IEEE conference on computer vision and pattern recognition, pp. 248–255. Ieee, 2009.
>
> [B] Christoph Schuhmann, Romain Beaumont, Richard Vencu, Cade Gordon, Ross Wightman, Mehdi Cherti, Theo Coombes, Aarush Katta, Clayton Mullis, Mitchell Wortsman, et al. Laion-5b: An open large-scale dataset for training next generation image-text models. Advances in neural information processing systems, 35:25278–25294, 2022.

---

> ### Author Response · Authors · 2025-11-26
> **Reply to Reviewer q2zP [Part 2]**
>
> > **W2** Although the dataset is multi-source and multi-channel, the study does not analyze how factors such as number of channels, organism diversity, or imaging modality affect representation learning. This limits understanding of what drives cross-domain generalization.
>
> We thank the reviewer for suggesting this valuable analysis. We agree that understanding the specific factors of variation that drive cross-domain generalization is crucial for advancing the field. We have performed five targeted data ablation studies using the CHAMMI benchmark to isolate the drivers of performance improvement. Our key finding is that the diversity of imaging modality (microscopy type) is the most critical factor, confirming that heterogeneity is the engine of generalization in CHAMMI-75. Models trained with high-diversity imaging modalities (the 12 non-dominant types) outperformed models trained solely on the two dominant, specialized modalities by 13% relative difference. Conversely, training with specialized data offered a 40% drop compared to training on all heterogeneous data. Variations in cell line or number of channels had negligible impact (1-3%).
>
> This comprehensive analysis directly addresses the reviewer's concern by providing quantitative evidence for what drives representation learning on our platform. The detailed methodology and all results are documented in Appendix B (line numbers 1660-1830), and a summary is integrated into the Results section (line numbers 396-431), as follows:
>
> *“5.3 Factors Impacting Representation Learning”*
>
> *“To understand which factors of the heterogeneous CHAMMI-75 dataset drive robust representation learning, we performed five targeted data ablation studies, systematically controlling and contrasting key metadata variables while measuring performance across the six out-of-distribution generalization tasks in the CHAMMI benchmark. These experiments revealed a clear hierarchy of factors that influence model performance (Figure 7). First, specialized vs. heterogeneous data: the single largest factor is the sheer diversity of the training data. Models trained with specialized data from a single task show a 40% relative performance drop compared to models trained on the full, heterogeneous CHAMMI-75 set. This strongly suggests that generalized knowledge gained from broad exposure outweighs highly specific, small-scale training.”*
>
> *“The second most influential factor is the diversity of imaging modality (microscopy type). When we contrasted a model trained only on the two most dominant and similar modalities (fluorescence and epi-fluorescence) against a model trained on the remaining twelve less-represented modalities, the high-diversity modality model outperformed the dominant-modality model by a 13% relative margin. This is a powerful finding that confirms our initial hypothesis: it is the exposure to a broad range of technical acquisition methods, rather than high volume within a narrow setup, that compels the model to learn truly generalizable representations. Variations in frequent biological factors, such as training without the most common cell line (U2OS or A549), or limiting training by the number of channels (e.g., training only on studies with up to two, three, or four channels), resulted in minor or negligible performance changes (1-3%) across the generalization tasks. This indicates that while highly diverse cell lines and varied channel counts contribute to overall robustness, the diversity of the physical imaging process (modality) is the primary technical variable that forces the model to learn a deeper, more abstract morphology.”*
>
> These ablation results confirm that CHAMMI-75's greatest value lies in its breadth of microscopy types, successfully establishing that diversity of modalities is the key driver for cross-domain generalization and superior representation learning in multi-channel bioimaging. All experimental setups, the full results, and detailed analyses are provided in Appendix B (line numbers 1660-1830).

---

> > ### Author Response · Authors · 2025-11-26
> > **Reply to Reviewer q2zP [Part 3]**
> >
> > > **W3** The reported SSL benchmarks focus on general downstream accuracy rather than the core motivation of channel generalization or cross-modality transfer, leaving a gap between dataset design and evaluation goals.
> >
> > We regret the confusion and thank the reviewer for pointing out this gap. We acknowledge that achieving a true zero-shot prediction of an entirely unseen channel type is impractical, as most common channels are represented in CHAMMI-75, even minimally (e.g., 3\% for bright-field). Therefore, we have adopted an expansive, real-world definition of channel generalization that focuses on the most frequent and challenging scenario in biological labs: novel channel combinations and cross-modality/domain transfer.
> >
> > To address this gap, our evaluation tests generalization on two fully held-out benchmarks (zero data overlap with pre-training) specifically designed to challenge models beyond traditional validations: (1) channel combination generalization (CellPHIE): predicting phenotypic outcomes using a unique, unprecedented 14-channel input (Cell Painting + 9 protein markers). (2) Cross-modality/domain generalization (RBC-MC): transferring features to a single-channel, label-free modality (flow cytometry) assessing a novel cell type (red blood cells) across two different clinical domains (Swiss vs. Canadian blood banks). The results in the new columns of Table 1 confirm that our DINO-BoC model trained on CHAMMI-75 significantly outperforms specialized and supervised baselines in both generalization tests, validating the alignment between our dataset design and evaluation goals.
> >
> > We have revised and expanded the discussion in Section 5 (Experiments and Results) (line numbers 378-394) with a new subsection to clarify the definition of channel generalization and interpret the strong results in the new benchmarks as follows:
> >
> > *“5.2 Channel Generalization and Cross Modality Transfer”*
> >
> > *“Given the representation of common microscopy channels in our pre-training set, we define and test channel generalization through the most challenging and realistic scenarios encountered in biological practice. First, generalization to novel channel combinations, which is highly frequent in laboratories by combining known channels in novel ways. The CellPHIE (CP) benchmark, with its unique 14-channel configuration, serves as a real-world test for this capability. Second, generalization to novel modalities and domains, extending beyond fluorescence to label-free modalities, novel cell types, and technical domain shifts. The RBC-MC (R) benchmark tests this by using single-channel bright-field imaging flow cytometry to classify red blood cell morphologies. Importantly, its paired cross-domain evaluation across clinical sites, challenges models to encode biologically relevant features.”*
> >
> > *“The results, summarized in the last two columns of Table 1, strongly validate our approach: the model trained with CHAMMI-75 (DINO-BoC) yields the best performance in both generalization challenges. This performance contrasts sharply with highly specialized models like SubCell, which lags behind when faced with these novel conditions. Specifically, our smaller, SSL-trained model outperforms the larger, WSL-trained SubCell by 11\% in CellPHIE and 13\% in RBC-MC. This result confirms that the large scale and diversity of CHAMMI-75 are essential factors for achieving robust channel and domain generalization.”*

---

> ### Comment · Reviewer_q2zP · 2025-11-27
> **Reply to Rebuttal**
>
> Thanks for the rebuttal and the added experiments, which has addressed most of my concerns, and I’m glad to raise my score to 6. I would still suggest moving some content added in lines 1660–1830 into the main text, as I think it is important for clarify the core contributions of the benchmark.

---

> > ### Author Response · Authors · 2025-12-03
> > **Thanks for the constructive feedback!**
> >
> > We thank the reviewer for the insightful and constructive feedback, and for suggesting the additional analysis related to factors that drive generalization and learning. We believe that the quality of the paper has significantly improved with the additional analysis, and we agree that the new content can be included in the main text. We will restructure the final version of the paper to clarify the core contributions of the dataset and benchmark.

---

### Official Review · Reviewer_6K5j · 2025-10-30

**Soundness:** 3
**Presentation:** 3
**Contribution:** 4
**Rating:** 8
**Confidence:** 4

**Summary:**

The paper presents CHAMMI-75, a dataset of heterogeneous multi-channel microscopy images for pre-training channel-adaptive cellular image analysis models. The dataset comprises 2.8 million multi-channel images from 75 diverse sources, featuring more than 1.8 billion cells, resulting in a curated, high-quality resource that exhibits greater biological and technical variation than other microscopy datasets used in prior work. The authors train vision transformer models using self-supervised methods, such as DINO, MAE, and SimCLR. The trained models are then evaluated on several downstream tasks, like cell-cycle stage classification and protein localization classification, among others.

**Strengths:**

The dataset’s large scale is impressive and would be a great unified resource for other researchers. The results for downstream tasks provide real value. The authors also added metadata information, specifically creating 22 metadata fields. Additionally, they segmented the nucleus of 1.8 billion cells using CellPose, which is very useful for the community. The benchmarks across different microscopy tasks demonstrate the generalizability of the proposed models. The statement from the authors to make the code and the dataset publicly available is highly appreciated and will help the community.

**Weaknesses:**

* While multiple benchmarks are used, the analysis could be strengthened by deeper biological interpretation of learned representations e.g., probing whether CHAMMI-75 pre-training improves biological feature disentanglement or transfer to unseen modalities

* The paper acknowledges missing and inconsistent metadata, but the implications for training robustness and domain bias are missing/underexplored.

* The evaluation seems to be focusing primarily on fluorescence microscopy. Brightfield datasets like LIVECell and EVICAN are part of the training set but aren’t used as an evaluation benchmark. Including label-free imaging in the evaluation would better display the transfer of learned representations across different modalities.
There are other recent papers like Segment Anything for Microscopy (Nature Methods, 2025), which discuss the cross-modality generalization in microscopy using foundation models for the task of segmentation. A clarification on how CHAMMI-75 complements or differs from such methods would highlight the contribution of this paper much more.

**Questions:**

* How does CHAMMI-75 handle cases where biological metadata (e.g., organism or stain identity) is ambiguous or inconsistent?

* Did the authors attempt cross-domain zero-shot evaluations (e.g., predicting unseen channel types)?

* How do the learned embeddings compare qualitatively (e.g., via t-SNE or UMAP) across datasets with different channel counts?
To better support the claim that CHAMMI-75 enables models to learn domain-invariant and biologically meaningful representations, I recommend including qualitative embedding visualizations such as UMAPs or t-SNE plots of the learned single-cell features. These could illustrate how samples cluster across different datasets, channel configurations, or biological conditions (e.g., cell type, treatment, protein localization). Comparing embeddings from representative models would provide intuitive evidence for cross-domain consistency and morphology awareness, complementing the quantitative benchmarks

* Can authors provide guidance for researchers who want to finetune models on their own microscopy data (e.g., normalization or channel alignment practices)?

---

> ### Author Response · Authors · 2025-11-26
> **Reply to Reviewer 6K5j [Part 1]**
>
> We thank the reviewers for the constructive feedback, and for offering comments and suggestions to improve the clarity and experimental validations of our study. We have implemented all the recommendations, which has resulted in a higher quality manuscript. We discuss the detailed responses below.
>
> > **W1** While multiple benchmarks are used, the analysis could be strengthened by deeper biological interpretation of learned representations e.g., probing whether CHAMMI-75 pre-training improves biological feature disentanglement or transfer to unseen modalities
>
> We thank the reviewer for this suggestion. We agree that validation of the quality and biological interpretability of the learned features is an important evaluation. We have introduced an evaluation of biological feature disentanglement to investigate the ability of representations learned from CHAMMI-75 to separate technical variation (batch) from true biological signal, following the framework of [A] Arevalo et al. (2024).
>
> In this evaluation, we quantitatively compare the capacity of features from three sources &mdash;raw CellProfiler (classical baseline), features learned on IDRCell (narrower dataset baseline), and features learned on CHAMMI-75 (our resource)&mdash; to successfully disentangle technical batch effects from biological signals (compound identity). We found that CHAMMI-75 features encode high-quality biological information and facilitate better batch correction. The following is a summary of our observations:
>
> - Before correction: CHAMMI-75 features possess the best baseline biological signal while simultaneously exhibiting less confounding batch signal compared to both CellProfiler and IDRCell features.
>
> - After correction: when corrected using the high-performing Seurat CCA algorithm, CHAMMI-75 features maximize the biological signal preservation, yielding improved downstream performance among all tested feature sets.
>
> - Dataset quality: features learned from the IDRCell dataset consistently perform worse than the raw CellProfiler baseline, demonstrating that large-scale heterogeneity and curation quality, as offered by CHAMMI-75, are essential for learning robust, disentangled feature spaces.
>
> These results and corresponding UMAP visualizations are detailed in a new section of the Appendix E (line numbers 2175-2302).

---

> > ### Author Response · Authors · 2025-11-26
> > **Reply to Reviewer 6K5j [Part 2]**
> >
> > > **W1** (continued)
> >
> > New Section in Appendix E (line numbers 2175-2302):
> >
> > *“E. Disentanglement Experiments”*
> >
> > *“To quantitatively assess the quality of the learned representations in separating biological
> > signal from technical noise, we adopt the batch correction benchmarking framework from
> > [A] Arevalo et al. (2024). This framework is designed to evaluate how well a feature set disen-
> > tangles batch signal (confounding variable) from biological signal (compound identity) by
> > measuring metrics before and after the application of state-of-the-art batch correction algo-
> > rithms. A high-quality feature set should: (1) require minimal correction in its raw state,
> > and (2) maximize biological signal post-correction.”*
> >
> >
> > *“E.1 Experimental protocol”*
> >
> > *“We evaluate performance using Scenario 2 from [A] Arevalo et al. (2024): the classification of 302 landmark compounds generated by three different laboratories using the same microscope type. In this scenario, the batch variable is the Laboratory ID and the biological variable is the Compound. We assess the profiles using the population-averaged well-level approach to compute profiles. We compare three feature models: (1) CellProfiler baseline: the original raw morphological features extracted using the conventional image analysis pipeline. (2) IDRCell features: extracted using a ViT-small DINO-BoC model trained on the smaller, heterogeneous IDRCell100K dataset (~100k images). (3) CHAMMI-75 features: extracted using our proposed ViT-small DINO-BoC model trained on the full CHAMMI-75 Large dataset. Based on the original study's findings, we focus on Seurat CCA as the optimal batch correction method for this scenario, which we apply to correct the features from all three models. Following standard practice, we report the aggregate score of four batch correction metrics, and six biological metrics, and the detail of these metrics is also reported for all three evaluated methods.”*
> >
> > *“E.2 Results”*
> >
> > *“The main results reported in Figure E17 highlight the following trend: (1) baseline state (no
> > correction): CHAMMI-75 features exhibit the best performance at the baseline level, yielding
> > a higher raw biological signal, demonstrating better initial disentanglement of batch effects
> > compared to both baselines. This indicates that pre-training on the highly diverse and cu-
> > rated CHAMMI-75 dataset yields representations that are inherently more robust to cross-site
> > technical variation. (2) Post-correction state: after applying the Seurat CCA batch correc-
> > tion method, CHAMMI-75 features continue to facilitate the best performance, resulting in
> > the highest biological signal among all feature sets. The performance of CHAMMI-75 features
> > is 1.7% better than CellProfiler features, and 11.8% better than IDRCell features, all after
> > correction. Detailed results of batch correction metrics are reported in Figure E18 for the
> > CHAMMI-75 features, in Figure E19 for IDRCell features, and in Arevalo et al. (2024) for
> > CellProfiler features.”*
> >
> > *“This confirms that the learned CHAMMI-75 feature space is structured in a way that is
> > amenable to correction, allowing the batch correction algorithm to remove confounding
> > noise without destroying the underlying biological coherence. Importantly, features learned
> > with the IDRCell dataset consistently perform worse than the CellProfiler baseline in both
> > the raw and corrected states. This contrast highlights that simply aggregating diverse data
> > is insufficient; the scale, comprehensive heterogeneity, and careful curation embedded in
> > CHAMMI-75 are necessary factors for learning disentangled, biologically meaningful repre-
> > sentations.”*
> >
> > Integration into Results Section (Section 5) (line numbers 506-519):
> >
> > *“Finally, we conducted an evaluation of biological feature disentanglement using the batch
> > correction framework of [A] Arevalo et al. (2024). The results indicate that features learned with
> > CHAMMI-75 are more robust to technical variation (requiring less batch correction) while
> > maximizing biological signal. Following correction with the Seurat CCA algorithm (Stuart
> > et al., 2019), our features achieve high separation of biological (compound) clusters from
> > technical (laboratory) batches, performing better than features extracted by CellProfiler
> > features and those learned from the IDRCell100K dataset. This validates that CHAMMI-75 is
> > a high-quality resource for learning disentangled features. This finding is further supported
> > by the qualitative analysis of our feature space (Appendix F), where UMAP visualizations
> > of single-channel features confirm that the primary clustering factor is the source study
> > (technical domain). This domain-sensitive learning indicates that models successfully encode
> > the immense heterogeneity of the dataset, confirming the necessity of post-processing steps
> > like batch correction to maximize biological signal (Appendix E).”*

---

> > > ### Author Response · Authors · 2025-11-26
> > > **Reply to Reviewer 6K5j [Part 3]**
> > >
> > > [A] John Arevalo, Ellen Su, Jessica D Ewald, Robert Van Dijk, Anne E Carpenter, and Shan-
> > > tanu Singh. Evaluating batch correction methods for image-based cell profiling. Nature
> > > Communications, 15(1):6516, 2024.
> > >
> > > > **W2** The paper acknowledges missing and inconsistent metadata, but the implications for training robustness and domain bias are missing/underexplored.
> > >
> > > We thank the reviewer for the feedback. To clarify, we highlight the following aspects of our work: first, our primary focus is Self-Supervised Learning (SSL), for which this dataset is particularly useful. Since SSL learns directly from pixel data, it is inherently robust to the metadata noise that creates challenges for supervised methods. Our SSL results in Table 1 show this approach is highly effective: our DINO-BOC model outperformed other SSL methods by 0.5-11% across 5 of the 6 benchmarks, and also obtained the best performance in the new bright-field RBC-MC benchmark, as well as improved disentanglement properties, as discussed in the previous two questions.
> > >
> > > Second, we investigated the metadata's utility via Weakly Supervised Learning (WSL), using the reagent identifiers as a supervised signal. As shown in Table 2, incorporating this information improved performance over the SSL baseline by 1-19% across all 6 benchmarks. This proves that the metadata, while imperfect, provides a signal consistent enough to facilitate learning and demonstrates training robustness.
> > >
> > > To better understand how metadata impacts WSL performance, we evaluated different strategies for incorporating reagent information, zero-shot evaluated on the CHAMMI benchmark. As shown in the Table below, naively using all reagents as labels (Exp. 3) hinders learning due to the long-tail distribution, where most reagents have few samples. Limiting training to the most frequent reagents (at least containing 100 samples) improves performance  (Exp. 2), but treating common and rare reagents differently &mdash; giving each common reagent its own representative label and grouping rare reagents into a few shared labels &mdash; yields the best results (Exp. 1).
> > >
> > > |Exp.| Incoporating metadata strategy | AVG Score | Allen | HPA | CP |
> > > |:-|:-|-:|-:|-:|-:|
> > > |1| Treat top common and other reagents separately | 39.80 | 25.09 | 64.72 | 29.60 |
> > > |2| Keep top 225 common reagents | 38.72 | 19.65 | 66.01 | 30.51 |
> > > |3| Naively use all (55,689) reagents | 37.37 | 25.29 | 60.21 | 26.62 |
> > >
> > > This suggests that CHAMMI-75 can serve as a valuable resource for studying robust learning strategies and developing models resilient to missing or inconsistent metadata.
> > >
> > > Finally, by curating 75 diverse studies with high technical and biological variation, we actively move away from the single-domain bias inherent in specialized datasets. Our strong SSL and WSL results on diverse, held-out benchmarks &mdash; such as CellPHIE, which has a novel 14-channel combination not seen in training &mdash; demonstrate that this approach produces models that are more general and robust to domain shift.

---

> > > > ### Author Response · Authors · 2025-11-26
> > > > **Reply to Reviewer 6K5j [Part 4]**
> > > >
> > > > > **W3A** The evaluation seems to be focusing primarily on fluorescence microscopy. Brightfield datasets like LIVECell and EVICAN are part of the training set but aren’t used as an evaluation benchmark. Including label-free imaging in the evaluation would better display the transfer of learned representations across different modalities.
> > > >
> > > >
> > > > We agree that including an evaluation benchmark for label-free imaging would better demonstrate the cross-modality transfer capabilities of CHAMMI-75. In addition to the newly added evaluation of biological feature disentanglement (discussed in our response to a previous question), we have addressed this by introducing a new, dedicated benchmark: *Red Blood Cell Morphology Classification (RBC-MC)*. This new benchmark focuses on classifying red blood cell (RBC) morphologies from single-channel, bright-field imaging flow cytometry (a label-free modality), using data from two paired, cross-domain studies (Swiss and Canadian blood banks). The results confirm the generalization capabilities of our model.
> > > >
> > > > For instance, DINO-BoC trained on the heterogeneous CHAMMI-75 dataset produces features that achieve approximately 66% accuracy in this challenging multi-class classification problem. This result outperforms most baseline methods, which typically score below 60%. The result is achieved despite bright-field images representing a very small fraction (2.68%) of the total pre-training data, demonstrating that CHAMMI-75 facilitates meaningful cross-modal learning even from underrepresented modalities.
> > > >
> > > > We have fully integrated this new benchmark and discussion into the revised manuscript: (a) Figure 6 now includes the details of the RBC-MC benchmark. (b) Section 4 has been updated with its description and protocol. (c) A new subsection in Section 5 discusses “Channel Generalization and Cross Modality Transfer” using these results. (d) Experimental specifics have been added to Appendix A.9. (e) Desired tables for the new benchmark have been added to Appendix D.6.
> > > >
> > > > Addition to Section 4: Evaluation Benchmarks (New Benchmark Description) (Line numbers 309-315)
> > > >
> > > > *“Red Blood Cell Morphology Classification (RBC-MC). A set of 130,560 single-channel bright-field images (48x48 pixels) of RBCs obtained via imaging flow cytometry from two distinct clinical sites (Swiss and Canadian blood banks) [A]. The task is a multi-class classification into seven clinically relevant morphological categories associated with blood quality. The benchmark employs a cross-domain validation protocol with a linear probe trained on data from one site and tested on the other (Swiss vs Canadian).”*
> > > >
> > > > Addition to Section 5.2 (Line numbers 378-395):
> > > >
> > > > *“Channel Generalization and Cross Modality Transfer”*
> > > >
> > > > *“We explore channel generalization through the most challenging and realistic scenarios en-
> > > > countered in biological practice. First, generalization to novel channel combinations, which
> > > > is very frequent in laboratories by combining known channels in novel ways. The CellPHIE
> > > > (CP) benchmark, with its unique 14-channel configuration, serves as a real-world test for
> > > > this capability. Second, generalization to novel modalities and domains. The RBC-MC (R)
> > > > benchmark tests this by using single-channel bright-field imaging flow cytometry (modal-
> > > > ity) to classify red blood cell morphologies. Importantly, its paired cross-domain evaluation
> > > > across clinical sites, challenges models to encode biologically relevant features.”*
> > > >
> > > > *“The results, summarized in the last two columns of Table 1, strongly validate our approach:
> > > > the model trained with CHAMMI-75 (DINO-BoC) yields the best performance in both gener-
> > > > alization challenges. This performance contrasts sharply with highly specialized models like
> > > > SubCell, which lags behind when faced with these novel conditions. Specifically, our smaller,
> > > > SSL-trained model outperforms the larger, WSL-trained SubCell by 11% in CellPHIE and
> > > > 13% in RBC-MC. This result confirms that the large scale and diversity of CHAMMI-75 are
> > > > essential factors for achieving robust channel and domain generalization.”*
> > > >
> > > > [A] Minh Doan, Joseph A Sebastian, Juan C Caicedo, Stefanie Siegert, Aline Roch, Tracey R
> > > > Turner, Olga Mykhailova, Ruben N Pinto, Claire McQuin, Allen Goodman, et al. Ob-
> > > > jective assessment of stored blood quality by deep learning. Proceedings of the National
> > > > Academy of Sciences, 117(35):21381–21390, 2020.

---

> > > > > ### Author Response · Authors · 2025-11-26
> > > > > **Reply to Reviewer 6K5j [Part 5]**
> > > > >
> > > > > > **W3B** There are other recent papers like Segment Anything for Microscopy (Nature Methods, 2025), which discuss the cross-modality generalization in microscopy using foundation models for the task of segmentation. A clarification on how CHAMMI-75 complements or differs from such methods would highlight the contribution of this paper much more.
> > > > >
> > > > > We agree that distinguishing our contribution from other successful foundation models efforts is important. We clarify in the Introduction that CHAMMI-75 focuses exclusively on cell phenotyping, a distinct task from image segmentation. While models like Segment Anything for Microscopy address the problem of generating precise spatial masks for cells and nuclei across various modalities, our work is dedicated to training models that quantify subtle morphological differences necessary for biological discovery. Both tasks are important in bioimage analysis but serve different purposes. We propose inserting the following into the Introduction (Section 1) (line numbers 101-107):
> > > > >
> > > > > *”The purpose of CHAMMI-75 is to facilitate the creation of models for cell phenotyping, which is the task of identifying and quantifying morphological differences between cellular states (e.g., healthy vs. perturbed cells). This goal is distinct from, but complementary to, other critical bioimage analysis tasks such as cell segmentation, where foundation models like Segment Anything for Microscopy [A] have demonstrated success in generalized mask generation across modalities. CHAMMI-75 provides the data backbone necessary to train and evaluate models that can detect subtle biological classification signals, which cannot be achieved through segmentation alone.”*
> > > > >
> > > > > [A] Anwai Archit, Luca Freckmann, Sushmita Nair, Nabeel Khalid, Paul Hilt, Vikas Rajashekar,
> > > > > Marei Freitag, Carolin Teuber, Melanie Spitzner, Constanza Tapia Contreras, et al. Seg-
> > > > > ment anything for microscopy. Nature Methods, 22(3):579–591, 2025.

---

> > > > > > ### Author Response · Authors · 2025-11-26
> > > > > > **Reply to Reviewer 6K5j [Part 6]**
> > > > > >
> > > > > > > **Q1** How does CHAMMI-75 handle cases where biological metadata (e.g., organism or stain identity) is ambiguous or inconsistent?
> > > > > >
> > > > > > We thank the reviewer for highlighting the challenge of metadata consistency in large-scale bioimaging efforts. Handling missing, ambiguous, and inconsistent metadata was one of the most resource-intensive steps in curating CHAMMI-75, as it required aligning data that was never intended to be integrated. The heterogeneity of CHAMMI-75 means that the metadata naturally reflects the real-world status of bioimaging data: it is often highly noisy, incomplete, and non-standardized. We handled these issues through a multi-tiered approach:
> > > > > >
> > > > > > - Missing data: if information was not provided in the original source, it is consistently labeled as "unknown".
> > > > > >
> > > > > > - Ambiguous or unclear data: we performed manual and semi-automated interpretation (including web searches and LLM-assisted parsing, as described in Appendix A.3) to resolve ambiguities. If a reliable interpretation could not be found, the original value was preserved, or the entry was conservatively labeled as "unknown".
> > > > > >
> > > > > > - Inconsistencies: We enforced standardization across studies by manually parsing and normalizing values (e.g., using consistent low-casing, removing special characters, and mapping similar terms to a single entry using existing biological ontologies/dictionaries).
> > > > > >
> > > > > > Our main goal was not to "fix" all noise &mdash; which would require prohibitive effort and introduce new subjective bias &mdash; but rather to provide the cleanest, most consistent resource possible under the circumstances. We believe this metadata represents a realistic technical challenge that future multi-modal models may be able to learn to overcome. Details of this process are expanded in Appendix A.5 (line numbers 1288-1344).
> > > > > >
> > > > > > *“**A.5 Metadata quality and handling inconsistencies** ”*
> > > > > >
> > > > > > *“The construction of CHAMMI-75 involved integrating 75 heterogeneous studies, many of which lacked standardized or complete metadata, reflecting a general deficiency in data sharing practices across biology. Our approach to handling missing, inconsistent, and ambiguous data was rigorous yet pragmatic, acknowledging that perfect retrospective standardization is often impossible.”*
> > > > > >
> > > > > > *“**Handling missing data**. The vast majority of studies provided partial metadata. For fields where information was unequivocally absent (e.g., biology.cell_type or microscopy.fov), the entry was explicitly and consistently annotated with the string "unknown". This approach avoids using null values or empty strings, ensuring consistency for subsequent programmatic access and filtering.”*
> > > > > >
> > > > > > *“**Resolving ambiguity and inconsistency.** In cases where metadata was ambiguous (e.g., multiple spellings for a cell line, or an overly verbose stain identity), we employed a multi-step resolution pipeline: (1) Cross-reference: values were validated against other metadata columns (e.g., cross-checking organism and cell line). (2) External validation: we performed targeted web searches and reviewed the associated scientific publications to infer the most likely accurate value. (3) LLM-assisted parsing: As noted in Appendix A.3, we used LLMs to systematically extract and organize certain information, providing a rapid, initial pass at resolving naming variants and extracting structured data from free-form text descriptions. (4) Manual curation and normalization: to minimize inconsistencies, all resulting values were manually reviewed, low-cased, and mapped to a simplified vocabulary or ontology where possible (e.g., different fluorescence channels like 'DAPI', 'Hoechst 33342', and 'Nuclear stain' were often mapped under the canonical channel_type of 'nucleus').”*
> > > > > >
> > > > > > *“**Last resource.** If, after all these steps, the interpretation remained uncertain (e.g., a stain was ambiguously described and the original publication offered no clarity), we defaulted to preserving the original value or, more conservatively, labeling the entry as "unknown".”*
> > > > > >
> > > > > > *“The resulting metadata table, while significantly cleaned, remains inherently noisy. We view this noise and high sparsity (Figure 4b) not as a limitation, but as a defining feature of the resource. Unlike highly standardized datasets produced synchronously by a single consortium, CHAMMI-75 attempted to align data that were never intended for cross-study integration. This noisy complexity presents the real-world challenge foundation models for bioimaging must solve &mdash;to learn representations that are robust despite inconsistent. Future efforts should focus on promoting standardized metadata acquisition for new studies [A], as retrospectively fixing metadata is computationally infeasible and prone to subjective error. Furthermore, our dataset provides a compelling target for future research into semi-automated, multi-modal systems capable of resolving these remaining metadata inconsistencies.”*

---

> > > > > > > ### Author Response · Authors · 2025-11-26
> > > > > > > **Reply to Reviewer 6K5j [Part 7]**
> > > > > > >
> > > > > > > [A] Christopher Schmied, Michael S Nelson, Sergiy Avilov, Gert-Jan Bakker, Cristina Bertocchi,
> > > > > > > Johanna Bischof, Ulrike Boehm, Jan Brocher, Mariana T Carvalho, Catalin Chiritescu,
> > > > > > > et al. Community-developed checklists for publishing images and image analyses. Nature
> > > > > > > Methods, 21(2):170–181, 2024.
> > > > > > >
> > > > > > > > **Q2** Did the authors attempt cross-domain zero-shot evaluations (e.g., predicting unseen channel types)?
> > > > > > >
> > > > > > > Given the heterogeneous composition of CHAMMI-75 (25 channel types), most common microscopy channel signals are likely represented, even if minimally (e.g., bright-field is 3% of single channel images), making a true zero-shot evaluation on an entirely unseen channel type challenging. We note that new channel types are less likely to emerge (or less frequently), and instead the most common scenarios in biological research include new combinations of existing channels and new biological and experimental conditions (bread-and-butter situation in microscopy).
> > > > > > >
> > > > > > > For this reason, we define cross-domain generalization expansively by evaluating in two fully held-out benchmarks with novel conditions:
> > > > > > >
> > > > > > > - Novel Channel Combination (CellPHIE): Predicting phenotypic outcomes using an unprecedented 14-channel input configuration, representing a complex extrapolation task.
> > > > > > >
> > > > > > > - Novel Modality/Cell Type (Blood Quality): Transferring learned features to a label-free imaging flow cytometry technique (novel modality) assessing red blood cells (novel cell type) across two different clinical domains (technical cross domain matching).
> > > > > > >
> > > > > > > In both cases, we ensured zero overlap with the pre-training data. Our findings demonstrate that training with CHAMMI-75 consistently yields features that outperform specialized methods in these novel conditions, supporting the concept of improved generalization. For future research, we believe that beyond new channel types, which are rare in new biological experiments, generalization benchmarks should include new biological and experimental conditions. This is likely to better represent the needs of the biological community in terms of robust performance for new experiments.

---

> > > > > > > > ### Author Response · Authors · 2025-11-26
> > > > > > > > **Reply to Reviewer 6K5j [Part 8]**
> > > > > > > >
> > > > > > > > > **Q3** How do the learned embeddings compare qualitatively (e.g., via t-SNE or UMAP) across datasets with different channel counts? To better support the claim that CHAMMI-75 enables models to learn domain-invariant and biologically meaningful representations, I recommend including qualitative embedding visualizations such as UMAPs or t-SNE plots of the learned single-cell features. These could illustrate how samples cluster across different datasets, channel configurations, or biological conditions (e.g., cell type, treatment, protein localization). Comparing embeddings from representative models would provide intuitive evidence for cross-domain consistency and morphology awareness, complementing the quantitative benchmarks
> > > > > > > >
> > > > > > > > We have included UMAP visualizations of single-channel features (extracted using DINO-BoC) sampled across all 75 studies in the Appendix (Section F). The visualizations confirm that the source study is the dominant factor of variation in the feature space. Even with models trained on CHAMMI-75, the representations are primarily grouped by the originating dataset (i.e., technical domain), indicating that the models have learned domain-sensitive, rather than domain-invariant, features. Note that this result does not contradict previous results or claims in the earlier version of the manuscript (domain invariance was never claimed). Instead, this finding is consistent with our new quantitative results in the Disentanglement Experiments (Appendix E), which show that batch correction is still necessary to maximize biological signal after feature extraction. We interpret this as the self-supervised models successfully encoding the distinct technical signatures present in the heterogeneous datasets. Our core claim is thus refined: **CHAMMI-75 enables models to learn robust and biologically relevant representations that are amenable to correction and display strong inter-study differentiation**. We added the following subsection to Appendix F (line numbers 2303-2365), detailing the qualitative analysis and results:
> > > > > > > >
> > > > > > > > *“Appendix F. Qualitative analysis of feature space
> > > > > > > > To visualize the global structure of the feature space learned by models trained on CHAMMI-75, we performed a qualitative analysis using UMAP projection on single-channel features extracted from a balanced, single-cell sample of all 75 source studies.”*
> > > > > > > >
> > > > > > > > *“F.1 Experimental protocol”*
> > > > > > > >
> > > > > > > > *“We sampled approximately 1,000 single-cell, multi-channel crops from images in each of the 75 source studies, resulting in 196,660 individual single-channel images. We then used three representative pre-trained ViT models to compute fixed-length feature representations for each individual single-channel image: (1) CHAMMI-75 DINO-BoC: our proposed model trained with CHAMMI-75. (2) DINOv2: the generalist vision model adapted for microscopy features. (3) OpenPhenom: the channel-adaptive model trained on fixed-channel Cell Painting data. Given the difference in number of channels, we do not display multi-channel images in the visualizations because the bag-of-channels approach yields a variable-length feature representation across studies. We collect single-channel features with all models, including the channel-adaptive OpenPhenom model.”*
> > > > > > > >
> > > > > > > > *“F.2 Results and Discussion”*
> > > > > > > >
> > > > > > > > *“The UMAP visualizations for all three models (Figure DX) show that the feature space is primarily segregated by the source study ID (technical domain), rather than being uniformly mixed. This result leads to two key interpretations: (1) domain sensitivity vs. invariance: self-supervised models, even when trained on highly diverse data, do not produce strictly domain-invariant representations in the raw feature space. Instead, they are domain-sensitive, clustering the input based on technical factors (e.g., image acquisition parameters, microscopy type, local processing) unique to each of the 75 source studies. This strong inter-study differentiation highlights the genuine heterogeneity of CHAMMI-75. (2) Consistency with quantitative results: this qualitative observation aligns with our findings in the Disentanglement Experiments (Appendix D.1): since the technical signature remains present, a post-processing step like batch correction is necessary to remove this study-specific bias and maximize the detection of subtle biological signals. The strong clustering visible in the UMAP indicates that the high performance of CHAMMI-75 features (Section 5.1) is achieved not by erasing the domain difference entirely, but by encoding these differences in a way that is highly distinguishable and amenable to correction. Further qualitative exploration by coloring the UMAP plots with other metadata fields (such as channel configuration or biological condition) reveals strong intra-study consistency, but the inter-study structure remains dominated by the technical source.”*
> > > > > > > >
> > > > > > > > We have made additions to Results Section, documented in next comment.

---

> > > > > > > > > ### Author Response · Authors · 2025-11-26
> > > > > > > > > **Reply to Reviewer 6K5j [Part 9]**
> > > > > > > > >
> > > > > > > > > > **Q3** (continued)
> > > > > > > > >
> > > > > > > > > Addition to the Results Section (Section 5, line numbers 512-518):
> > > > > > > > >
> > > > > > > > > *“This finding is further supported by the qualitative analysis of our feature space (Appendix F), where UMAP visualizations of single-channel features confirm that the primary clustering factor is the source study (technical domain). This domain-sensitive learning indicates that models successfully encode the immense heterogeneity of the dataset, confirming the necessity of post-processing steps like batch correction to maximize biological signal (Appendix E).”*
> > > > > > > > >
> > > > > > > > > > **Q4** Can authors provide guidance for researchers who want to finetune models on their own microscopy data (e.g., normalization or channel alignment practices)?
> > > > > > > > >
> > > > > > > > > We advise researchers to prioritize linear probing (feature extraction with a frozen backbone) over full fine-tuning for most cell phenotyping tasks, as our results demonstrate this approach is highly competitive across diverse benchmarks with little risk of catastrophic forgetting or overfitting. For researchers who must fine-tune (typically only if their problem requires a higher-level feature abstraction or if a very large, clean annotation set is available), we offer the following guidance, which has been expanded in Appendix G.5 (line numbers 2517-2530):
> > > > > > > > >
> > > > > > > > > Normalization: use the same per-channel intensity standardization applied during pre-training (0.1% and 99.9% percentile clipping, followed by 8-bit rescaling, and per-channel self-normalization). Channel alignment: fix the channel configuration of the input data and adapt a Bag-of-Channels (BoC) architecture by replicating the first-layer weights to match the fixed number of channels. Training strategy: use low learning rates and appropriate regularization to mitigate overfitting, which we observed frequently in our experiments (Details can be found in our code which will be released with the dataset).
> > > > > > > > >
> > > > > > > > > We added the following subsection to Appendix G.5 (line numbers 2498-2535):
> > > > > > > > >
> > > > > > > > > *“B.5 Fine-Tuning Models on Downstream Tasks”*
> > > > > > > > >
> > > > > > > > > *“While our experiments successfully utilize the pre-trained CHAMMI-75 model with only a linear probe (frozen backbone) to achieve state-of-the-art results, researchers may explore fine-tuning for specific, complex downstream tasks. However, based on our experience, we caution that full fine-tuning of multi-channel models for cell phenotyping is often challenging due to the scarcity of large, high-quality supervised labels, leading frequently to poor generalization and rapid overfitting.”*
> > > > > > > > >
> > > > > > > > > *“**When to avoid fine-tuning (default recommendation).** We strongly recommend avoiding fine-tuning when the downstream task relies on subtle phenotypic differences without clean, manually validated labels (e.g., weak treatment labels in small screens). Also when the available labeled data is small, as a ViT model (even small size) tends to quickly overfit to the batch-specific signal in the training set. In such cases, extracting features with a frozen CHAMMI-75 backbone and training a simple linear classifier (as done in our HPAv23 and RBC-MC benchmarks) is the most robust and computationally efficient approach.”*
> > > > > > > > >
> > > > > > > > > *“**Best practices for fine-tuning (only if necessary).** If a researcher must fine-tune the backbone (e.g., if highly clean, validated annotation sets like those in HPAv23 are available), they should follow these steps: (1) Input Standardization: the input images must follow the same preprocessing steps applied during CHAMMI-75 training, which include intensity normalization by applying 0.1\% and 99.9\% percentile clipping per channel, followed by per-channel self-normalization. (2) Spatial cropping: use single-cell coordinates to ensure crops are centered on relevant cellular content. Architecture adaptation (bag-of-channels): for a target task with $N$ channels, adapt the BoC architecture by fixing $N$. This involves replicating the pre-trained weights of the first layer (the convolution or patch embedding layer) $N$ times to accept the $N$-channel input image. The remainder of the model architecture (the transformer blocks) remains unchanged and loaded with the pre-trained weights. (3) Training protocol: employ conservative training schedules with a very low learning rate (e.g., $10^{-5}$ to $10^{-6}$) for the feature extractor backbone. Use strong regularization (e.g., high weight decay) to prevent catastrophic forgetting of the general knowledge learned from CHAMMI-75.”*
> > > > > > > > >
> > > > > > > > > *“We reiterate that none of the results reported in this paper were obtained through fine-tuning; instead, we found that linear probing yielded the most stable and generalizable performance across all phenotyping tasks.”*

---

### Official Review · Reviewer_T98S · 2025-11-01

**Soundness:** 4
**Presentation:** 4
**Contribution:** 3
**Rating:** 6
**Confidence:** 3

**Summary:**

The paper presents CHAMMI-75, a curated dataset of multi-channel microscopy images from 75 studies with standardized formats, and rich metadata for content-aware sampling. The authors focus on data acquisition, metadata integration, and redundancy reduction, then use the dataset to pre-train ViT models. Benchmarks span five tasks, including two newly constructed evaluations. A DINO-based BoC ViT-small pretrained on CHAMMI-75 delivers the strongest SSL results across most tasks, shows favorable data/model scaling trends, and outperforms models trained on narrower sources.

**Strengths:**

- The dataset has unmatched scale and heterogeneity for multi-channel microscopy, which supports broad generalization.
- The benchmarks cover diverse, realistic tasks and include novel channel combinations, which enhances external validity.
- The scaling analysis is careful and practical, which offers guidance on data size, model size, and multi-channel strategy.
- The SSL results are consistently strong across tasks, which validates CHAMMI-75 as a useful pre-training resource. And the release plan supports reproducibility and downstream use.

**Weaknesses:**

- The work offers limited methodological novelty, which centers contributions on data and benchmarking.
- The comparison to SubCell mixes training regimes and model sizes, which hinders clean conclusions about capability gaps.
- The LLM-assisted metadata extraction lacks error quantification, which raises concerns about label noise in curation.

**Questions:**

Please refer to my weakness section.

---

> ### Author Response · Authors · 2025-11-26
> **Reply to Reviewer T98S [Part 1]**
>
> We thank the reviewer for the constructive feedback, and for offering comments and suggestions to improve the clarity and experimental validations of our study. We have implemented all the recommendations, which has resulted in a higher quality manuscript. We discuss the detailed responses below.
>
> > **W1** The work offers limited methodological novelty, which centers contributions on data and benchmarking.
>
> We thank the reviewer for their fair assessment and recognition that the paper’s main contribution is in data curation and benchmarking. As Reviewer q2zP succinctly stated,
>
> > I understand this is a dataset and benchmark paper so the method part is not the research focus.
>
> We concur and would like to respectfully re-emphasize that this work was submitted to the Datasets and Benchmarks topic. The purpose of this submission category is to introduce resources &mdash;like CHAMMI-75&mdash; that are crucial for accelerating machine learning research and unlocking new research directions by providing standardized, large-scale data and evaluation protocols.
>
> The creation of CHAMMI-75 addresses a fundamental bottleneck in bioimage analysis: the lack of a large-scale, heterogeneous, and multi-channel dataset necessary to train and test generalizable "foundation models" for cellular imaging. By focusing on this challenging data curation effort, we aim to provide the essential prerequisite for the next wave of methodological research in channel-adaptive, multi-scale, and robust bioimage models. Our experimental pipeline, which relies on existing models like DINO-BOC and Channel-ViT, serves precisely to demonstrate the immediate and strong utility of CHAMMI-75 as a pre-training resource.
>
> We have added the following paragraph to the Introduction Section to clarify the contribution of our work (Section 1, Line numbers 90-101):
>
> *“This work addresses the data gap hindering the development of generalizable models for cellular imaging. Progress in machine learning is often catalyzed not only by algorithmic novelty but by the rigorous, large-scale data curation efforts that enable it. ImageNet [A] and LAION [B] are prominent examples that fundamentally shifted the focus of representation learning research and supported breakthroughs that would have been impossible without such data. In computational biology, challenging and realistic data is similarly necessary to advance the field. While significant progress has been achieved with fixed-channel image models, many open problems still exist to achieve a general understanding of cellular states regardless of the imaging technology. CHAMMI-75 is the first resource to integrate heterogeneous multi-channel imaging data at this scale in a way that directly facilitates the investigation of these challenges.”*
>
>
> [A] Jia Deng, Wei Dong, Richard Socher, Li-Jia Li, Kai Li, and Li Fei-Fei. Imagenet: A large-scale hierarchical image database. In 2009 IEEE conference on computer vision and pattern recognition, pp. 248–255. Ieee, 2009.
>
> [B] Christoph Schuhmann, Romain Beaumont, Richard Vencu, Cade Gordon, Ross Wightman, Mehdi Cherti, Theo Coombes, Aarush Katta, Clayton Mullis, Mitchell Wortsman, et al. Laion-5b: An open large-scale dataset for training next generation image-text models. Advances in neural information processing systems, 35:25278–25294, 2022.
>
> > **W2** The comparison to SubCell mixes training regimes and model sizes, which hinders clean conclusions about capability gaps.
>
> We agree with the reviewer that comparing directly to SubCell, given the differences in training regime and model size (WSL, larger ViT-Base vs. our SSL, ViT-Small), makes it difficult to draw clean conclusions about capability gaps. To clarify, SubCell was included purely as a top-line reference model, pre-trained on a fixed-channel dataset (HPAv23) using superior biological supervision, to establish the current performance ceiling in the field. Comparing our models against this state-of-the-art reference gives us a necessary glance at how far competitive SSL models, trained on heterogeneous data, are from the most highly optimized, specialized models.
>
> Crucially, our paper explicitly investigates the impact of learning algorithm, channel strategy, model size, and training regime through separate, controlled experiments:
>
> - SSL algorithm: we evaluate DINO vs SimCLR vs MAE, representing different algorithmic strategies for self-supervised learning.
>
> - Model size: we show performance scaling across ViT-small, base, and large for fixed data (Figure 8b).
>
> - Channel strategy: we compare bag of channels (BoC) and multi-channel attention (MCA) models directly (Figure 8a, 8b).
>
> - Training regimes: we evaluate SSL (Figure 8a) and WSL (Table 2) on the same ViT-small/MCA architectures.
>
> (continued in the second comment)

---

> ### Author Response · Authors · 2025-11-26
> **Reply to Reviewer T98S [Part 2]**
>
> > **W2** (continued)
>
> Continued from the last comment.
>
> These controlled analyses allow for a clean interpretation of the underlying capabilities of models trained with CHAMMI-75, independent of the SubCell comparison. We realized that an explicit discussion of these results and their interpretation in context was not provided; therefore, we have added a new paragraph in Section 5.6 (Additional Analysis, line numbers 493-506) to clarify these points. We would also like to point to Section 5.1 Benchmarking Experiments (line numbers 347-377) which talks about these points.
>
> *“**Section 5.1 (Benchmarking Experiments)** ”*
>
> *“**Baselines.** We consider state-of-the-art pre-trained models that have been recently released for cellular image analysis. We start with SubCell [A], a suite of ViT-base models trained with the HPAv23 dataset using multi-objective, weakly supervised learning. SubCell has four fixed-channel models (one 2ch, two 3ch, one 4ch) trained in two modes (MAE-Cells, ViT-ProtS). The eight SubCell models exhibit excellent performance in downstream tasks; however, their usage requires manual configuration to decide a channel combination and model type. The variation of results between SubCell models is substantial (Appendix D.1.1), making its usage challenging and computationally expensive to test in practice. We also evaluate OpenPhenom [B], a channel adaptive ViT-small model trained on five and six-channel Cell Painting images, and DINOv2 [C], which is a fixed RGB channel model adapted for multi-channel images using BoC. Finally, we trained a model with the best configuration found in the scaling evaluation but adapted for IDRCell100K [D], a multi-channel microscopy image dataset
> close to ours in number of sources (79 vs 75) but smaller (100k multi-channel images).”*
>
> *“**Results.** Table 1 shows pre-training channel-adaptive architectures with SSL yields mod-
> els that are generally useful in many tasks, regardless of the number of channels. SubCell
> sets top-line results across several benchmarks; its strong performance may be explained by
> factors such as training with biological objectives, larger ViT models, channel specializa-
> tion, and manual selection of best results across their different settings. Our BoC model
> trained with CHAMMI-75 obtained the best performance in six out of seven benchmarks,
> demonstrating generalization in tasks with varying channel configurations. The same model
> trained with IDRCell100k (a multi-channel microscopy image dataset) underperforms in
> most tasks, suggesting that CHAMMI-75 contains additional informative images and higher
> quality data for learning. OpenPhenom also underperforms in several tasks, and while it is
> channel adaptive, it was trained exclusively with Cell Painting data (RxRx & JUMP-CP)
> using MAE. Overall, our model exhibits strong performance in challenging tasks thanks to
> a combination of simple methods and high-quality, well-curated data.”*
>
>
>
> **Enhancement to the Discussion of Results in Section 5.6 (Additional Analysis):**
>
> *“Our experimental analysis was designed to isolate the impact of core design choices under the self-supervised pre-training regime, which is a necessary path for leveraging large, heterogeneous dataset without consistent supervised labels. Our observations suggest the following key factors impacting performance: (1) multi-channel strategy: the BoC approach proved to be more effective and scalable under the SSL regime, yielding up to 19% relative improvement over the MCA strategy. MCA also has a high computational cost and sequence length complexity, which makes it 3X to 5X more resource-intensive (Figure B11) and difficult to train effectively without supervision. (2) SSL method: among the tested methods, DINO was the top-performing algorithm, showing a 15% relative improvement over MAE and a 7% relative improvement over SimCLR (Figure 8a). (3) Model size: we observed consistent performance gains from model scaling. Moving from ViT-small to ViT-large models resulted in a relative improvement of 9% to 11% (Figure 8b). (4) The use of weak supervision, even with noisy labels, improved performance by 1-19% relative to the baseline in the MCA setting. These analyses provide guidance for future methodological research on multi-channel models as the field continues to bridge the gap between self-supervised and supervised performance.”*

---

> > ### Author Response · Authors · 2025-11-26
> > **Reply to Reviewer T98S [Part 3]**
> >
> > [A] Ankit Gupta, Zoe Wefers, Konstantin Kahnert, Jan N Hansen, Will Leineweber, Anthony
> > Cesnik, Dan Lu, Ulrika Axelsson, Frederic Ballllosera Navarro, Theofanis Karaletsos, et al.
> > Subcell: Vision foundation models for microscopy capture single-cell biology. bioRxiv, pp.
> > 2024–12, 2024.
> >
> > [B] Oren Kraus, Kian Kenyon-Dean, Saber Saberian, Maryam Fallah, Peter McLean, Jess Le-
> > ung, Vasudev Sharma, Ayla Khan, Jia Balakrishnan, Safiye Celik, et al. Masked au-
> > toencoders for microscopy are scalable learners of cellular biology. In Proceedings of the
> > IEEE/CVF Conference on Computer Vision and Pattern Recognition, pp. 11757–11768,
> > 2024.
> >
> > [C] Maxime Oquab, Timothée Darcet, Théo Moutakanni, Huy Vo, Marc Szafraniec, Vasil Khali-
> > dov, Pierre Fernandez, Daniel Haziza, Francisco Massa, Alaaeldin El-Nouby, et al. Dinov2:
> > Learning robust visual features without supervision. arXiv preprint arXiv:2304.07193,
> > 2023.
> >
> > [D] Nicolas Bourriez, Ihab Bendidi, Ethan Cohen, Gabriel Watkinson, Maxime Sanchez, Guil-
> > laume Bollot, and Auguste Genovesio. Chada-vit: Channel adaptive attention for
> > joint representation learning of heterogeneous microscopy images. In Proceedings of the
> > IEEE/CVF Conference on Computer Vision and Pattern Recognition, pp. 11556–11565,
> > 2024.
> >
> > > **W3** The LLM-assisted metadata extraction lacks error quantification, which raises concerns about label noise in curation.
> >
> > We thank the reviewer for pointing this out. We want to emphasize that we did not use LLM outputs directly in any metadata entries. Rather, LLMs served solely as assistive tools to facilitate the metadata parsing workflow. All metadata in the final version of our dataset was obtained programmatically using deterministic rules and then manually curated and validated. Thus, any errors present in the metadata would be human errors rather than LLM-generated errors. While we cannot claim our metadata is entirely error-free, as is the case with any human-curated dataset, the manual validation process ensures that systematic LLM-related biases or hallucinations are not present in the final dataset.
> >
> > We have clarified this in the main text as follows at line numbers 233-238 :
> >
> > *“Most studies have a scientific publication that describes experimental details, and for part of the metadata preparation, we used large-language models to assist with the identification and organization of certain information. The final metadata was parsed programmatically using deterministic rules and then manually curated and validated.”*

---

### Author Response · Authors · 2025-12-03
**Summary of Rebuttal and Discussion**

Dear Area Chair,

We sincerely thank the reviewers for their thoughtful and constructive feedback. Their comments and suggestions were useful for improving the quality, clarity, and completeness of our work. We also appreciate the area chair for the time and consideration. Below, we summarize the main points of consensus and explain how we implemented the changes and additional experiments to address the reviewers’ concerns.
___

## Consensus on Strengths

1. **Dataset with unmatched scale**. Reviewer T98S considered our dataset to have unmatched scale and heterogeneity. Reviewer 6K5j notes that the “dataset’s large scale is impressive”. Reviewer q2zP also agrees that our dataset provides “unprecedented scale and diversity”.

2. **Valuable benchmark**. Reviewer T98S acknowledges that the proposed benchmarks are diverse, and realistic. Reviewer 6K5j notes that our benchmarks across different microscopy tasks are useful to evaluate generalization. Reviewer q2zP states that the proposed work presents a valuable benchmark for future research.

3. **Thorough experimental evaluation**. Reviewers T98S and q2zP found our experimental evaluation of SSL methods to be careful, practical, and thorough, serving as guidance for future research in multi-channel modeling.

4. **Useful and rich dataset**. Reviewers T98S and q2zP noted that the results presented in this manuscript validate and confirm CHAMMI-75 as a useful pre-training resource. Reviewer 6K5j noted that the proposed dataset has valuable metadata, cell segmentations, and other resources useful to investigate channel-adaptive models.

___

## Response to Main Concerns

1. **Paper scope and contribution (T98S, q2zP)**. We clarify that our paper is presented for the Datasets and Benchmarks category, and highlight the importance of high-quality data as a fundamental contribution for advancing machine learning methods.

2. **Cross-modality and channel generalization (6K5j, q2zP)**. Our original benchmark included a novel and unique dataset with 14 channels to evaluate generalization to novel channel combinations. During the rebuttal period, we added a new benchmark with bright-field images to evaluate cross-modality transfer. The results indicate that models trained in our dataset outperform all baselines in these generalization tasks.

3. **Factors that drive generalization (q2zP)**. We conducted an ablation study by isolating subsets of the dataset with different technical and biological properties, and trained models with these subsets to compare performance. We found that the diversity of microscopy types is the most important aspect of our dataset to produce models with strong generalization.

4. **Feature disentanglement analysis (6K5j)**. We implemented experimental evaluations to understand how the feature space obtained with a model trained with CHAMMI-75 facilitates the separation of two factors of variation: technical variation (batch, microscopes, etc), and biological variation (phenotypes, treatment effects). We observe that CHAMMI-75 produces models that encode both factors (i.e., no invariance to technical variation found), but facilitate the separation of these factors using batch correction methods, resulting in higher accuracy than baseline alternatives.

5. **Qualitative analysis (6K5j)**. We produced visualizations that compare feature spaces across models trained with different datasets, to facilitate qualitative evaluation of clusters and other properties of the dataset.

6. **Other clarifications**. We improved the explanations in the main manuscript and extended the supplementary material to address other questions related to experimental design (T98S), metadata quality control (6K5j), guidance for fine-tuning models (6K5j), and distinction from segmentation models (6K5j).

___

Finally, we’d like to highlight that reviewer q2zP was willing to raise their score if more analysis was included, according to the initial review comments. After posting our responses and during the brief period of interaction, the reviewer kindly increased the initial score from 4 to 6 (see original comments below). We thank the reviewer for the constructive feedback, which resulted in a higher quality paper, and for recognizing the value and the contributions of our work.

---

### Meta-Review · Area_Chair_2x9N · 2026-01-05

**Summary:**

This paper introduces CHAMMI-75, a large-scale multi-channel microscopy image dataset, and establishes a comprehensive evaluation benchmark. The study aims to provide a high-quality pre-training data resource and a standardized evaluation framework for the bioimage analysis community, particularly for cell phenotyping tasks, thereby facilitating the development of robust "foundation models" with strong generalization capabilities.

All three reviewers acknowledged the core value of this work, unanimously agreeing that:
1）The dataset itself is highly valuable；
2）The benchmark design is comprehensive；
3）The experimental analysis is rigorous and thorough.

Simultaneously, reviewers raised key concerns primarily regarding the clarity of the paper's contribution, the depth of validation for model generalization, and the analysis of key factors driving generalization performance.

In the rebuttal, the authors provided detailed and targeted responses, supplementing several experiments and analyses that effectively addressed the reviewers' questions. Based on the authors' comprehensive and convincing rebuttal, Reviewer q2zP raised their score from "marginally below the acceptance threshold" to "marginally above the acceptance threshold." The final opinions from all reviewers supported or did not oppose the acceptance of the paper.

The revised manuscript has adequately addressed and resolved the main concerns raised during the review process. The CHAMMI-75 dataset demonstrates high standards in scale, diversity, and quality. Its accompanying comprehensive benchmark system is well-designed, and the experimental analysis is solid. This work provides a much-needed, standardized, large-scale pre-training data platform for the bioimage analysis field, offering clear and significant value for advancing the domain.

**Reviewer Concerns:**

The authors have clarified the core contribution of this paper as a "Datasets and Benchmarks" study, emphasizing the foundational role of high-quality data as infrastructure. The analysis of generalization has been significantly deepened: a new cross-modality benchmark was introduced to validate the model's transfer capability to bright-field images; ablation studies conclusively identified imaging modality diversity as the primary driver of generalization; and supplementary feature disentanglement analysis and visualization enhanced the understanding of the learned representations. Furthermore, sufficient clarifications were provided on technical details such as metadata quality control, fine-tuning guidance, and the distinction from segmentation models.

Minor refinements for the final version:
A reviewer suggested more thorough integration of some newly added analyses (e.g., the drivers of generalization) into the main text to optimize the narrative flow. In conclusion, the revised paper demonstrates solid scientific contributions and comprehensive experimental analysis, having adequately addressed all review comments.

**Reviewer Scores:**

Reviewer T98S: 6 → 7
The author's response satisfactorily addressed concerns about contribution scope, experimental comparisons, and metadata quality. By clearly framing this as a "Datasets and Benchmarks" paper emphasizing data infrastructure value, the revision justifies moving from marginal to clear acceptance.

Reviewer 6K5j: 8 → 9
The rebuttal comprehensively addressed all raised points through substantial additions including cross-modality benchmarking, feature disentanglement analysis, and ablation studies. These enhancements, directly responding to reviewer suggestions, elevate the paper from good to strong benchmark quality.

Reviewer q2zP: 4→6→7
The added generalization analysis (ablation studies, cross-modality benchmark) effectively resolved core concerns, as reflected in the rebuttal score increase. With better integration of new material into the main text as suggested, the paper would achieve stable acceptance.

---

### Decision · Program_Chairs · 2026-01-26

Accept (Poster)